# Toroidal topology of population activity in grid cells

Richard J. Gardner[1,6 ✉], Erik Hermansen[2,6], Marius Pachitariu[3], Yoram Burak[4,5], Nils A. Baas[2 ✉], Benjamin A. Dunn[1,2 ✉], May-Britt Moser[1] & Edvard I. Moser[1 ✉]

The medial entorhinal cortex is part of a neural system for mapping the position of an individual within a physical environment[1]. Grid cells, a key component of this system, fire in a characteristic hexagonal pattern of locations[2], and are organized in modules[3] that collectively form a population code for the animal's allocentric position[1]. The invariance of the correlation structure of this population code across environments[4,5] and behavioural states[6,7], independent of specific sensory inputs, has pointed to intrinsic, recurrently connected continuous attractor networks (CANs) as a possible substrate of the grid pattern[1,8–11]. However, whether grid cell networks show continuous attractor dynamics, and how they interface with inputs from the environment, has remained unclear owing to the small samples of cells obtained so far. Here, using simultaneous recordings from many hundreds of grid cells and subsequent topological data analysis, we show that the joint activity of grid cells from an individual module resides on a toroidal manifold, as expected in a two-dimensional CAN. Positions on the torus correspond to positions of the moving animal in the environment. Individual cells are preferentially active at singular positions on the torus. Their positions are maintained between environments and from wakefulness to sleep, as predicted by CAN models for grid cells but not by alternative feedforward models[12]. This demonstration of network dynamics on a toroidal manifold provides a population-level visualization of CAN dynamics in grid cells.

The idea of a CAN has become one of the most influential concepts in theoretical systems neuroscience[13–15]. A CAN is a network in which recurrent synaptic connectivity constrains the joint activity of cells to a continuous low-dimensional repertoire of possible coactivation patterns in the presence of a wide range of external inputs. Few systems are more suitable for analysis of CAN dynamics than the spatial mapping circuits of the rodent brain, owing to the continuous, low-dimensional nature of space, and the availability and interpretability of data from these circuits[1–6]. In medial entorhinal cortex (MEC) and surrounding areas, head direction cells[16] encode orientation whereas grid cells[2] encode position. CAN models conceptualize the neural representations of these variables as spanning periodic one- or two-dimensional (1D or 2D) continua on a ring[17–19] or a torus[1,8–11], respectively. In this scheme, activity within the neural network stabilizes as a localized bump when cells are ordered according to their preferred firing directions or locations in physical space. The activity bump may be smoothly translated along the network continuum by speed and direction inputs, or by external sensory cues.

In agreement with CAN models[1,8–11], head direction cells[16,20,21] and modules of grid cells[4–7] maintain fixed correlation structures. In head direction cells, cell samples of a few dozen have been sufficient to demonstrate that the network activity traverses a ring[22–24], but for grid cells, the number of possible locations in the two-dimensional

state space has been too large for the topology of the manifold to be uncovered. Here we take advantage of recently developed high-site-count Neuropixels silicon probes[25,26] to determine in many hundreds of simultaneously recorded grid cells whether, as predicted by two-dimensional CAN models[8–11], the population activity in an individual grid-cell module resides on a toroidal manifold, independently of behavioural tasks and states and decoupled from the position of the animal in physical space. We focused on individual modules because (i) these are the unit networks of CAN models[1,8–10]; and (ii) topological analysis of multi-module representations would require even larger numbers of cells[27].

## Visualization of toroidal manifold

We recorded extracellular spikes of a total of 7,671 single units in layers II and III of the MEC–parasubiculum region in freely moving rats with unilateral or bilateral implants (total of 4 recordings, in 2 rats with bilateral single-shank probes and 1 rat with a unilateral 4-shank probe; from 546 to 2,571 cells per recording; Extended Data Fig. 1). During recordings, the rats were engaged in foraging behaviour in a square open-field (OF) enclosure or on an elevated track, or they slept in a small resting box. Using a clustering-based approach, we identified six grid modules across all rats (4 recording sessions, from 140 to 544

[1]Kavli Institute for Systems Neuroscience and Centre for Neural Computation, Norwegian University of Science and Technology, Trondheim, Norway. [2]Department of Mathematical Sciences, Norwegian University of Science and Technology, Trondheim, Norway. [3]HHMI Janelia Research Campus, Ashburn, VA, USA. [4]Edmond and Lily Safra Center for Brain Sciences, The Hebrew University of Jerusalem, Jerusalem, Israel. [5]Racah Institute of Physics, The Hebrew University of Jerusalem, Jerusalem, Israel. [6]These authors contributed equally: Richard J. Gardner, Erik Hermansen. ✉e-mail: richard.gardner@ntnu.no; nils.baas@ntnu.no; benjamin.dunn@ntnu.no; edvard.moser@ntnu.no

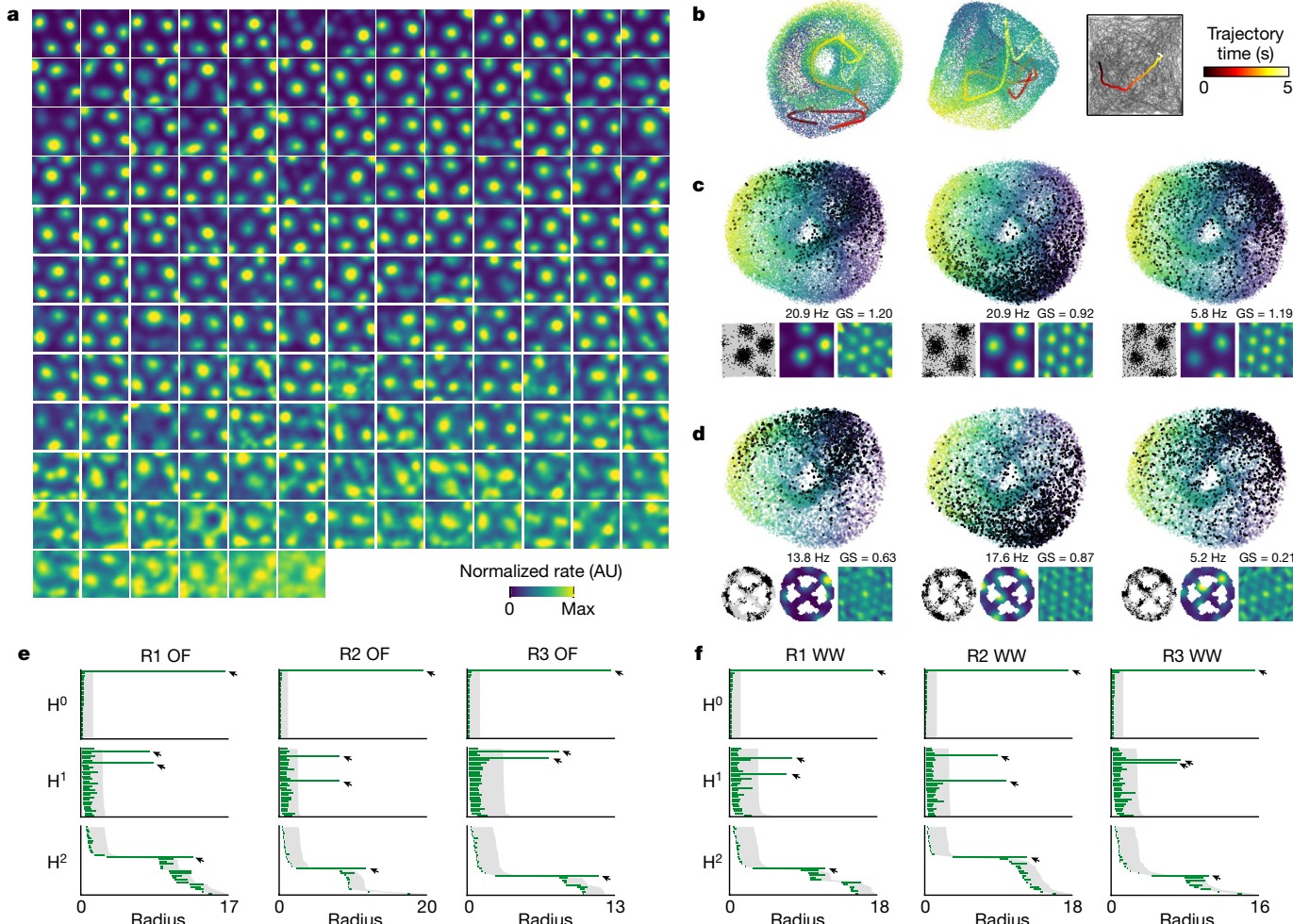

**Fig. 1 | Signatures of toroidal structure in the activity of a module of grid cells. a**, Firing rates of 149 grid cells co-recorded from the same module and shown, in order of spatial information content, as a function of rat position in OF arena (rates colour-coded, max 0.2–35.0 Hz; rat 'R' day 1, module 2; Extended Data Fig. 2b). **b**, Nonlinear dimensionality reduction reveals torus-like structure in the population activity of a single grid module (same 149 cells; 3 different views of same point cloud). Each dot represents the population state at one time point (dots coloured by first principal component). Bold line shows a 5-s trajectory, demonstrating smooth movement over the toroidal manifold. Right, corresponding trajectory in OF. **c**, Toroidal positions of spikes from three grid cells from the module in **a**. Each panel shows the same 3D point cloud of population states as in **b**, with black dots indicating when the cell fired. Insets show: left: the cell's 2D firing locations in OF (black dots on grey trajectory);

middle: colour-coded firing rate map in OF (range 0 to max); right: colour-coded autocorrelogram of the rate map (range −1 to +1). Maximum rate and grid score (GS) are indicated. **d**, Same as in **c** (same cells) but with the rat running on an elevated, wheel-shaped track ('wagon-wheel track'; WW). Note preserved toroidal field locations. **e**, **f**, Barcodes indicate toroidal topology of grid-cell population activity. Results of persistent cohomology analyses (30 longest bars in the first three dimensions: $H^0$, $H^1$ and $H^2$) are shown for three grid modules from one rat (R1–R3 day 1, $n$ = 93, 149 and 145 cells, respectively), in OF (**e**) and WW (**f**). Grey shading indicates longest lifetimes among 1,000 iterations in shuffled data (aligned to lower values of original bars). Arrows show four most prominent bars across all dimensions (all longer than in shuffled data). One prominent bar in dimension 0, two in dimension 1 and one in dimension 2 indicates cohomology equal to that of a torus.

grid cells per session; 7.8% to 25.6% of total number of cells; Extended Data Fig. 2a–d, g, h). Each grid module cluster contained a mixture of nondirectional ('pure') grid cells and conjunctive grid × direction cells[28], from 66 to 189 grid cells per module (total pure and conjunctive grid cells; Extended Data Fig. 2g). We initially limited our analyses to the subset of pure grid cells because (i) the expected toroidal topology might be distorted by additional directional modulation; and (ii) detection of topology in conjunctive cells may require a larger number of cells than recorded here[27].

To visually inspect the structure of the population activity of grid cells for signatures of toroidal topology, we constructed a three dimensional (3D) embedding of the $n$-dimensional population activity of a module of $n$ = 149 pure grid cells (Fig. 1a). For this, we applied a two-stage dimensionality reduction procedure on the matrix of firing rates. First, to improve robustness to noise, we conducted a

principal component analysis (PCA). We retained the first six principal components, which explained a particularly large fraction of the variance for all grid modules in the OF condition (with a similar tendency seen during sleep; Extended Data Fig. 4a). Next, we applied uniform manifold approximation and projection (UMAP) to reduce the six principal components into a 3D visualization. This visualization revealed a torus-like structure (Fig. 1b, Supplementary Video 1). Movement of the rat in the OF was accompanied by similarly continuous movement of the population activity across the toroidal manifold (Fig. 1b). When the activity of individual cells was plotted with reference to the 3D population representation, spikes for each cell were localized within a single patch of the population state space (Fig. 1c). The offsets between the firing locations of individual cells in the arena corresponded with the relative firing locations of the cells in the toroidal state space.

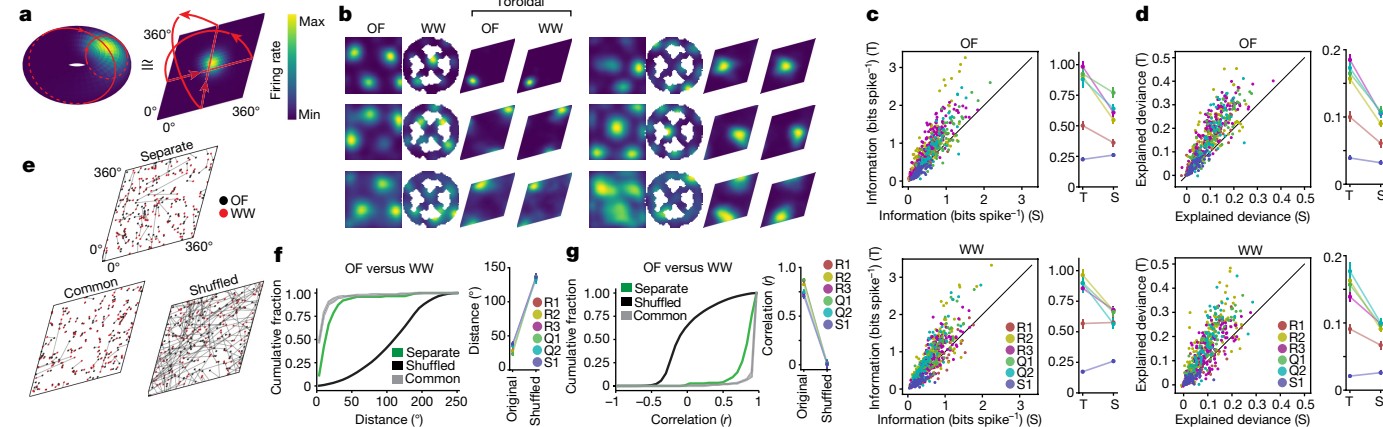

**Fig. 2 | Cohomological decoding of position on an inferred state space torus. a**, **b**, Individual grid cells have distinct firing fields on the inferred torus (Extended Data Fig. 5). Toroidal coordinates for population activity vectors were decoded from the two significant 1D holes (red circles in **a**) in the barcodes in Fig 1e, f. **a**, Left, 3D embedding of the toroidal state space displaying colour-coded mean firing rate of one grid cell as a function of toroidal position. Right, a 2D torus may be formed by gluing opposite sides of a rhombus. **b**, Representative grid cells from module R2 day 1 showing tuning to toroidal coordinates (all R2 cells: Supplementary Fig. 1). Each row of four plots corresponds to one cell. Left to right, colour-coded maps of cells' firing rates across the environment (OF or WW) and on the inferred torus (toroidal OF, toroidal WW, aligned to common axes). **c**, **d**, Toroidal information content (**c**) and explained deviance (**d**) for toroidal position (T) versus spatial position

(S) in OF (top) and WW (bottom). Explained deviance is an $R^2$-statistic (range 0–1) expressing goodness-of-fit of GLM models for S or T. Left, scatterplots with dots showing individual cells; colour indicates module (inset). Right, mean ± s.e.m. for each module. $n$ = 93 (R1), 149 (R2), 145 (R3), 94 (Q1), 65 (Q2) and 73 (S1) cells. **e**, **f**, Distances between toroidal firing field locations. **e**, Field locations of all R2 cells in OF and WW. Lines connect fields of the same cell. Toroidal OF and WW axes were aligned either separately ('separate') or commonly to OF ('common'). **f**, Left, cumulative frequency distribution of field distances (all R2 cells; green curve, separate alignment; grey lines, common alignment (to either OF or WW); black curve, shuffled data, $n$ = 1,000 shuffles). Right, mean distance between field centres (±s.e.m.) for all modules. $n$ cells as in **c**, **d**. **g**, Same as **f**, but showing Pearson correlations between pairs of toroidal rate maps.

## Quantification of toroidal topology

Although the UMAP projection allowed a toroidal point cloud to be visualized, the method does not lend itself to straightforward quantification of the topology of the state space or comparison of representations across experiments. We therefore turned to the framework of persistent cohomology, a toolset from topological data analysis in which the structure of neural data can be classified by identifying holes of varying dimensionality in topological spaces assigned to point clouds of the cells' firing rates[22,23]. In applying this toolset, we replace each point of the point cloud by a ball of common radius. The union of balls results in a topological space in which the number of holes of different dimensions can be counted. By increasing this radius from zero until all the balls intersect, we observe the lifetime of each hole— the range of radii from when the hole first appears until it disappears (see Extended Data Fig. 3C). The lifetimes of the holes are depicted as bars and the totality of bars referred to as the barcode. For a torus, the barcode must display four bars of substantial length: a 0D hole (a single component connecting all points); two 1D holes (describing circular features); and a 2D hole (a cavity; Extended Data Fig. 3B).

Persistent cohomology analyses allowed us to classify the shape of the six-dimensional representation that serves as an intermediate step in UMAP (Extended Data Fig. 3A). We constructed barcodes for each of the six individual modules of grid cells recorded in the OF arena (three modules from rat 'R', 2 from rat 'Q' and 1 from rat 'S', henceforth named R1, R2, R3, Q1, Q2 and S1). The barcodes showed clear indications of toroidal characteristics. For all six modules, we detected four long-lived bars representing a single 0D hole, two 1D holes and a 2D hole. Their lifetimes were significantly longer than the lifetime of any bar obtained in 1,000 shuffles of the data in which spike times were randomly rotated (Fig. 1e, f, Extended Data Fig. 6Aa; $P$ < 0.001). The findings suggest that network dynamics during OF foraging resides on a low-dimensional manifold with the same barcode as a torus. We noted the appearance of additional short bars in the barcodes for all modules, but these are expected for toroidal point clouds[27], as we confirmed with

simulated data from several CAN models[10,11] and point clouds sampled from idealized tori, which in each case exhibited similar features (see Extended Data Fig. 7).

## Tori persist despite grid distortions

The appearance of a torus in the point cloud, and the mapping of the activity of individual grid cells onto the torus (Fig. 1c), are consistent with a relationship between position in 2D physical space and position in the dimensionality-reduced neural state space. However, in many environments, this relationship may not be isometric, as the grid pattern is distorted by geometrical features of the environment, such as walls and corners[29–31] or discrete landmarks and reward locations[32,33]. We thus asked whether such geometric features could similarly distort the toroidal organization of network activity in the point cloud. We tested rats on an elevated running track shaped like a wagon wheel with four radial spokes ('wagon-wheel track' (WW); Fig. 1d, f). Spatial autocorrelation analyses confirmed that the strict periodicity of the grid pattern was compromised in this task (Extended Data Fig. 2e, f). Despite these distortions of the grid pattern in individual cells, toroidal tuning was maintained in the transformed population activity (Fig. 1d). The persistent cohomology analysis continued to identify one 0D hole, two 1D holes and one 2D hole with lifetimes that substantially exceeded those of shuffled data (Fig. 1f, Extended Data Fig. 6Ab). We also determined how the neural population activity mapped onto the torus by calculating angular coordinates from each of the two 1D holes identified by the barcode ('cohomological decoding'; Extended Data Fig. 5). The two angular coordinates defined directions intersecting at 60°, identifiable as a twisted torus (Fig. 2a). Consistent with CAN models, the vast majority of grid cells were tuned to a single location on the torus in each module and across environments, independent of geometry and local landmarks (Fig. 2b, Extended Data Fig. 4f, Supplementary Information).

To test how faithfully location in the environment is mapped onto the toroidal representation, we next asked whether grid-cell activity is predicted better by the cells' tuning to the inferred torus than by their

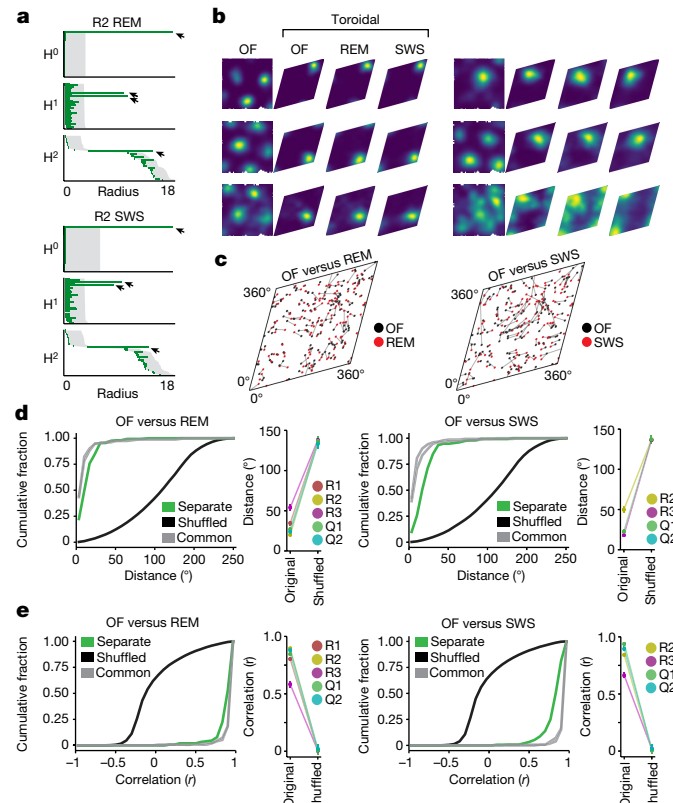

**Fig. 3 | Preservation of toroidal structure during sleep. a**, Barcodes indicating toroidal topology for grid-cell module R2 day 2 ($n = 152$ cells) during REM sleep and SWS (as in Fig. 1e, f). **b**, Toroidal rate maps showing preserved toroidal tuning for individual cells across environments and brain states (as in Fig. 2b; all cells shown in Extended Data Fig. 10). From left: rate map for OF in physical coordinates; and rate maps for OF, REM sleep and SWS in toroidal coordinates. **c**, Distribution of toroidal field centres (as in Fig. 2e) in OF and sleep ($n$ as in a). **d**, **e**, Left, cumulative distributions of distances between toroidal field centres (**d**) and Pearson correlation $r$ values (**e**) of rate maps for all R2 grid cells, as in Fig. 2f, g, but comparing OF with REM or SWS. Right, mean value ± s.e.m. for all modules. $n = 111$ (R1), 152 (R2), 165 (R3), 94 (Q1), 65 (Q2) and 72 (S1) cells. $n = 1,000$ shuffles.

tuning to physical space. For five out of six grid modules in OF and four out of six in WW, the information content conveyed about position, in bits per spike, was higher for position on the torus than for position in physical space (Fig. 2c; R2, R3, Q1, Q2: all $P < 0.001$, $W > 1,932$ in OF and WW; R1: $P < 0.001$, $W = 4,010$ in OF, $P = 0.586$, $W = 2,129$ in WW; S1: $P = 1.000$ in OF and WW, $W = 620$ in OF, $W = 129$ in WW; Wilcoxon signed-rank test). We verified this difference by comparing the cross-validated prediction of two Poisson generalized linear model (GLM)-based encoding models of each cell's activity that included toroidal position (decoded as above) and 2D spatial position. For both environments (OF and WW), the toroidal covariate was closer to a perfectly fitted model of the data than was the physical position covariate in five out of six grid-cell modules (Fig. 2d; R1, R2, R3, Q1, Q2: $P < 0.001$, $W > 2,045$ in OF and WW; S1: $P < 0.001$, $W = 1,941$ in OF, $P = 1.000$, $W = 727$ in WW; Wilcoxon signed-rank test). Together, these differences point to toroidal structure as the primary feature of the population activity of grid cells, superior to that of the 2D coordinates of the animal's current position in the physical environment.

If grid cells operate on a toroidal manifold determined by intrinsic network features, this manifold may be expressed universally across environments, independently of sensory inputs. We tested this proposition by assessing, on the inferred tori, whether the locations of firing fields of different grid cells were maintained between OF and WW

(Fig. 2b, Supplementary Information). To compare the toroidal parametrizations, we aligned the axes of the toroidal coordinates (Extended Data Fig. 5b). First, we compared, for each cell, the distance between the centres of mass of the toroidal rate maps in OF and WW (Fig. 2e, f, Extended Data Fig. 6Ba). This distance was substantially shorter (mean ± s.e.m. of mean distances for all modules: 31.5 ± 6.3 degrees) than that of control data in which the order of the rate maps in one environment was shuffled (135.8 ± 1.7 degrees; maximum possible distance $\sqrt{2} \cdot 180 \approx 254.6$ degrees; data versus shuffled: $P < 0.001$ in all modules). Second, we calculated the pairwise Pearson correlations of binned toroidal rate maps across the two environments (Fig. 2g, Extended Data Fig. 6Ba). Consistent with the centre-of-mass comparison, the correlations between OF and WW were higher in observed data (mean ± s.e.m. of mean $r$ values for all modules: 0.79 ± 0.07) than in shuffled data ($r = 0.01 ± 0.01$; $P < 0.001$ for all modules). Very similar results were obtained when applying the toroidal parametrization from the same environment (either OF or WW) to activity from both environments (Fig. 2f, g, 16.0 ± 3.4 degrees; $r = 0.95 ± 0.02$; $P < 0.001$ for all modules and both mappings). Together, these findings suggest that physical space is mapped onto the same internal low-dimensional manifold irrespective of the specific environment.

## Toroidal topology persists during sleep

If population activity is mapped onto the same toroidal manifold independently of sensory inputs, the toroidal topology should also be maintained during sleep. To test this idea, the rats rested in a high-walled, opaque box placed in the centre of the OF or WW track. Periods of rapid-eye-movement (REM) sleep and slow-wave sleep (SWS) were identified on the basis of the low-frequency rhythmic content of the aggregated multi-unit activity in combination with prolonged behavioural immobility (Extended Data Fig. 9).

Persistent cohomology analysis of the sleep population activity suggested toroidal topology in five of the six grid modules during REM and four out of six modules during SWS (modules R2, R3, Q1 and Q2 for both sleep stages and module R1 only in REM; Fig. 3a, Extended Data Fig. 6Ac, d). In the remaining module (S1), there were no long-lived bars in dimensions 1 or 2 (Extended Data Fig. 6Ac, d), indicating an absence of toroidal structure during sleep, perhaps because of an insufficient number of cells in this module (72 cells; Extended Data Fig. 4e). The barcode results were supported by the toroidal mapping, which revealed sharply tuned firing fields on the REM and SWS tori (99.3 ± 1.6% and 99.1 ± 1.8%, respectively, of the grid cells in each module had higher information content than shuffled data, and in 95.3 ± 7.2% and 98.6 ± 2.4% of cells the toroidal tuning explained the activity better than a null model that assumes a constant firing rate; Fig. 3b, Extended Data Figs. 6C, 10, Supplementary Information). In addition, the spatial arrangements of toroidal firing locations of different cells were maintained between wake, REM and SWS states (Fig. 3c, Extended Data Fig. 6Bb, c). For between-condition pairs of rate maps, the mean distance (±s.e.m.) between the peak firing locations (OF versus REM 31.5 ± 15.4 degrees, OF versus SWS 29.8 ± 14.3 degrees) was well below the distribution of shuffled distances (Fig. 3d, Extended Data Fig. 6Bb, c; 135.8 ± 2.3 degrees in both REM and SWS, $P < 0.001$ for all 5 and 4 modules, respectively). Similarly, the mean correlations of pairs of toroidal rate maps (REM versus OF $r = 0.80 ± 0.15$, SWS versus OF $r = 0.83 ± 0.12$) were substantially larger than in shuffled versions of the data (Fig. 3e, Extended Data Fig. 6Bb, c; $r = 0.01 ± 0.01$ in both REM and SWS, $P < 0.001$ for all 5 and 4 modules, respectively). Thus, the toroidal structure is maintained in both sleep conditions, despite the lack of external spatial inputs.

## Classes of grid cells

We next investigated why toroidal structure was not visible during REM in module S1 and during SWS in modules R1 and S1 (Fig. 4a, Extended

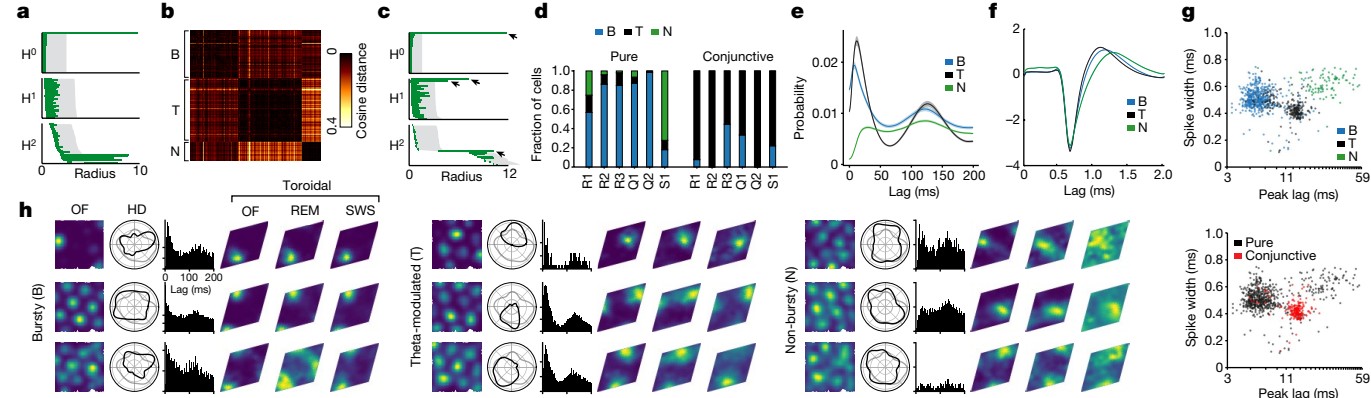

**Fig. 4 | Differential toroidal tuning of grid-cell subpopulations. a**, Barcode of all pure R1 grid cells (day 2, $n = 111$ cells) does not indicate toroidal structure during SWS. **b**, Matrix of cosine distances between pairs of spike-train autocorrelograms of grid cells in module R1. Rows and columns show 189 grid cells (pure and conjunctive) sorted by cluster identity. Three clusters were identified, appearing as dark (that is, similar) squares along the matrix diagonal. On the basis of temporal firing patterns (**e**), they were named 'bursty' (B), 'theta-modulated' (T) and 'non-bursty' (N). **c**, Barcode of the 'bursty' class of R1 ($n = 69$ cells) indicates toroidal structure. Symbols as in **a**. Arrows point to the four most persistent features. **d**, Fractions of grid cells in each class, shown for each grid module. Left, pure grid cells only, right, conjunctive grid × head-direction cells only. For $n$ see Extended Data Fig. 2g. **e**, Average temporal autocorrelogram for cells in each class. Shaded area shows mean ± s.e.m. (bursty $n = 523$, theta-modulated $n = 229$, non-bursty $n = 95$ cells). For each class, note short-latency peak (burst-firing) and long-latency peak (theta-modulation). **f**, Average spike template waveforms of cells from each class ($n$ as in **e**). Shaded area indicates mean ± s.e.m. **g**, Cell classes have different burst-firing characteristics, as expressed by latency of first autocorrelogram peak ($x$ axis) and peak-to-peak spike width ($y$ axis). Cells (dots) are colour-coded by class ($n$ as in **e**) or by identity (pure or conjunctive, $n = 659$ or 188, respectively). **h**, Example cells from each class (one row of plots per cell). Plots from left to right: OF firing rate map; head-direction (HD) tuning curve (black) compared to occupancy of head directions (light grey); temporal autocorrelogram; toroidal firing rate maps for OF, REM and SWS.

Data Fig. 6Ad). Previous studies of medial entorhinal spiking activity have described cell populations with distinct burst-firing and theta-modulation characteristics[34–36]; therefore, we asked whether a lack of toroidal structure was due to heterogeneity in the composition of the module. We quantified each cell's temporal modulation characteristics using the spike train temporal autocorrelogram from the OF session, and by applying clustering to the matrix of autocorrelograms we obtained three cell classes (Fig. 4b). Each class was distributed across multiple modules (Fig. 4d). Within each module, cells from the three classes showed overlapping grid spacing and orientation properties (Extended Data Fig. 8a). We named the classes 'bursty' (B), 'non-bursty' (N) and 'theta-modulated' (T), following the most prominent autocorrelogram feature of each class (Fig. 4e). We also examined the spike waveforms of the cells, and found that each class showed a characteristic spike width (Fig. 4f, g), suggesting that they differ in morphology or biophysical properties.

The firing rates of the cells during SWS exhibited marked correlation structure within—but not between—classes (Extended Data Fig. 8b). Even though our classification strategy was not influenced by the cells' directional tuning, class T contained 80% of all conjunctive grid cells and only 11% of all pure grid cells, supporting the idea that conjunctive grid cells are a distinct population. Accordingly, in modules R1 and S1, which contained the largest numbers of T cells, pairwise correlations of T cells' spike trains were more strongly related to head-direction tuning than to toroidal tuning (Extended Data Fig. 8c). When we subdivided module R1 into the three classes (Fig. 4b), we found that during SWS toroidal topology was detectable only in B cells (Fig. 4c). By decoding toroidal position from B cells, we were able to recover the selectivity of each cell with respect to toroidal position in module R1 (Fig. 4h). The toroidal tuning locations were preserved between OF and SWS in each cell class in R1 (Extended Data Fig. 8d, B: distance of 26.4 ± 6.1 degrees and correlation of $r = 0.85 ± 0.02$, T: 43.6 ± 3.9 degrees and $r = 0.74 ± 0.02$, N: 29.9 ± 3.5 degrees and $r = 0.80 ± 0.02$; mean values of shuffled versions of each class were between 135.4 ± 5.2 and 136.4 ± 6.2 degrees, and between $r = 0.00 ± 0.07$ and $r = 0.02 ± 0.03$; comparison between observed and shuffled $P < 0.001$ for all 3 classes and both measures). However, in R1 as well as all other modules, toroidal spatial

information and explained deviance were highest for B cells and lower for N and T cells in OF, REM and SWS (Extended Data Fig. 8e) (information content: $P < 10^{-56}$, $H > 255$; Kruskal–Wallis test; $P < 10^{-9}$, $Z > 6.4$; Dunn test with Bonferroni correction; explained deviance: $P < 10^{-20}$, $H > 96$; Kruskal–Wallis test; $P < 10^{-12}$, $Z > 7.4$; Dunn test with Bonferroni correction, for OF, REM and SWS). Collectively, these results show that the B cell population (containing the majority of our grid cells) represents the torus most robustly across behavioural conditions. The weaker toroidal representation in T cells may partly be an effect of the higher dimensionality of the code carried by conjunctive grid × direction cells. Indeed, running cohomology analysis on T cells from modules S1 and R1 (which contained the most T cells) revealed a circular feature that corresponded to the animal's head direction (Extended Data Fig. 8f, g).

## Discussion

Our findings, from many hundreds of simultaneously recorded grid cells, show that population activity in grid cells invariably spans a manifold with toroidal topology, with movement on the torus matching the animal's trajectory in the environment. The toroidal representation was most stably encoded by the bursting subclass of grid cells. Toroidal topology was not simply inherited from the encoded variable, as 2D space is not characterized by toroidal topology, as opposed to pitch and azimuth of head orientation, which in bats together span a torus and thus naturally map onto a toroidal neural code[37]. Using cohomological decoding, we were able to demonstrate, in each environment and in both sleep and awake states, that the toroidal coordinates of individual grid cells in individual grid modules were maintained, independently of external sensory inputs or environment-induced deformations of hexagonal symmetry in the rate maps[29–33]. The uniform and consistent toroidal structure of the manifold suggests that distortions in grid patterns occur in the mapping between physical space and the toroidal grid code rather than in the grid code itself.

The invariance of the toroidal manifold across environments and brain states is informative about the mechanisms that underlie grid-cell activity. Although toroidal topology can be generated by both CAN[1,8–10] and feedforward[12] mechanisms, the persistence of an invariant toroidal

manifold under conditions that give rise to changes in the correlation structure of place-cell activity in the hippocampus[6,7] is predicted only by CAN models. While the findings do not exclude co-existing feedforward mechanisms[12,38], they point to intrinsic network connectivity as the mechanism that underlies the rigid toroidal dynamics of the grid-cell system. What kind of network architecture keeps the activity on a toroidal manifold—whether it is geometrically organized[1,8–10] or acquired from random connectivity by synaptic weight adjustments through learning[39–41]—remains to be determined, as does the mode of connectivity with other CANs in the entorhinal–hippocampal system[22,23].

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

# Methods

## Rats

The data were collected from three experimentally naive male Long Evans rats (Rats Q, R and S, 300–500 g at time of implantation). The rats were group-housed with three to eight of their male littermates before surgery and were singly housed in large Plexiglas cages (45 × 44 × 30 cm) thereafter. They were kept on a 12-h light–12-h dark schedule, with strict control of humidity and temperature. All procedures were performed in accordance with the Norwegian Animal Welfare Act and the European Convention for the Protection of Vertebrate Animals used for Experimental and Other Scientific Purposes. Protocols were approved by the Norwegian Food Safety Authority (FOTS ID 18011 and 18013).

## Electrode implantation and surgery

The rats were implanted with Neuropixels silicon probes[25,26] targeting the MEC–parasubiculum (PaS) region. Two rats were implanted bilaterally with prototype Neuropixels 'phase 3A' single-shank probes and with one probe targeting MEC–PaS in each hemisphere; the third rat was implanted with a prototype Neuropixels 2.0 multi-shank probe in the left hemisphere. Probes were inserted at an angle of 25° from posterior to anterior in the sagittal plane. Implantation coordinates were AP 0.05–0.3 mm anterior to the sinus and 4.2–4.7 mm lateral to the midline. The probes were inserted to a depth of 4,200–6,000 µm. The implant was secured with dental cement. The detailed implantation procedure has been described elsewhere[6,26]. After surgery, the rats were left to recover for approximately 3 h before beginning recording. Postoperative analgesia (meloxicam and buprenorphine) was administered during the surgical recovery period.

## Recording procedures

The details of the Neuropixels hardware system and the procedures for freely moving recordings have been described previously. In brief, electrophysiological signals were amplified with a gain of 500 (for phase 3A probes) or 80 (for 2.0 probes), low-pass-filtered at 300 Hz (phase 3A) or 0.5 Hz (2.0), high-pass-filtered at 10 kHz, and then digitized at 30 kHz (all steps performed by the probe's on-board circuitry). The digitized signals were multiplexed by an implant-mounted 'headstage' circuit board and were transmitted along a lightweight 5-m tether cable, made using either micro-coaxial (phase 3A) or twisted pair (2.0) wiring.

Three-dimensional motion capture (OptiTrack Flex 13 cameras and Motive recording software) was used to track the rat's head position and orientation, by attaching a set of five retroreflective markers to implant during recordings. The 3D marker positions were projected onto the horizontal plane to yield the rat's 2D position and head direction. An Arduino microcontroller was used to generate digital pulses, which were sent to the Neuropixels acquisition system (via direct TTL input) and the OptiTrack system (via infra-red LEDs), to permit precise temporal alignment of the recorded data streams.

## Behavioural procedures

Data were obtained from four recording sessions performed within the first 72 h after recovery from surgery. The recordings were performed while the rats engaged in three behavioural paradigms, each in a different arena within the same room. Abundant distal visual and sonic cues were available to the rat. On each day of recording, the rat remained continuously connected to the recording apparatus across the various behavioural sessions that were performed. Occasionally it was necessary to remove twists that had accumulated in the Neuropixels tether cable. In such cases, the ongoing behavioural task was paused while the experimenter gently turned the rat to remove the twists. During pre-surgical training, the rats were food-restricted, maintaining their weight at a minimum of 90% of their free-feeding body weight. Food was generally removed 12–18 h before each training session. Food restriction was not used at the time of recording.

## Open-field foraging task

Rats foraged for randomly scattered food crumbs (corn puffs) in a square open-field (OF) arena of size 1.5 × 1.5 m, with black flooring and enclosed by walls of height 50 cm. A large white cue card was affixed to one of the arena walls (height same as the wall; width 41 cm; horizontal placement at the middle of the wall). At the time of the surgery, each rat was highly familiar with the environment and task (10–20 training sessions lasting at least 20 min each).

## Wagon-wheel track foraging task

The wagon-wheel (WW) track task was designed to function as a 1D version of the 2D OF foraging task. The track's geometry comprised an elevated circular track with two perpendicular cross-linking arms spanning the circle's diameter. The track was 10 cm wide and was bounded on both sides by a 1-cm-high lip. Each section of the track was fitted with a reward point, placed halfway between the two nearest junctions, in the centre of the section. Each reward point consisted of an elevated well that could be remotely filled with chocolate milk via attached tubing. To encourage foraging behaviour, a pseudorandom subset of the wells (between one and four of the eight wells) was filled at a given time, and the rat was allowed to explore the full maze freely and continuously. Wells were refilled as necessary when the rat consumed rewards. Each rat was trained to high performance on the foraging task before the surgery (collecting at least 30 rewards within a 30-minute session). Training to this level of performance took 5–10 half-hour sessions.

## Natural sleep

For sleep sessions, the rat was placed in a black acrylic 'sleep box' with a 40 × 40-cm square base and 80-cm-high walls. The black coating of walls was transparent to infrared, which allowed the 3D motion capture to track the rat through the walls. The bottom of the sleep box was lined with towels, and the rat had free access to water. During recording sessions in the sleep box, the main room light was switched on and pink noise was played through the computer speakers to attenuate disturbing background sounds. Sleep sessions typically lasted 2–3 h, but were aborted prematurely if the rat seemed highly alert and unlikely to sleep.

## Spike sorting and single-unit selection

Spike sorting was performed with KiloSort 2.5[26]. In brief, the algorithm consists of three principal stages: (1) a raw-data alignment procedure that detects and corrects for shifts in the vertical position of the Neuropixels probe shank relative to the surrounding tissue; (2) an iterative template-matching procedure that uses low-rank, variable-amplitude waveform templates to extract and classify single-unit spikes; and (3) a curation procedure which detects appropriate template merging and splitting operations based on spike train auto- and cross-correlograms. Some customizations were made to the standard KiloSort 2.5 method to improve its performance on recordings from the MEC–PaS region, where there is a particularly high spatiotemporal overlap of spike waveforms owing to the high density of cells. Therefore, the maximum number of spikes extracted per batch in step 1 above was increased, as was the number of template-matching iterations in step 2. To improve the separation between cells with very similar-looking waveforms, the upper limit on template similarity was raised from 0.9 to 0.975 in step 2 and to 1.0 on step 3, while supervising manually all merge and split operations from step 3, using a custom-made GUI running in MATLAB. The manual supervision ensured that Kilosort 2.5 did not automatically merge pairs of units with a dip in the cross-correlogram, which in our data was often due to out-of-phase spatial tuning. The merge and split operations were repeated several times to ensure the best separation between single units.

Single units were discarded if more than 1% of their interspike interval distribution consisted of intervals less than 2 ms. In additions, units were excluded if they had a mean spike rate of less than 0.05 Hz or greater than 10 Hz (calculated across the full recording duration).

## Single-unit spike waveforms

During spike sorting, Kilosort assigned each unit with a 2 ms spike waveform template on each recording channel. To calculate a representative single waveform for each unit, the peak-to-peak amplitude of the template was calculated on every channel, and the templates from the three highest-amplitude channels were averaged to generate the representative spike waveform. To calculate spike width, a unit's representative waveform was finely interpolated (from 61 to 1,000 points) using a cubic spline function. Spike width was defined as the time difference between the waveform's negative peak (to which the waveform was aligned by Kilosort), and the following positive peak.

## Spatial position and direction tuning

During awake foraging sessions in the OF arena or wagon-wheel track, only time epochs in which the rat was moving at a speed above 2.5 cm s$^{-1}$ were used for spatial or toroidal analyses. To generate 2D rate maps for the OF arena, position estimates were binned into a square grid of 3 × 3-cm bins. The spike rate in each position bin was calculated as the number of spikes recorded in the bin, divided by the time the rat spent in the bin. To interpolate the values of unvisited bins, two auxiliary matrices were used, $M_1$ and $M_2$, setting visited bins equal to the value of the original rate map in $M_1$ and to 1 in $M_2$, and setting unvisited bins to zero in both. One iteration of the image-processing 'closing' operation was then performed (binary dilation followed by erosion, filling out a subset of the non-visited bins) on $M_2$, using a disk-shaped structuring element, first padding the matrix border by one bin. Both matrices were then spatially smoothed with a Gaussian kernel of smoothing width 2.75 bins. Finally, the rate map was obtained by dividing $M_1$ by $M_2$. Rate-map spatial autocorrelograms and grid scores were calculated as described previously[28]. The selectivity of each cell's position tuning was quantified by computing its spatial information content[42], measured in bits per spike (see 'Information content').

Head-direction tuning curves were calculated by binning the head-direction estimates into 6° bins. The spike rate in each angular bin was calculated as the number of spikes recorded in the bin divided by the time that the rat spent in the bin. The resultant tuning curve was smoothed with a Gaussian kernel with $\sigma = 2$ bins, with the ends of the tuning curve wrapped together. The selectivity of head-direction tuning was quantified using the mean vector length (MVL) of the tuning curve. This was calculated according to:

$$\mathrm{MVL} = \frac{|\sum_{j=1}^{M} \mathbf{f}_j \exp(i\boldsymbol{\alpha}_j)|}{\sum_{j=1}^{M} \mathbf{f}_j},$$

where vector $\mathbf{f}$ represents the tuning curve values (firing rates), vector $\boldsymbol{\alpha}$ represents the corresponding angles, $M$ is the number of tuning curve values, and $|\cdot|$ represents the absolute value of the enclosed term.

## Grid module classification

A novel method was implemented to detect populations of cells corresponding to grid modules by finding clusters of cells that expressed similar spatially periodic activity in the open field (Extended Data Fig. 2). Contrary to previous clustering-based methods for grid modules[3], this approach makes no assumptions about the specific geometry of the grid pattern, thus making it less susceptible to the detrimental effects of geometric distortions such as ellipticity[3,30].

For each MEC–PaS cell in a given recording, a coarse-resolution rate map of the OF session was constructed, using a grid of 10 × 10-cm bins, with no smoothing across bins. The 2D autocorrelogram of this rate map was calculated, and the central peak was removed by excluding all bins located less than 30 cm from the autocorrelogram centre. Bins located more than 100 cm from the autocorrelogram centre were also excluded. The autocorrelograms for all cells were subsequently converted into column vectors, z-standardized, then concatenated to form

a matrix with spatial bins as rows and cells as columns. The nonlinear dimensionality reduction algorithm UMAP[43,44] was then applied to this matrix, yielding a two-dimensional point cloud in which each data point represented the autocorrelogram of one cell (Extended Data Fig. 2a–d; UMAP hyperparameters: 'metric'='manhattan', 'n_neighbors'=5, 'min_dist'=0.05, 'init'='spectral'). In the resultant 2D point cloud, cells with small absolute differences between their autocorrelogram values were located near to one another. The point cloud was partitioned into clusters using the DBSCAN clustering algorithm (MATLAB function 'dbscan', minimum 30 points per cluster, eta = 0.6–1.0). In every recording, the largest cluster was mainly composed of cells that either lacked strong spatial selectivity or were spatially selective but without clear periodicity. All remaining clusters contained cells with high grid scores, and with similar grid spacing and orientation (Extended Data Fig. 2a–d); cluster membership was therefore used as the basis for grid module classification. In one recording (rat 'R' day 1), two clusters were identified that had similar average grid spacing and orientation (labelled as 'R1a' and 'R1b' in Extended Data Fig. 2a–d), suggesting that they represented the same grid module. R1b appeared to comprise cells with higher variability in the within-field firing rates of the spatial rate maps, accompanied by more irregularities in the autocorrelograms. These two clusters were merged together in subsequent analysis (in which the resultant cluster is called 'R1').

A subset of the cells that were assigned to grid module clusters by the above procedure were tuned to both location and head direction (conjunctive grid × direction cells). These cells, which were defined as having a head-direction tuning curve with mean vector length above 0.3, were discarded from further analysis.

## Classification of sleep states

SWS and REM periods were identified on the basis of a combination of behavioural and neural activity, following previously described approaches[6,45,46]. First, sleep periods were defined as periods of sustained immobility (longer than 120 s with a locomotion speed of less than 1 cm s$^{-1}$ and head angular speed of less than 6° s$^{-1}$). Qualifying periods were then subclassified into SWS and REM on the basis of the amplitude of delta- and theta-band rhythmic activity in the recorded MEC–PaS cells. Spike times for each cell were binned at a resolution of 10 ms and the resultant spike counts were binarized, such that '0' indicated the absence of spikes and '1' indicated one or more spikes. The binarized spike counts were then summed across all cells (Extended Data Fig. 9A). The rhythmicity of this aggregated firing rate with respect to delta (1–4 Hz) and theta (5–10 Hz) frequency bands was quantified by applying a zero-phase, fourth-order Butterworth band-pass filter, then calculating the amplitude from the absolute value of the Hilbert transform of the filtered signal, which was smoothed using a Gaussian kernel with $\sigma = 5$ s and then standardized ('z-scored'). The ratio of the amplitudes of theta and delta activity was hence calculated (theta/delta ratio, 'TDR'). Periods in which TDR remained above 5.0 for at least 20 s were classified as REM; periods in which TDR remained below 2.0 for at least 20 s were classified as SWS (Extended Data Fig. 9B).

Spectral analysis was performed on 10-ms-binned multi-unit activity using the multi-tapered Fourier transform, implemented by the Chronux toolbox (http://chronux.org/, function 'mtspectrumsegc'). Non-overlapping 5-second windows were used, with a frequency bandwidth of 0.5 Hz and the maximum number of tapers.

## Visualization of toroidal manifold

For each module of grid cells, spike times of co-recorded cells in the OF were binned for each cell at a resolution of 10 ms, and the binned spike counts were convolved with a Gaussian filter with $\sigma = 50$ ms. Time bins in which the rat's speed was below 2.5 cm s$^{-1}$ were then discarded. To account for variability of average firing rates across cells, the smoothed firing rate of each cell was z-scored. For computational reasons, the time bins were downsampled, taking every 25th time

bin (equating to 250-ms intervals between selected samples). Collectively, the downsampled firing rates of the full population of cells formed a matrix with time bins in rows and cells in columns. PCA was applied to this matrix (treating time bins as observations and cells as variables), and the first six principal components were retained (Extended Data Figs. 3Aa–c, 4a–d). UMAP[43,47] was then run on these six principal components (with time bins as observations and principal components as variables). The hyperparameters for UMAP were: 'n_dims'=3, 'metric'='cosine', 'n_neighbours'=5000, 'min_dist'=0.8 and 'init'='spectral'.

For visualizing the toroidal manifold during WW, smoothed firing rates were first calculated by the same procedure described above for OF. Subsequently, to allow comparison of the toroidal manifold between OF and WW, the same PCA and UMAP transformations calculated for the OF data were re-applied to the WW data, by supplying the fitted OF UMAP transformation as the argument 'template_file' to the 'run_umap' function in the MATLAB implementation[47].

## Preprocessing of population activity

Each topological analysis was based on the activity of a single module of grid cells, during a single experimental condition in one recording session. Topological analysis of multi-module and conjunctive grid × direction cell activity was not considered as we expect such data to exhibit higher-dimensional topological structure requiring a higher number of cells[27]. The experimental conditions were: open-field foraging (OF), wagon-wheel track foraging (WW), slow-wave sleep (SWS), and rapid eye-movement sleep (REM). Sleep epochs of the same type were collected from across the recording and concatenated for analysis purposes. Similarly, in one case (rat 'S'), two WW task sessions were concatenated to increase the sample size.

In total there were 27 combinations of module (Q1, Q2, R1, R2, R3, S1) and experimental condition (OF day 1, OF day 2, WW, REM, SWS).

Preprocessing of spike trains began by computing delta functions centred on the spike times (valued 1 at time of firing; 0 otherwise), and convolving these temporally with a Gaussian kernel with $\sigma = 50$ ms (OF, WW and REM) or 25 ms (SWS). Samples of the smoothed firing rates of all cells ('population activity vectors') were then computed at 50-ms intervals. The awake states were further refined by excluding vectors which originated from time periods when the rat's speed was below 2.5 cm s$^{-1}$.

Computing the persistent cohomology of a point cloud is computationally expensive and may be sensitive to outliers (for example, spurious points breaking the topology of the majority of points in the point cloud). For this reason, it is common to preprocess the data by downsampling and dimension-reducing the point cloud. The same preprocessing procedure was used for all datasets in the present study.

First, the data points were downsampled by keeping the 15,000 most active population activity vectors (as measured by the mean population firing rate). During SWS, this selection criterion had the consequence of automatically discarding population activity vectors during down-states, when neural activity is near-silent. As noise is inherently more prevalent and cosine distances less reliable in high-dimensional spaces ("the curse of dimensionality")[48], dimensionality-reduction and a normalization of distances were subsequently performed. The reduced point cloud was z-scored and projected to its six first principal components, thus reducing noise while keeping much of the variance (see Extended Data Fig. 4a). This was supported by the lack of grid structure and the clear drop in explained deviance after six components (see Extended Data Fig. 4b, c). The explained deviance was computed by fitting a GLM model to each component individually, using the spatial coordinates as covariate, suggesting that the higher components are less spatially modulated and possibly better described by other (unknown) covariates. Consistent with this, the toroidal structure was most clearly detected in the barcodes when comparing the ratio of the lifetimes of the two most persistent H$^1$ bars

versus the third longest-lived H$^1$ bar for the barcodes obtained when using different numbers of components in the analysis (see Extended Data Fig. 4d). These analyses both indicated that dimensionality reduction was required to firmly demonstrate the toroidal topology in the grid cells. The empirical findings are supported theoretically; see 'Theoretical explanation of the six-dimensionality proposed by PCA' in Supplementary Methods.

To further simplify the low-dimensional point cloud, a different downsampling technique was introduced, based on a point-cloud density strategy motivated by a topological denoising technique introduced previously[49] and a fuzzy topological representation used in UMAP[43,50]. Parts of the open-source implementation of the latter were copied in this computation. This approach consisted of assigning, for each point, a neighbourhood strength to its $k$ nearest neighbours, and subsequently sampling points that represent the most tight-knit neighbourhoods of the point cloud in an iterative manner. First, we defined $m'_{i,i_j} = \exp\left(-\frac{d_{i,i_j}}{\sigma_i}\right)$, where $d_{i,i_j}$ is the cosine distance between point $x_i$ and its $j$-th nearest neighbour and $\sigma_i$ is chosen to make $\sum_{j=1}^{k} m'_{i,i_j} = \log_2 k$, using $k = 1,500$. The neighbourhood strength was then obtained by symmetrizing: $m_{i,i_j} = m'_{i,i_j} + m'_{j,j_i} - m'_{i,i_j} \cdot m'_{j,j_i}$. Finally, the point cloud was reduced to 1,200 points by iteratively drawing the $i$-th point as: $\max_{x_i} \sum_{j \in \tilde{I}} m_{j,j_i}$, where $\tilde{I}$ denotes the indices of the points not already sampled. In other words, for each iteration, the sampled point is the one with the strongest average membership of the neighbourhoods of the remaining points.

To compute the persistent cohomology of the downsampled point cloud, the neighbourhood strengths were first computed for the reduced point cloud (using $k = 800$) and its negative logarithm was taken, obtaining a distance matrix. This matrix was then given as input to the Ripser implementation[51,52] of persistent cohomology, returning a barcode. In short, the barcode gave an estimate of the topology of the fuzzy topological representation of the six principal components of the grid-cell population activity. Thus, in essence, the first step of UMAP was applied before describing the resulting representation with persistent cohomology, instead of using it to project each point of the point cloud to a representation of user-specified dimensionality for visualization (Extended Data Fig. 3Ad, e). This gives a more direct and stable quantification of the global data structure, without having to choose an initialization[53] or optimize a lower-dimensional representation.

## Persistent cohomology

Persistent cohomology, a tool in topological data analysis, was used to characterize the manifold assumed to underlie the data. This has clear ties with persistent homology and the main result (the barcode) is identical, thus the two terms are often used interchangeably. Persistent cohomology was chosen because the computation is (to our knowledge) faster and is required to obtain cocycle representatives, which are necessary to perform decoding (see 'Cohomological decoding'). Persistent (co-)homology has previously been successful in analysing neural data, describing the ring topology of head direction cell activity[22–24], the spherical representation of population activity in primary visual cortex[54], and the activity of place cells[55–58].

The general outline of the algorithm is as follows. Each point in the cloud is replaced by a ball of infinitesimal radius, and the balls are gradually expanded in unison. Taking the union of balls at a given radius results in a space with holes of different dimensions. The range of radii for which each hole is detected is tracked; this is referred to as the 'lifetime' of the hole and is represented by the length of a bar. The totality of bars is referred to as the barcode.

The software package Ripser[51,52] was used for all computations of persistent cohomology. Ripser computes the persistent cohomology of 'Vietoris-Rips complexes' (which approximate the union of balls for different radii), constructed based on the input distance matrix and a

choice of coefficients (in our case, $\mathbb{Z}_{47}$-coefficients), and outputs the barcode and cocycle representatives for all bars. The prime 47 was chosen as homology and cohomology coincide in this case and as it is unlikely that this divides the torsion subgroup of the homology of the space. Torsion may indicate, for example, orientability of a manifold and in choosing 47 as our prime, we disregard all but 47-torsion. Testing with other primes (for example, 43) gave similar results (data not shown) and the Betti numbers stayed the same regardless of choice of prime.

To verify that the lifetimes of prominent bars in the barcodes were beyond chance, shuffled distributions were generated for the persistence lifetimes in each dimension. In each shuffling, the spike train of each cell was shifted independently in time by rolling the firing rate arrays a random length between 0 and the length of the session. The same preprocessing and persistence analysis were then performed on the shifted spike trains as for the unshuffled data. This was performed 1,000 times, and each time a barcode was obtained. The barcodes were concatenated for all shuffles and the maximum lifetime was found for each dimension. This lifetime served as a significance criterion for the bar lifetimes. It is noted, however, that this is a heuristic and that statistics of barcodes are still not well established.

## Cohomological decoding

As there are other spaces with similar barcodes as for a torus, the results identified by the barcode were further investigated, using the 'cohomological decoding' procedure introduced previously[59] to calculate a toroidal parametrization of the point clouds of population activity. This assigns to each point corresponding positions on each of the two circular features identified by the 1D bars with the longest lifetime, resulting in coordinates that further characterize the underlying shape of the data.

Cohomological decoding is motivated by the observation that the 1D cohomology (with integer coefficients) of a topological space $X$ is equivalent to the set of homotopy-equivalent classes of continuous maps from $X$ to the circle ($S^1$)[60]; that is:

$$H^1(X;\mathbb{Z}) \cong [X, S^1].$$

This subsequently means that for each 1D bar existing at a given radius, there exists a corresponding continuous map from the Vietoris-Rips complex of that radius to the circle. Thus, we may first use persistent cohomology to detect which elements represent meaningful (long-lived) features of the data and choose a radius for which these features exist. As the vertices of the Vietoris-Rips complex are points in the point cloud, the circular values of the corresponding maps at the vertices describe circular coordinates of the data.

In the present case, persistent cohomology was first applied to the grid-cell population activity and $X$ was identified as the Vietoris-Rips complex for which the two longest-lived one-dimensional bars in the barcode (representing each of the two circles of the torus) existed. To define the desired toroidal coordinates on a domain that was as large as possible, we chose the complex given at the scale of the birth plus 0.99 times the lifetime of the second longest-lived one-dimensional bar in the barcode[22,59,61]. Next, the cocycle representatives (given by the persistent cohomology implementation of Ripser[51,52]) of each of the chosen 1D bars defined $\mathbb{Z}_{47}$-values for each of the edges in the complex. These edge values were then lifted to integer coefficients and subsequently smoothed by minimizing the sum over all edges (using the scipy implementation 'lsmr'). The values on the vertices (points) of each edge followed from the edge values and gave the circular parametrizations of the point cloud. The product of the two parametrizations thus provided a mapping from the neural activity to the two-dimensional torus—that is, giving a toroidal coordinatization (decoding) of the data.

As persistent cohomology was computed for a reduced dataset of 1,200 points and therefore circular parametrizations were obtained only for this point cloud, each parametrization was interpolated to the population activity from the rest of the session(s). First, the 1,200

toroidal coordinates were weighted by the normalized ('z-scored') firing rates of the cells at those time points, obtaining a distribution of the coordinates for each grid cell. The decoded toroidal coordinates were then computed by finding the mass centre of the summed distributions, weighted by the population activity vector to be decoded. These activity vectors were calculated by first applying a Gaussian smoothing kernel of 15-ms standard deviation to delta functions centred on spike times, sampling at 10-ms intervals and then z-scoring the activity of each cell independently. Time intervals that contained no spikes from any cell were subsequently excluded. When decoding was used to assess or compare the tuning properties of single cells (for example, comparison of toroidal versus spatial description), the coordinates were computed using the weighted sum of the distributions of the other cells; that is, the contribution of the cell to be assessed or compared was removed. When comparing preservation of toroidal tuning across two sessions, coordinates were interpolated either using the toroidal parametrization in each session independently ('Separate') or using the same toroidal parametrization in both sessions ('Common').

## Toroidal rate map visualization

For visualization, toroidal firing rate maps were calculated in the same way as the physical space covariate (see 'Spatial position and direction tuning'), first binning the toroidal surface into a square grid of 7.2° × 7.2° bins and computing the average spike rate in each position bin. However, for toroidal maps, it was necessary to address the 60° angle between the toroidal axes before smoothing. After binning the toroidal coordinates, the rate map was 'straightened' by shifting the bins along the x axis ('horizontally') the length of ($y$ mod 2)/2 bins, where $y$ is the vertical enumeration of the given bin. Copies of the rate map were then tiled in a three-by-three square (similar to Extended Data Fig. 5d), before applying the closing and smoothing operations as for the spatial firing rate map. The single toroidal rate map was finally recovered by cutting out the centre tile, rotating it 90° and defining 15° shear angles along both the x and the y axis to correct for the 60° offset between them.

## Comparison of spatial periodicity

Differences in grid periodicity between OF and WW environments were quantified for a given cell by comparing the grid scores in the two behavioural conditions. Two alternative methods were used to generate the spatial autocorrelograms for this comparison: (1) comparing the autocorrelograms for OF and WW directly; and (2) comparing autocorrelograms for OF and WW after first equalizing the spatial coverage between the two conditions.

For method (1), rate maps were calculated as specified in the above section 'Spatial position and direction tuning', using the same grid of 3 × 3-cm bins for both environments. This set of bins spanned the entirety of the OF arena and covered most of the WW track apart from some small regions at the outer extrema, which were discarded for the purpose of this analysis. For each of the two rate maps, the autocorrelogram was computed and the grid score was calculated.

Method (2) was similar to method (1), except that the cell's OF rate map was converted into a 'masked OF' rate map, by removing all bins that were unvisited by the rat in the WW session. This effectively equalized the position coverage between the two conditions, and thus allowed for a more valid comparison.

## Toroidal versus spatial description

The explanatory significance of the toroidal description was evaluated by comparing statistical measures of how well the toroidal coordinates explained neural activity on the torus and in physical space. For a fair comparison, it was important to avoid overfitting, which might occur if a toroidal parametrization of a point cloud is used to describe that same set of data points. Two precautions were taken to avoid such overfitting: first, the data were decoded using the toroidal parametrization

from a different condition (an OF session for a WW recording and a WW session for an OF recording), and second, the cell for which the statistical measurement was made was omitted from the decoding.

The comparison of toroidal and environmental representations also accounted for tracking error in the physical position estimate, which mainly resulted from the approximately 4 cm vertical offset of the tracking device above the rat's head. This causes a discrepancy when the angle $\alpha$ between the animal's zenith and the axis of gravitation is different from 0°, measured as $4\tan(\alpha)$ cm. The mean discrepancy in the recorded position data was measured to 1.5 cm. To account for this error of the position estimate, proportional Gaussian noise was added to the toroidal coordinates, using a standard deviation of 1.5 cm/$\Omega$, where $\Omega$ denotes the grid spacing of the particular grid-cell module, estimated from the mean period of the fitted cosine waves of the toroidal coordinates in the open field (see 'Toroidal alignment').

## Information content

The information content ($I$) was calculated as previously described[42], to quantify and compare the amount of information carried by single-cell activity about the location on the torus and physical space per spike. Both covariates were binned in a $M = 15 \times 15$ grid of square bins. For each bin $j$, the average firing rate $\mathbf{f}_j$ (given in spikes per second), and the occupancy ratio, $\mathbf{p}_j$, were computed. The information content for each grid cell was then given as:

$$I = \frac{1}{\bar{\mathbf{f}}} \sum_{j=1}^{M} \mathbf{f}_j \log_2 \frac{\mathbf{f}_j}{\bar{\mathbf{f}}} \mathbf{p}_j,$$

where $\bar{\mathbf{f}}$ is the mean firing rate of the cell across the entire session.

Note that although the rate maps for physical space have multiple firing fields, whereas the toroidal rate maps have single firing fields, we expect the spatial information to be comparable, as the measure primarily depends on the ratio of bins with high firing activity. This number should be comparable as the firing field size (in bins) will be inversely related to the number of fields in the rate map, assuming that the discretization of the map captures the relevant firing rate variations. For example, given a similar binning of space, a larger OF environment will include more fields, but the number of bins per field will decrease correspondingly. The binning used should be sufficient to resolve the smallest fields, as the same discretization was used in classifying the grid cells in the recorded population.

## Deviance explained

Deviance explained was computed to measure how well a Poisson GLM model fitted to the spike count was at representing the data, using either the toroidal coordinates or the tracked position as regressors. A similar set-up was used to that of a previous study[62], with a smoothness prior for the GLM to avoid overfitting.

Both the toroidal and spatial coordinates were binned into a $15 \times 15$ grid of bins, and GLM design matrices were built with entries $X_i(t) = 1$ if the covariate at time $t$ fell in the $i$-th bin and $X_i(t) = 0$ otherwise.

The Poisson probability of recording $k$ spikes in time bin $t$ is:

$$P(k|\mu(t), \beta) = \exp(-\mu(t)) \frac{\mu(t)^k}{k!},$$

where $\mu(t) = \exp\left(\sum_i \beta_i X_i(t)\right)$ is the expected firing rate in time bin $t$. The parameters $\beta$ of the Poisson GLM were optimized for each covariate by minimizing the cost function:

$$L(\beta|\mu(t), \gamma, k) = -\sum_t \ln(P(k(t)|\mu(t), \beta)) + \frac{1}{2}\gamma \sum_{i,j \in N} (\beta_i - \beta_j)^2,$$

where $N$ is the set of neighbour pairs. The first term is the negative log-likelihood of the spike count in the given time bin, whereas the second term puts a penalty on large differences in neighbouring parameters, enforcing smoothness in the covariate response of the predicted spike count.

The parameters, $\beta$, were initialized to zero and then modified to minimize the loss function by first running two iterations of gradient descent, before optimizing using the 'l-bfgs-b'-algorithm (as implemented in the 'scipy.optimize'-module) with 'gtol'=1e-5 as the cut-off threshold, and finally running two more iterations of gradient descent. A three-fold cross validation procedure was used, repeatedly fitting the model to two-thirds of the data and testing on the held-out last third.

The smoothness hyperparameter $\gamma$ was optimized a priori on each grid-cell module based on the summed likelihood, testing $\gamma \in (1, \sqrt{10}, 10, \sqrt{1{,}000})$, and found to be either 1 or $\sqrt{10}$ in all cases.

Similarly, after fitting a null model (using only the intercept term) and the saturated model (perfectly fitting each spike count), the deviance explained could be computed as:

$$1 - \frac{\text{ll}_s - \text{ll}_p}{\text{ll}_s - \text{ll}_0},$$

where $\text{ll}_p$, $\text{ll}_0$ and $\text{ll}_s$ denote the cross-validated log likelihood of the fitted model, the null model and the saturated model, respectively. This provides a normalized comparison describing the difference between the fitted model and the idealized model.

## Toroidal alignment

To infer a geometric interpretation of the tori, as characterized via the cohomological decoding, and compare the toroidal parametrizations across modules and conditions, two cosine waves of the form $\cos(\omega t + k)$ were fitted to the OF mappings of the decoded circular coordinates (Extended Data Fig. 5a), where $t$ is the centre $100^2$-bins of a $540° \times 540°$-valued $150^2$-bin grid rotated $\theta$ degrees. The parameters $(\omega, k, \theta)$ were optimized by minimizing the square difference between the cosine waves and the cosine of the mean of the circular coordinates in $100^2$ bins of the physical environment (smoothed using a Gaussian kernel with 1-bin standard deviation). Estimates were first obtained by finding the minimum when testing all combinations in the following intervals, each discretized in 10 steps: $\omega \in [1,6]$, $\phi \in [0, 360)$ and $\theta \in [0,180)$. The parameters of the cosine waves were further optimized using the 'slsqp'-minimization algorithm (as implemented in the 'scipy.optimize'-module using default hyperparameters). The period of each cosine wave was computed as $1.5\text{ m}/\omega$, giving a spatial scale estimate of the grid-cell modules.

As circular coordinates have arbitrary origin and orientation (that is, clockwise or counterclockwise evolution) we needed to realign the directions of the circular coordinates to compare these across modules and sessions (see Extended Data Fig. 4b). The clockwise orientation of each circular coordinate was first determined by noting whether $(\omega t + k)$ or $360° - (\omega t + k)$ best fit the spatial mapping of the circular means of the toroidal coordinates, and subsequently reoriented to obtain the same orientation for both coordinates. The coordinate for which $\cos(\theta)$ was largest (intuitively, the '$x$ axis') was then defined as the first coordinate (denoted $\phi_1$, with parameters $(\omega_1, k_1, \theta_1)$) and the other as the second coordinate ($\phi_2$). Although $(\phi_1, \phi_2)$ fully describe the toroidal location, the hexagonal torus allows for three axes, and the two axes obtained are thus oriented at either 60° or 120° relative to each other (see Extended Data Fig. 5b). The difference in directions was given by $\theta_1 - \theta_2$ and if this difference was greater than 90°, $\phi_2$ was replaced with $\phi_2 + 60° \cdot \phi_1$. Finally, the origin of the coordinates was aligned to a fixed reference, by subtracting the mean angular difference between the decoded coordinates and the corresponding coordinates obtained when using the toroidal parametrization of the reference OF session.

For visualization (Extended Data Fig. 5), it was furthermore necessary, in some cases, to rotate both vectors of the rhombi 30 degrees depending on whether one of the axes was directed outside of the box.

## Preservation of toroidal tuning

Centre-to-centre distance and Pearson correlation were computed between toroidal tuning maps of different sessions to measure the degree of preservation between the toroidal descriptions.

First, the preferred toroidal firing location for each cell was computed as the centre of mass of the toroidal firing distribution:

$$T_c = \arctan 2\left( \frac{\sum_i \sin \theta_i \cdot \mathbf{y}_i}{\sum_i \mathbf{y}_i}, \frac{\sum_i \cos \theta_i \cdot \mathbf{y}_i}{\sum_i \mathbf{y}_i} \right),$$

where $\mathbf{y}_i$ denotes the mean spike count of the given cell in the $i$-th bin whose binned toroidal coordinates are given by $\theta_i$. The distance between mass centres found in two sessions ('$S_1$' and '$S_2$') was then defined as:

$$d = \|\arctan 2(\sin(T_c^{S_2} - T_c^{S_1}), \cos(T_c^{S_2} - T_c^{S_1}))_2\|$$

where $\| \cdot \|_2$ refers to the $L_2$-norm.

Pearson correlation between two tuning maps was computed by flattening the smoothed 2D rate maps to 1D arrays and calculating the correlation coefficient, $r$, using the 'pearsonr'-function given in the 'scipy.stats'-library.

To determine how much the preservation of the toroidal representations across two sessions (measured with Pearson correlation and peak distance) differed from a random distribution, the indices of the cells in one of the sessions were randomly re-ordered before computing correlation and distance for the pair of conditions. This process was repeated 1,000 times, and the $P$ value was calculated from the rank of the original $r$ value or distance with respect to the shuffled distribution.

## Classification of grid cells

Temporal autocorrelograms were computed, for each cell, by calculating a histogram of the temporal lags between every spike and all surrounding spikes within a 200 ms window, using 1 ms bins. The histogram was then divided by the value of the zero-lag bin, which was subsequently set to zero. The autocorrelogram was smoothed using a gaussian kernel with smoothing window 4 ms. Considering the autocorrelograms of all modules during OF foraging (day 2 for R1–3) as a point cloud, the cosine distances between all points were calculated, and hence each point's 80 nearest neighbours were found. This defined a graph in which each point described a vertex and the neighbour pairs gave rise to edges. A density estimate was then calculated as the exponential of the negative distances summed over each neighbour for each point. The graph and the density estimate were given as the input to the Gudhi implementation[63] of ToMATo[64]. ToMATo uses a hill-climbing procedure to find modes of the density function and uses persistence to determine stable clusters. In the present case, the algorithm finds three long-lived clusters.

## Minimum number of cells for torus detection

To address the question of how many cells are minimally needed to expect to see toroidal structure, random samples of $n = 10, 20, ..., 140$ cells were taken from R2 ($n = 149$ cells) during OF foraging, and the same topological analysis was repeated as for the whole population. The cells were resampled 1,000 times for each number of cells in the subsample. To determine whether toroidal structure was detected, a heuristic was introduced based on the circular parameterization given by the two most persistent 1D bars in the barcode mapped onto physical space. An estimate of the resulting planar representation of the torus was obtained by fitting planar cosine waves to each mapping (see 'Toroidal alignment'). For the analysis to be determined 'successful' in detecting toroidal structure, we required: (i) the mean value of the least-squares fitting (across bins of the mapping) to be less than 0.25; (ii) the angle of the rhombus to be close to 60° (between 50° and 70°); and (iii) the side lengths to be within 25% of each other.

## Toroidal peak detection

The number of peaks per toroidal rate map was detected to assert the number of grid cells whose toroidal rate map portrayed single fields. First, 1,000 points were sampled from the toroidal distribution given by the mean activity of each cell in 150 × 150 bins of the stacked toroidal surface (that is, as described in 'Toroidal rate map visualization', each 50 × 50-binned toroidal rate map is first 'straightened' and subsequently stacked in 3 × 3 to address the toroidal boundaries) and then spatially smoothed using a Gaussian kernel with smoothing widths 0, 1, 2, ..., 10 bins with mode set to 'constant' in the 'scipy.gaussian_filter' function. Next, the points were clustered by computing a density estimate, using the Euclidean distance, and defining neighbours as points closer than 5 bins. Cluster labels were iteratively assigned to each point and all its neighbours in a downhill manner, instantiating a new cluster identity if the point was not already labelled. Finally, the centroids for each cluster were computed and counted as a peak depending on whether its position fell within the centre 50 × 50 bins of the stacked rate maps.

## Simulated CAN models

To confirm the expected outcomes of topological analyses of grid cell CAN models, grid cells were simulated using two different, noiseless CAN models (Extended Data Fig. 7).

First, a 56 × 44 grid cell network was simulated based on the CAN model proposed previously[9], but using solely lateral inhibition (for details see ref. [11]) in the connectivity matrix, $W$. The animal movement was given as the first 1,000 s of the recorded trajectory of rat 'R' during OF session, originally sampled at 10 ms, and interpolated to 2-ms time steps. The speed, $v(t)$, and head direction $\theta(t)$ of the animal was calculated as the (unsmoothed) displacement in position for every time step. The activity, $\mathbf{s}$, was updated as:

$$\mathbf{s}_{i+1} = \mathbf{s}_i + \frac{1}{\tau}(-\mathbf{s}_i + (I + \mathbf{s}_i \cdot W + \alpha v(t)\cos(\theta(t) - \bar{\theta}))_+),$$

where $(...)_+$ is the Heaviside function and $\bar{\theta}$ is the population vector of preferred head directions. The following parameters were used: $I = 1$, $\alpha = 0.15$, $l = 2$, $W_0 = -0.01$, $R = 20$ and $\tau = 10$, and let the activity pattern stabilize by first initializing to random and performing 2,000 updates, disregarding animal movement. For computational reasons, the activity was set to 0 if $\mathbf{s}_i < 0.0001$. The simulation was subsequently downsampled keeping only every 5th time frame.

Next, a 20 × 20 grid-cell network was simulated, for a synthetically generated OF trajectory ('random walk'), based on the twisted torus model formulated in a previous study[10]. The parameter values and the code for computing both the grid cell network (choosing a single grid scale by defining the parameter 'grid_gain' = 0.04) and the random navigation (using 5,000 time steps) were given by the implementation by Santos Pata[65].

## Idealized torus models

To compare the results of both the original and simulated grid cell networks with point clouds where the topology is known, a priori, to be toroidal, points were sampled from a square and a hexagonal torus. First, a 50 × 50 (angle) mesh grid $(\theta_1, \theta_2)$ was created in the square $[0, 2\pi] \times [0, 2\pi]$ and slight Gaussian noise ($\epsilon = 0.1 \cdot N(0,1)$) was added to each angle. The square torus was then constructed via the 4D Clifford torus parametrization: $(\cos(\theta_1), \sin(\theta_1), \cos(\theta_2), \sin(\theta_2))$. The hexagonal torus was constructed using the 6D embedding: $(\cos(\theta_1), \sin(\theta_1), \cos(a_1\theta_1 + \theta_2), \sin(a_1\theta_1 + \theta_2), \cos(a_2\theta_1 + \theta_2), \sin(a_2\theta_1 + \theta_2))$, where $a_1 = 1/\sqrt{3}$ and $a_2 = -1/\sqrt{3}$.

## Histology and recording locations

Rats were given an overdose of sodium pentobarbital and were perfused intracardially with saline followed by 4% formaldehyde.

The extracted brains were stored in formaldehyde and a cryostat was used to cut 30-μm sagittal sections, which were then Nissl-stained with cresyl violet. The probe shank traces were identified in photomicrographs, and a map of the probe shank was aligned to the histology by using two reference points that had known locations in both reference frames: (1) the tip of the probe shank; and (2) the intersection of the shank with the brain surface. In all cases, the shank traces were near-parallel to the cutting plane, therefore it was deemed sufficient to perform a flat 2D alignment in a single section where most of the shank trace was visible. The aligned shank map was then used to calculate the anatomical locations of individual electrodes (Extended Data Fig. 1).

### Data analysis and statistics
Data analyses were performed with custom-written scripts in Python and MATLAB. Open-source Python packages used were: umap (version 0.3.10), ripser (0.4.1), numba (0.48.0), scipy (1.4.1), numpy (1.18.1), scikit-learn (0.22.1), matplotlib (3.1.3), h5py (2.10.0) and gudhi (3.4.1.post1). Samples included all available cells that matched the classification criteria for the relevant cell type. Power analysis was not used to determine sample sizes. The study did not involve any experimental subject groups; therefore, random allocation and experimenter blinding did not apply and were not performed. All statistical tests were one-sided.

The most intensive computations were performed on resources provided by the NTNU IDUN/EPIC computing cluster[66].

### Additional discussion
The demonstration that populations of grid cells operate on a toroidal manifold, which is preserved across environments and behavioural states, confirms a central prediction of CAN models. The present observations provide the first—to our knowledge—population-level visualization of a two-dimensional CAN manifold, though there is accumulating evidence for one-dimensional CANs in a number of neural systems. The most powerful support for the latter has been obtained in fruit flies, in which CAN-like dynamics can be visualized in a ring of serially connected orientation-tuned cells of the central complex[67–69]. In mammals, analysis of data from dozens of simultaneously recorded head direction cells has shown that population activity in these cells faithfully traverses a conceptual ring[22–24], in accordance with ring-attractor models[17–19]. Dynamics along low-dimensional manifolds with line, ring, or sheet topologies is also thought to underlie a wide range of other mammalian brain functions that operate on continuous scales, spanning from visual orientation tuning[14] to neural operations underlying place-cell formation[70–72], as well as motor control[73], decision making and action selection[74–76], and certain forms of memory[39,77–80]. The present analyses provide a visualization of 2D CAN dynamics in pure grid cells within a module and, together with the previous work, point to a widespread implementation of CAN dynamics in the brain. The existence of CAN structure to constrain activity to low-dimensional manifolds does not preclude additional mechanisms for pattern formation, however. Grid cell patterns may emerge also by feedforward mechanisms[12,38,81–86]. Such mechanisms may operate in parallel with recurrent networks[87] and may even be the primary mechanism for grid-like firing at early stages of development, before the full maturation of recurrent connectivity[11,88–90].

### Reporting summary
Further information on research design is available in the Nature Research Reporting Summary linked to this paper.

### Data availability
The datasets generated during the current study are available at https://figshare.com/articles/dataset/Toroidal_topology_of_population_activity_in_grid_cells/16764508. Source data are provided with this paper.

### Code availability
Code for reproducing the analyses in this article is available at https://figshare.com/articles/dataset/Toroidal_topology_of_population_activity_in_grid_cells/16764508.

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

**Acknowledgements** We thank M. P. Witter for help with evaluation of recording locations, and A. M. Amundsgård, K. Haugen, K. Jenssen, E. Kråkvik, I. Ulsaker-Janke and H. Waade for technical assistance. The work was supported by a Synergy Grant to E.I.M. and Y.B. from the European Research Council ('KILONEURONS', grant agreement no. 951319); an RCN FRIPRO grant to E.I.M. (grant no. 286225); a Centre of Excellence grant to M.-B.M. and E.I.M. and a National Infrastructure grant to E.I.M. and M.-B.M. from the Research Council of Norway (Centre of Neural Computation, grant number 223262; NORBRAIN, grant number 295721); the Kavli Foundation (M.-B.M. and E.I.M.); the Department of Mathematical Sciences at the Norwegian University of Science and Technology (B.A.D., E.H. and N.A.B.); a direct contribution to M.-B.M. and E.I.M. from the Ministry of Education and Research of Norway; and grants to Y.B. from the Israel Science Foundation (grant no. 1745/18) and the Gatsby Charitable Foundation. Some of the computations were performed on resources provided by the NTNU IDUN/EPIC computing cluster.

**Author contributions** R.J.G., M.-B.M. and E.I.M. designed experiments. R.J.G. performed experiments. N.A.B., E.H., B.A.D., R.J.G., Y.B. and E.I.M. conceptualized and proposed analyses. E.H. and R.J.G. developed and performed the analyses. M.P. shared unpublished Kilosort software. R.J.G., E.H., B.A.D., Y.B., M.-B.M. and E.I.M. interpreted data. E.H. and R.J.G. visualized data. R.J.G., E.H., B.A.D., Y.B. and E.I.M. wrote the paper, with periodic input from all authors. E.I.M., M.-B.M., B.A.D. and N.A.B. supervised the project. E.I.M., M.-B.M. and Y.B. obtained funding.

**Competing interests** The authors declare no competing interests.

**Additional information**
**Correspondence and requests for materials** should be addressed to Richard J. Gardner, Nils A. Baas, Benjamin A. Dunn or Edvard I. Moser.

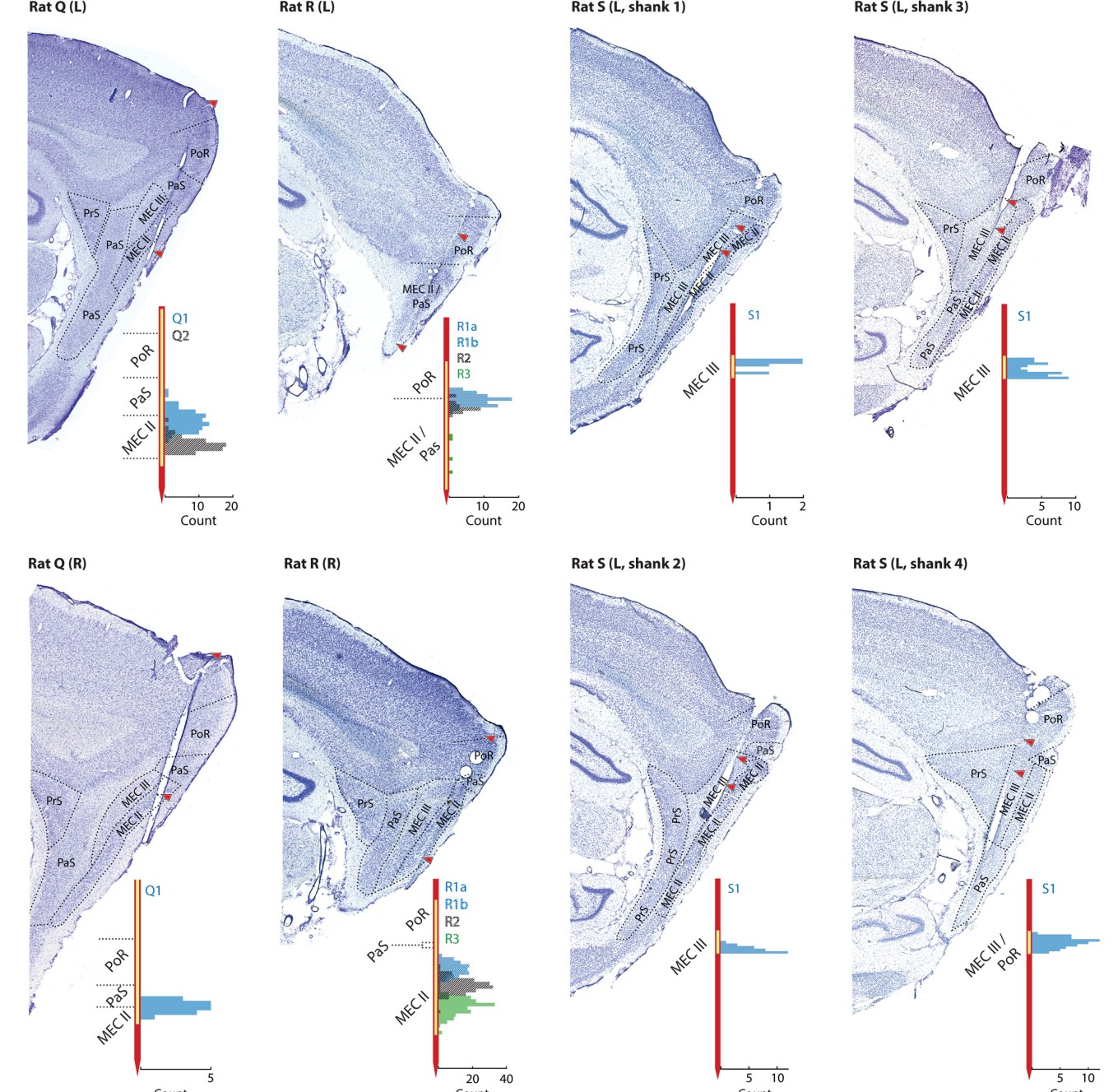

**Extended Data Fig. 1 | Nissl-stained sagittal brain sections showing recording locations for rats Q, R and S.** Red arrows indicate the dorsoventral range of the probe's active recording sites (corresponding to the yellow stripe in the inset). Stippled lines indicate borders between brain regions (MEC, medial entorhinal cortex; PaS, parasubiculum; PrS, presubiculum; PoR, postrhinal cortex). Layers are indicated for MEC (MECII, MECIII). Animal name, hemisphere (L, left; R, right) and shank number (for Rat 'S') are indicated in text above each section. Insets show, for each section, the number of grid cells recorded at each depth on the probe shank (histogram bin sizes 100 μm for Rats 'Q' and 'R', 75 μm for Rat 'S'; total numbers of cells are given in Extended Data Fig. 2g). Only the implanted portion of the probe shank is shown. Counts are colour-coded according to module identity. Module R1 is subdivided into the two UMAP clusters R1a and R1b (as shown in Extended Data Fig. 2), shown here as two stacked histograms. The yellow stripe on the probe shank indicates the range of active recording sites. The indicated locations of units are subject to measurement error, because the anatomical registration of probe shanks can only be approximately estimated, and furthermore because units may be detected on electrodes up to 50 μm away[91]. Note that several modules spanned across hemispheres (see Extended Data Fig. 2g). The cell counts shown for Rat 'R' are from Recording Day 1. The same set of recording sites was used for both recording sessions, and therefore the anatomical distributions of recorded cells were similar between the two sessions.

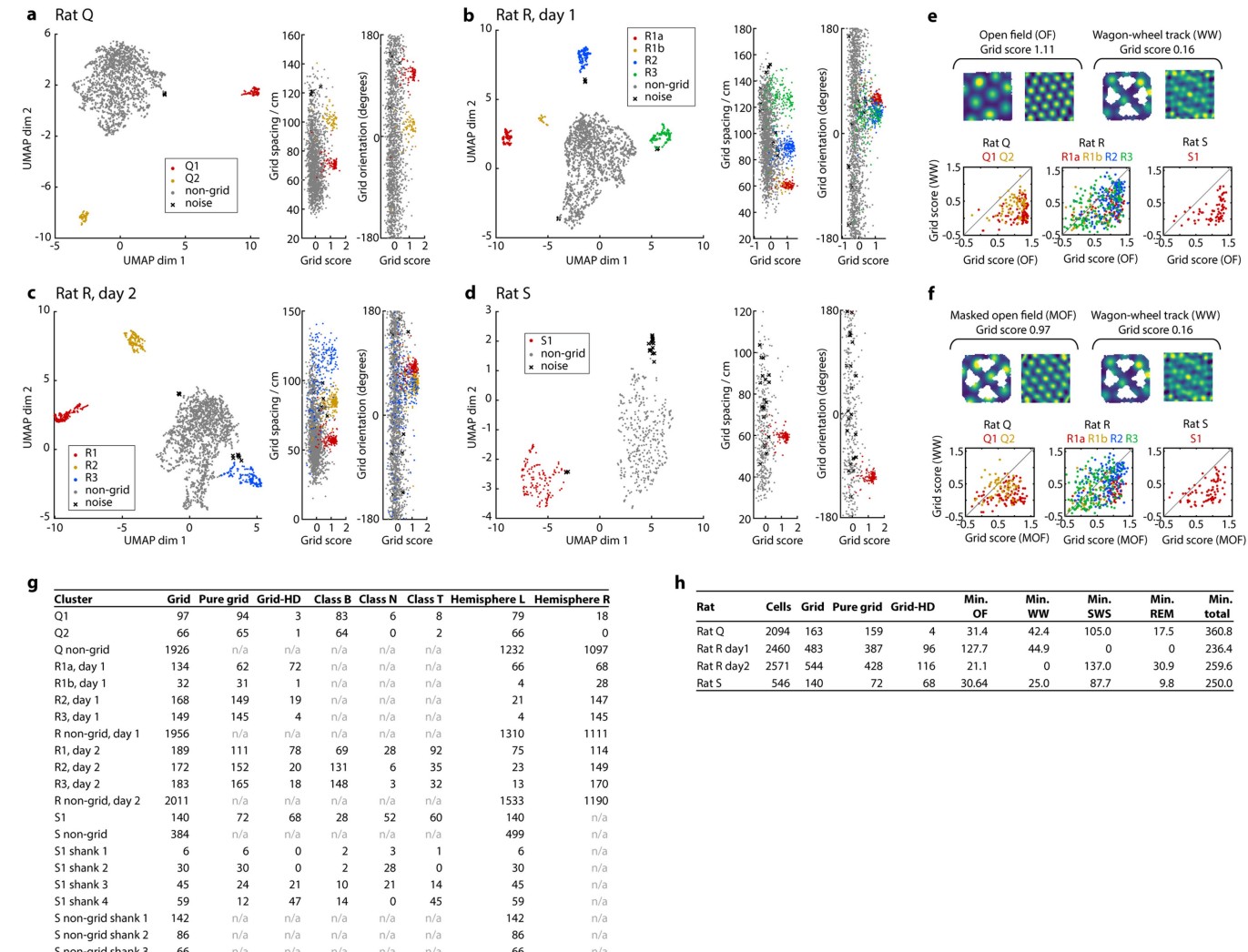

**g**

| Cluster | Grid | Pure grid | Grid-HD | Class B | Class N | Class T | Hemisphere L | Hemisphere R |
|---|---|---|---|---|---|---|---|---|
| Q1 | 97 | 94 | 3 | 83 | 6 | 8 | 79 | 18 |
| Q2 | 66 | 65 | 1 | 64 | 0 | 2 | 66 | 0 |
| Q non-grid | 1926 | n/a | n/a | n/a | n/a | n/a | 1232 | 1097 |
| R1a, day 1 | 134 | 62 | 72 | n/a | n/a | n/a | 66 | 68 |
| R1b, day 1 | 32 | 31 | 1 | n/a | n/a | n/a | 4 | 28 |
| R2, day 1 | 168 | 149 | 19 | n/a | n/a | n/a | 21 | 147 |
| R3, day 1 | 149 | 145 | 4 | n/a | n/a | n/a | 4 | 145 |
| R non-grid, day 1 | 1956 | n/a | n/a | n/a | n/a | n/a | 1310 | 1111 |
| R1, day 2 | 189 | 111 | 78 | 69 | 28 | 92 | 75 | 114 |
| R2, day 2 | 172 | 152 | 20 | 131 | 6 | 35 | 23 | 149 |
| R3, day 2 | 183 | 165 | 18 | 148 | 3 | 32 | 13 | 170 |
| R non-grid, day 2 | 2011 | n/a | n/a | n/a | n/a | n/a | 1533 | 1190 |
| S1 | 140 | 72 | 68 | 28 | 52 | 60 | 140 | n/a |
| S non-grid | 384 | n/a | n/a | n/a | n/a | n/a | 499 | n/a |
| S1 shank 1 | 6 | 6 | 0 | 2 | 3 | 1 | 6 | n/a |
| S1 shank 2 | 30 | 30 | 0 | 2 | 28 | 0 | 30 | n/a |
| S1 shank 3 | 45 | 24 | 21 | 10 | 21 | 14 | 45 | n/a |
| S1 shank 4 | 59 | 12 | 47 | 14 | 0 | 45 | 59 | n/a |
| S non-grid shank 1 | 142 | n/a | n/a | n/a | n/a | n/a | 142 | n/a |
| S non-grid shank 2 | 86 | n/a | n/a | n/a | n/a | n/a | 86 | n/a |
| S non-grid shank 3 | 66 | n/a | n/a | n/a | n/a | n/a | 66 | n/a |
| S non-grid shank 4 | 90 | n/a | n/a | n/a | n/a | n/a | 90 | n/a |

**h**

| Rat | Cells | Grid | Pure grid | Grid-HD | Min. OF | Min. WW | Min. SWS | Min. REM | Min. total |
|---|---|---|---|---|---|---|---|---|---|
| Rat Q | 2094 | 163 | 159 | 4 | 31.4 | 42.4 | 105.0 | 17.5 | 360.8 |
| Rat R day1 | 2460 | 483 | 387 | 96 | 127.7 | 44.9 | 0 | 0 | 236.4 |
| Rat R day2 | 2571 | 544 | 428 | 116 | 21.1 | 0 | 137.0 | 30.9 | 259.6 |
| Rat S | 546 | 140 | 72 | 68 | 30.64 | 25.0 | 87.7 | 9.8 | 250.0 |

**Extended Data Fig. 2 | Grid module identification and properties.**
**a**–**d**, Clustering of grid modules (**a**, Rat 'Q'; **b**, Rat 'R', day 1; **c**, Rat 'R', day 2; **d**, Rat 'S'). For all experiments, coarse spatial autocorrelograms were first calculated from all cells' OF firing rate maps (n cells as shown in **g**). UMAP was then used to reduce the $M$-dimensional autocorrelograms (where $M = 668$ spatial bins) to a two-dimensional point cloud, where each point represented the autocorrelogram of a single cell, and distances between points represented the similarity between autocorrelograms. Left scatterplot in **a**–**d**: 2D point cloud, with points colour-coded according to cluster ID. Clusters were identified by applying the density-based clustering algorithm DBSCAN to the 2D point cloud. In every recording, the largest cluster (in grey, labelled "main") comprised mainly non-grid cells, and the remaining smaller clusters (coloured) represented different modules of grid cells. The black crosses ("noise") are identified as outlier data points. The well-isolated clusters formed by grid cells support the notion that these cells are a distinct functional class, in contrast to the claim that grid-like characteristics are expressed by MEC cells to different extents[92]. Right pair of scatterplots in **a**–**d**: Combinations of three grid parameters (grid score, grid spacing and grid orientation) for co-recorded cells from each recording. Each dot corresponds to one autocorrelogram (one cell). Dots are coloured by cluster ID as in **a**. **e**, Comparison of grid-cell spatial periodicity in the open-field arena (OF) and on the wagon-wheel track (WW). Top: firing rate map and corresponding autocorrelogram for an example grid cell in OF (left) and WW (right). For the purposes of this comparison, the same position bins were applied to both environments, resulting in cropping of the

outermost parts of WW. Colour coding as indicated by scale bar; peak rates 16.1 Hz (OF) and 15.8 Hz (WW); range of autocorrelation values: −0.56 to 0.83 and −0.58 to 0.71, respectively. Note the more irregular appearance of the autocorrelogram for WW. Bottom: scatter plots showing grid scores of all grid cells in OF ($x$ axis) and WW ($y$ axis). Colours refer to the module assignment in **a**. Note the bias for points to lie in the lower-right quadrant, reflecting generally higher grid scores in OF than in WW. **f**, As for **e**, but controlling for differences in behavioural coverage of OF and WW environments. It is possible that the lower WW grid scores in **e** were a product of sparser behavioural coverage of the WW environment (animals visited only positions on the track). To control for this possibility, we created "masked OF" (MOF) rate maps by removing spatial bins from the original OF rate map which were not visited by the animal in WW. In all modules, grid scores in the "masked" OF condition were higher than in WW (grid score mean ± S.E.M. across all cells: OF: 0.677 ± 0.017, WW: 0.360 ± 0.017, $N = 618$ cells, $P$ values for the 6 modules ranged from $1.26 \times 10^{-14}$ to 0.03, $Z$-values ranged from 2.12 to 7.71, Wilcoxon signed-rank test). Top row shows the same example cell as in **e** after leaving the same subset of position bins in OF as in WW. Bottom row shows comparison of grid scores for MOF and WW. As in **e**, grid scores are lower for WW, indicating that grid periodicity is reduced in WW even when differences in spatial coverage are accounted for. **g**, Table showing total number of cells and number of pure grid cells and conjunctive grid × direction cells. **h**, Number of cells (as in **g**) broken down on recording sessions, with session lengths in minutes indicated for open field (OF), wagon wheel (WW), slow-wave sleep (SWS) and REM sleep.

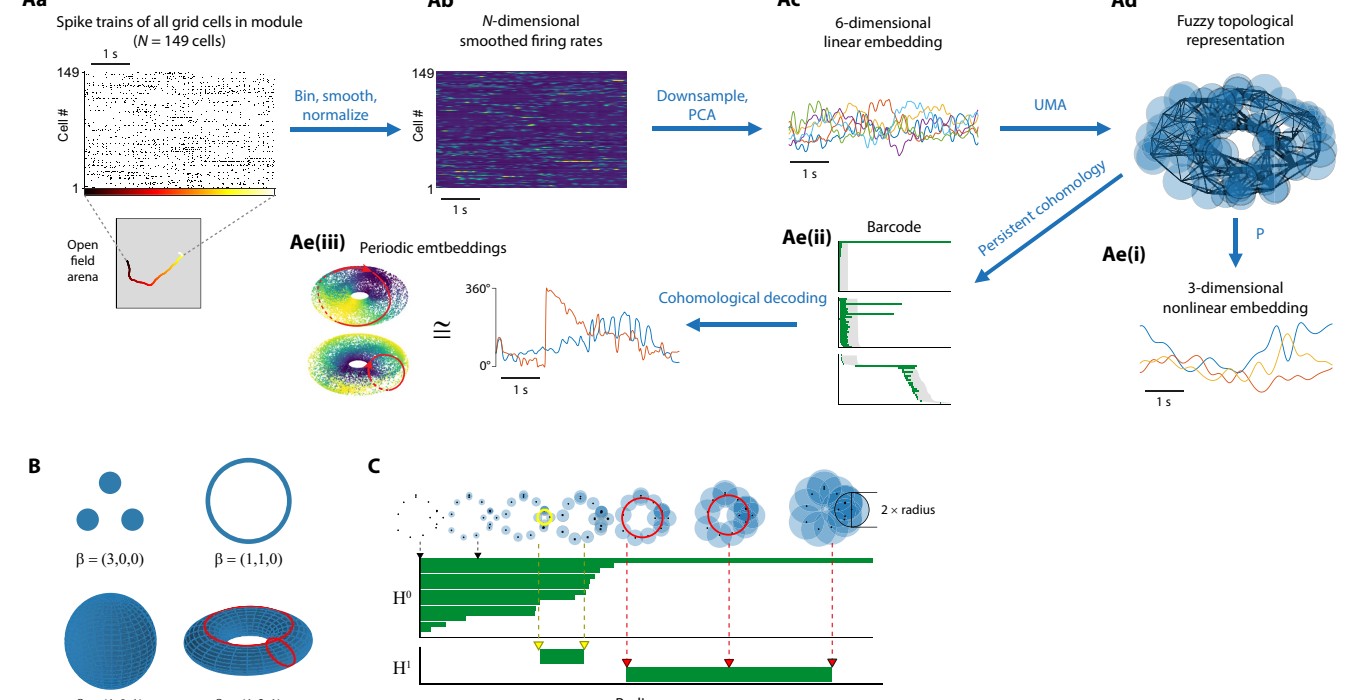

**Extended Data Fig. 3 | Preprocessing steps for visualization and detection of toroidal topology. A**, Flow diagram showing method for extracting low-dimensional embeddings of neural activity. The animal foraged in an OF arena while spikes from 149 grid cells shown in Fig. 1a were recorded (**Aa**; cells are ordered arbitrarily). A 5-second example behavioural trajectory is highlighted, with colour indicating elapsed time. The spike trains were binned in time ($N$ bins) and then smoothed and normalized, yielding a matrix of $N$-dimensional population activity vectors (**Ab**). After temporally downsampling and $z$-scoring the neural activity, PCA was applied to the $N$-dimensional neural activity, yielding a six-dimensional linear embedding (**Ac**). This preserved the grid structure in the activity (Extended Data Fig. 4b, c), while mitigating drawbacks associated with high-dimensional spaces (the "curse of dimensionality")[48]. The six principal components were then passed through a second, nonlinear, dimensionality reduction step by UMAP, which generated a three-dimensional nonlinear embedding (**Ae(i)**) allowing the toroidal structure to be visualized. UMAP consists of two steps: first, a fuzzy topological graph representation is constructed (i.e. a "Uniform Manifold Approximation" - UMA) using a distance metric in the high-dimensional space (**Ad**); second, to obtain the lower-dimensional projection (P), the coordinates of corresponding points in fewer dimensions are optimized to have a similar fuzzy topological representation. In the persistence analysis, we applied persistent cohomology to the fuzzy topological representation of the high-dimensional point cloud (**Ae(ii)**) and subsequently used cohomological

decoding to obtain a two-dimensional projection of the original $N$-dimensional point cloud (**Ae(iii)**); right, showing a 5-second snippet; left, embedded in 3D, points are coloured by each angular coordinate, whose direction is indicated by a red arrow). **B**, Cohomology can help differentiate topological spaces such as the union of three discs (upper left), a circle (upper right), a sphere (lower left) and a torus (lower right) by counting the number of topological holes ($\beta$) in different dimensions. A disc has a 0D hole (a connected component); a circle additionally has a 1D hole; a (hollow) sphere is a connected component and has a 2D hole (a cavity); a torus is a connected component with two 1D holes (illustrated with red circles) and one 2D hole (a cavity in the interior of the torus). **C**, Persistent cohomology tracks the lifetime of topological holes in spaces associated with point clouds. Top: The radius of balls centred at each data point in the point cloud is continuously increased (left to right). The union of the balls forms a space with possible holes. The lifetime of a hole during expansion of the radius is defined as the radial interval from when the hole first appears until it is filled in. Note the short lifetime of the hole marked with a red circle and the long lifetime of the hole indicated with a yellow circle. Second and third row: The lifetime of each hole of dimension zero ($H^0$) and one ($H^1$) in the example in the top row is indicated by the length of a bar (in green) in the barcode diagram. Two 1D holes are detected: the first bar, corresponding to the red hole in the top row, is short and regarded as noise, and the second, corresponding to the yellow hole, is substantially longer and captures the prominent topology of the point cloud.

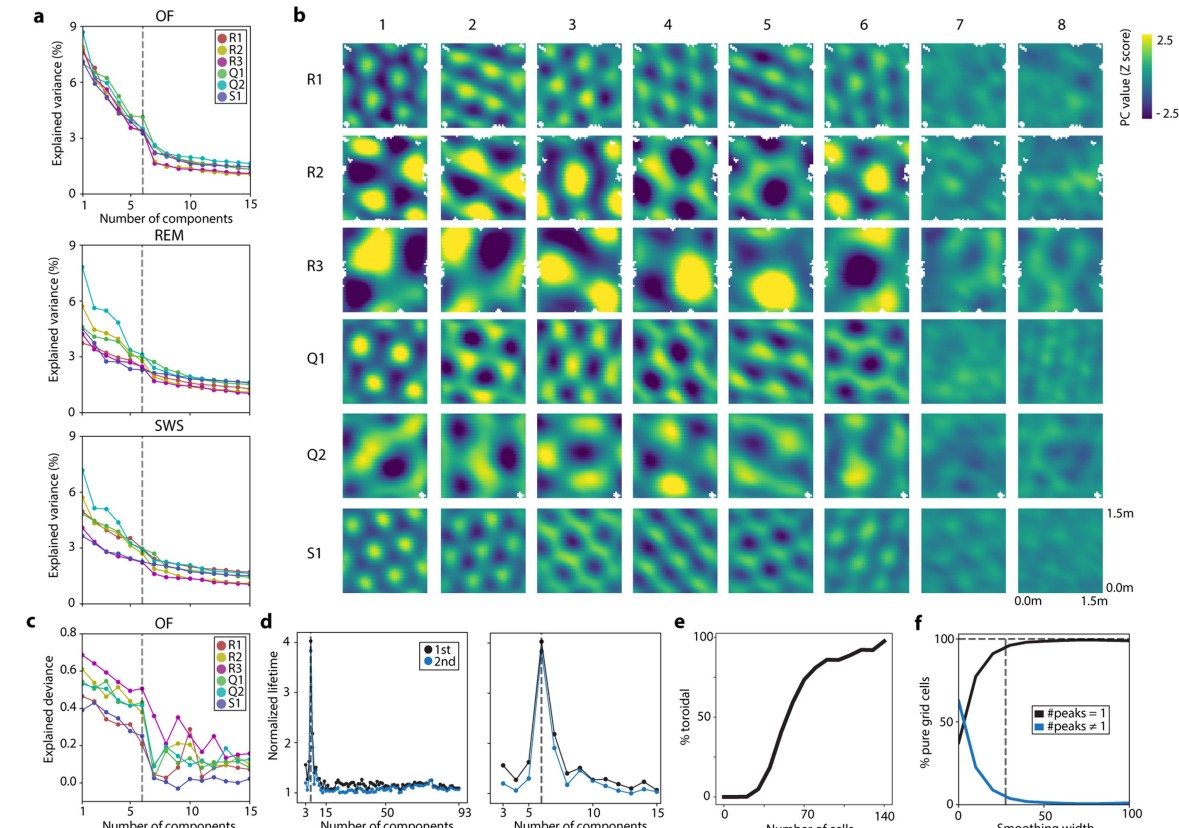

**Extended Data Fig. 4 | Analysis of principal components, number of cells and number of toroidal peaks. a**, Variance explained by the first 15 principal components (PCs) after applying PCA to the $n$-dimensional neural activity, shown for each module. Note that during OF, a particularly large amount of variance is explained by the first 6 PCs, followed by a sharp drop in the 7th PC, in all modules. A drop in variance explained is also seen after the 6th PC in REM and SWS. **b**, The first six PCs contain a grid-like representation at the population level. Each panel shows the mean value of one PC as a function of the animal's position in the OF. PC value is colour-coded as indicated by the scale bar. The 8 first PCs are arranged in descending order of explained variance (columns, from left to right), and are shown for each module (in rows). Note the presence of grid-like structure, which is particularly strong in the first six PCs, irrespective of the grid spacing. These six grid-like PCs correspond to the set with the highest explained variance in **a**. $z$-scored PC values are indicated by the scale bar (see Supplementary Methods for theoretical explanation of the six-dimensionality). **c**, Line plots showing the goodness-of-fit of a Gaussian GLM model based on the position in the spatial environment (OF) fitted to each principal component (components as in **a**). This is measured (as in Fig. 2d) as the explained deviance of the model showing that the six first components are better explained by space than the subsequent components for each module.

**d**, Line plots showing the lifetime of the two longest-lived $H^1$-bars (longest-lived – "1st", black; second longest-lived – "2nd", blue) divided by the lifetime of the third longest-lived $H^1$-bar as a function of number of principal components kept in the persistence analysis of R1 day 1 OF ($n = 93$ cells). This heuristic measures how clearly the two longest-lived $H^1$-bars (expected to be long for a torus) separates from the third (expected to be short), thus indicating how clearly the barcode displays toroidal topology. This is clearly the case when using 6 principal components in this dataset. **e**, The percentage of subsamples of R2 (resampled randomly 1,000 times per number of cells; total $n = 149$ cells) for which toroidal structure was detected in the parameterization given by the two most persistent 1D bars in the barcode (as in Extended Data Fig. 5). Note that approximately 60 cells were needed for the probability of detecting toroidal structure exceed 50%. **f**, Effect of varying spatial smoothing on the number of peaks in toroidal rate maps. The $y$ axis displays the percentage of single-peaked (black) and multi-peaked (blue) toroidal rate maps of all grid cells ($n = 2,727$ cells) pooled across modules and behaviour conditions. The vertical dashed line marks the smoothing width used in Extended Data Fig. 10, and the horizontal dashed line marks 100%. Note that cells with single peaks quickly describe the majority of the pooled cells.

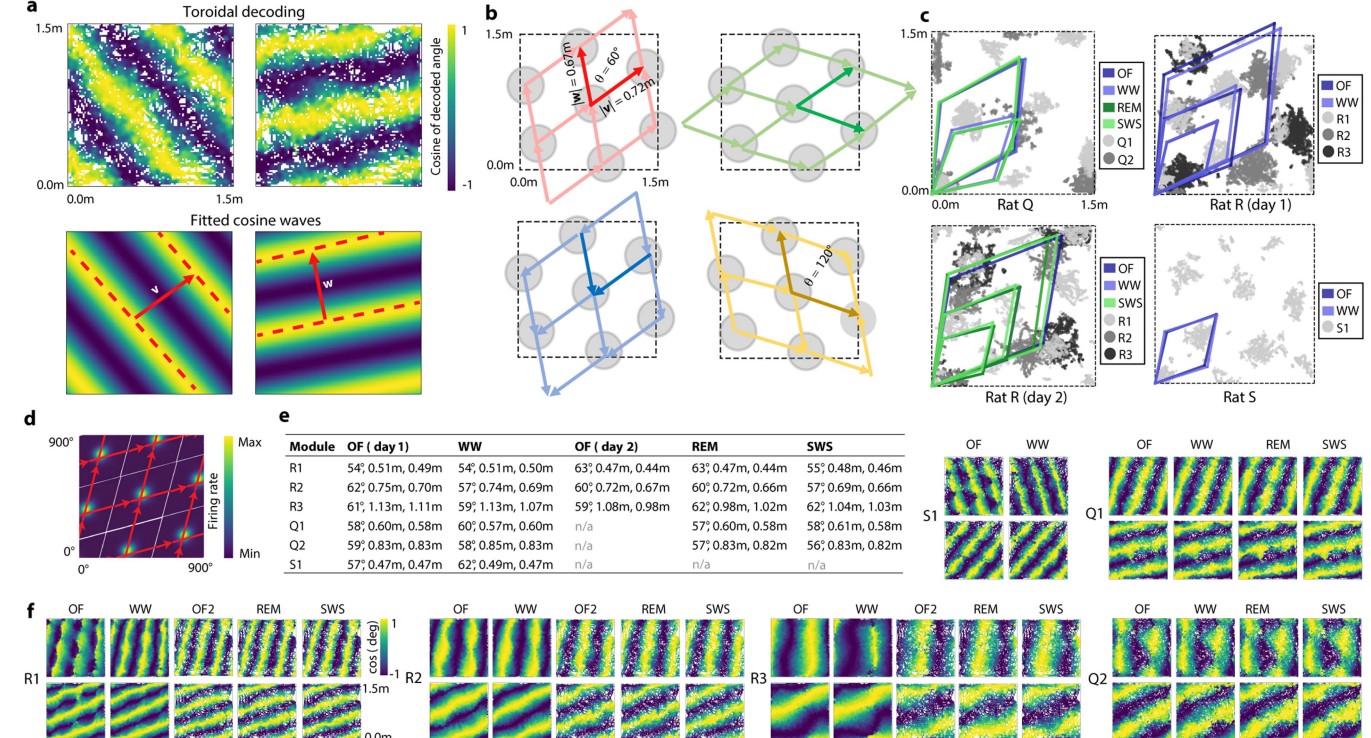

**Extended Data Fig. 5 | Mapping of decoded circular coordinates onto the open field allows geometrical interpretation of toroidal structure. a**, Top row: Toroidal coordinates given by cohomological decoding from activity of grid module R2 during OF foraging, mapped onto the recording box. In each plot, colour indicates the mean value of the cosine of each of the two circular coordinates. The mappings of both coordinates show 2D striped patterns, with similar periods but distinct angles. Bottom row: A cosine wave is fitted to each coordinate to obtain the direction of the toroidal axes. The period and angle of the cosine wave in the plane may be represented by spatial vectors, **v** and **w**, with corresponding length and orientation. Note the clear transversality of the two circles, expressed in the directions of the two vectors, further confirming the toroidal identification of the data. **b**, The periods and angles of the cosine waves in **a** reflect the scale and orientation of the grid module. Taking the origin of the vectors in **a** to be alike, we see that the vectors span a parallelogram with approximately equal side lengths (0.67m and 0.72m) and an angle of 60 degrees, suggesting a rhomboidal tile representing the toroidal structure (top left). When repeated across the environment, the tile depicts the hexagonal grid pattern of the grid-cell module, confirming that the product of the two decoded circles defines a hexagonal ("twisted") torus. As the orientation of the circular coordinates is arbitrary, the directions of the axes may be any of the following: reversely oriented (blue arrows), a different 60-degree pair of axes (green), or have a relative angle of 120 degrees (yellow). **c**, Rhombi of each module for each OF session (*n* cells as in Extended Data Fig. 2g), given by the cosine wave fitted to the toroidal coordinates (as in **b**). The toroidal parametrizations were obtained independently in different behavioural conditions (colour-coded), then used to decode the module's activity during OF foraging, and subsequently mapped as a function of the rat's position in the environment (see **f**). Positions of downsampled spikes from example cells of each module are shown in greyscale to illustrate grid scale and orientation. The consistent angle and side lengths suggest the geometry of the rhombus is retained across brain states and environments, with a constant scale relationship between modules. **d**, Mean value of a single neuron in rhomboidal coordinates displays a single bump (as in Fig. 2a), which, when repeated and arranged to tesselate a 2D surface, reveals a grid-like pattern in the activity of the grid cell, akin to its spatial firing. **e**, Table of side lengths and angles of the cosine waves that form the rhombi in **c**, shown for each grid module and each condition (*n* cells as in Extended Data Fig. 2g). **f**, Visualization of the cohomological decoding of toroidal coordinates as a function of physical space (one visualization for each grid module during each condition, with the toroidal parametrizations aligned to the same axes before creating the rate maps; *n* cells as in Extended Data Fig. 2g). All barcodes which indicated toroidal structure exhibited periodic stripes in the OF, with phase and orientation corresponding to the two-dimensional periodicity of the grid pattern of the respective module. SWS* refers to the decoding when considering only "bursty" (B) cells of R1 as given by the correlation clustering method described in Fig 4b.

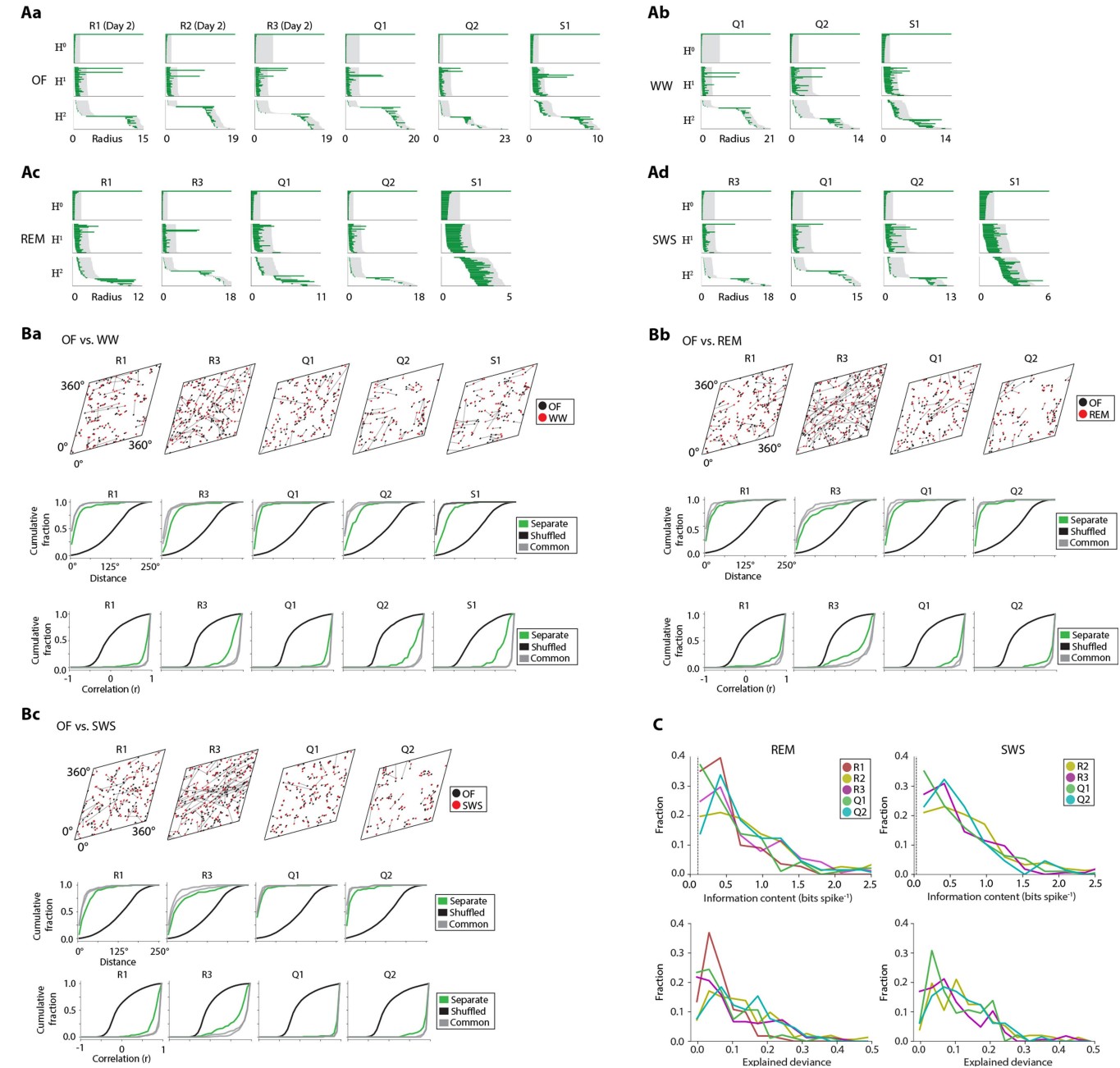

**Extended Data Fig. 6 | Barcodes and toroidal tuning statistics for grid modules or recording sessions not included in Figs. 2–4.** Data are shown for six grid-cell modules: R1, R3, Q1, Q2, S1 and R2 (*n* cells as in Extended Data Fig. 2g). Toroidal structure is clearly present across environments and behavioural states. **Aa–Ad**, Barcode diagrams (as in Fig 1e, f) showing the results of the persistent cohomology analysis on open-field (OF), wagon-wheel track (WW) or sleep (REM or SWS) data. **Ba–Bc**, Preservation of toroidal field centres between conditions: OF vs WW (1), OF vs REM (2) OF vs SWS (3). Top row in each panel: Distribution of grid cells' receptive field centres on the inferred torus for OF and WW as well as sleep states, similar to Fig 2e. Each dot signifies the field centre of an individual grid cell. Grey lines connect field centres of the same cell across conditions. Note the proximity of red-black pairs (after separate alignment for the two recording sessions of each panel). Middle and bottom rows: Cumulative distributions showing stability of grid cells' toroidal tuning between brain states, as in Fig. 2f, g. Distributions show peak field distance (middle) and Pearson correlation of pairs of toroidal rate maps (bottom). Labelling as in Fig. 2e–g. **C**, Top: Histograms of the information content carried by individual cells' activity about position on the inferred torus during REM (left) and SWS (right). Counts (fractions of the cell sample) are shown as a function of information content (in bins of 0.28 bits/spike) for all grid modules (colour-coded). The vertical dashed line (close to zero) shows mean information content for shuffled distributions (*n* = 1,000 shuffles). The majority of cells have a higher information content. Bottom: Explained deviance of a GLM model fitted to the spike count with toroidal coordinates during REM (left) and SWS (right) as regressor. Distributions show counts (fractions of the cell sample) as a function of explained deviance, in bins of 0.035, for all grid modules. Values larger than 0 indicate that the fitted model explains the data better than a null model that assumes a constant firing rate.

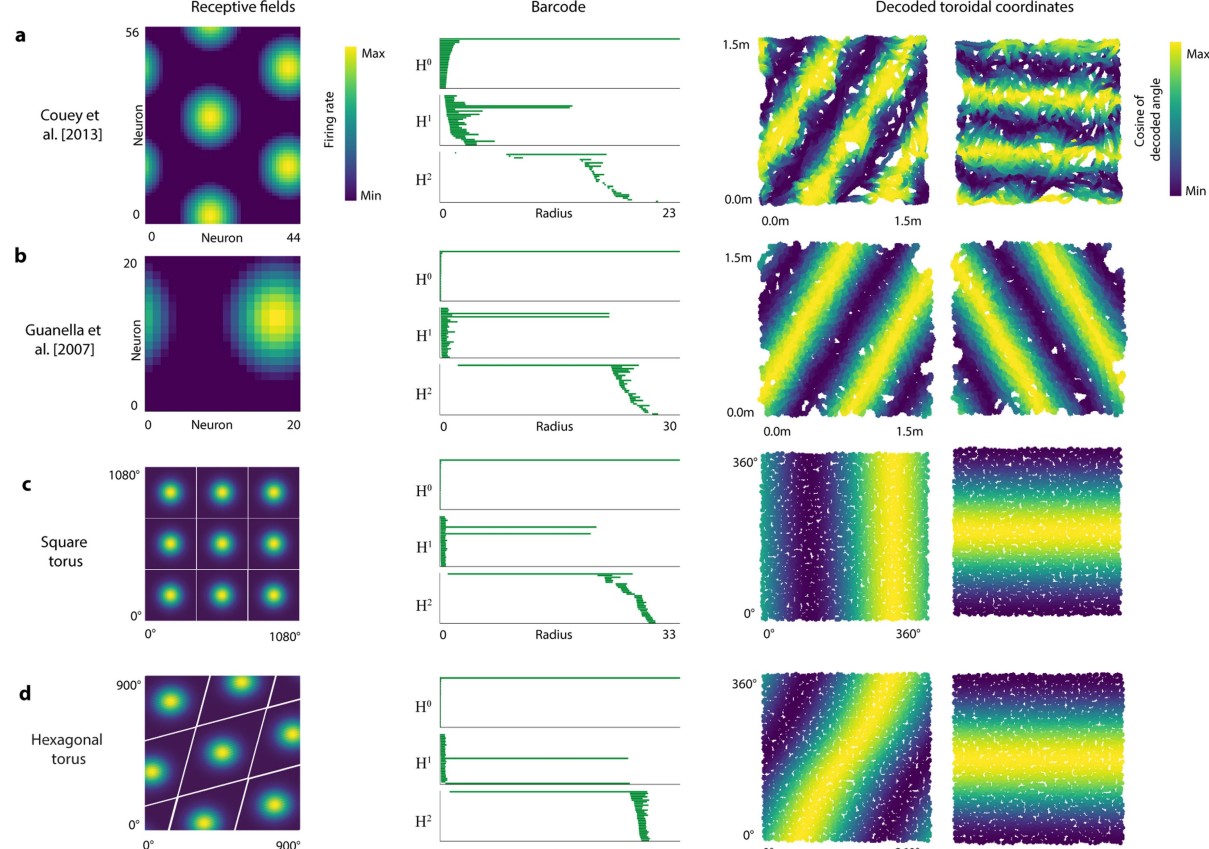

**Extended Data Fig. 7 | Barcodes and decoding of simulated firing activity for two grid-cell CAN models (with no noise), and for two point clouds randomly sampled on a hexagonal and a square torus. a**, Persistent cohomology analysis of a simulated grid-cell network based on the CAN model from Couey et al (2013)[11] during OF foraging. Left: Colour-coded firing rates for a single time frame of the 56 × 44 grid cells, shown at their respective positions on the neural sheet. Middle: Barcode of the simulated data. Arrows point to one 0D, two 1D and one 2D bar with long lifetimes, indicating toroidal structure. Right: Each coordinate of the toroidal parametrization of the two longest lived 1D features is mapped onto the spatial trajectory, colour-coded by its cosine value (as in Extended Data Fig. 5a, f). The resulting striped patterns of the two maps are oriented approximately 60 degrees relative to each other, as expected from a hexagonal torus network structure (see **d**). **b**, Analysis of a random sample of 100 grid cells (of a total of 400 cells) of a simulated grid cell network, using the twisted torus CAN model formulated by Guanella et al (2007)[10]. Left: Firing rates of the cells in the network at a single time frame. The model generates a single bump of activity based on both inhibitory and

excitatory, asymmetric connections representing a twisted torus. Barcode (middle) and cohomological decoding of toroidal position (right) are shown as in **a**. The barcode shows four prominent bars: one 0D bar, two 1D bars and one 2D bar, similar to that of a torus. Note that the pair of stripes in toroidal coordinates are oriented 60 degrees relative to each other. **c**, **d**, To verify the expected barcodes and decoding of a torus and compare with both real and synthetic grid cell data, we performed the same topological analysis on point clouds sampled from two idealized toroidal parametrizations ($n$ = 2,500 points): a 4D description of a square torus (**c**) and a 6D embedding of a hexagonal torus (**d**). Left: Representing the firing of a cell as a Gaussian function centred at a single toroidal coordinate on the toroidal sheet results in a square (**c**) and hexagonal (**d**) firing pattern, when arranged to tessellate a 2D surface. Middle: The expected barcode of a torus (one 0D, two 1D, and one 2D bar clearly longer than the other bars) is seen in both cases. Right: each sampled angle is coloured according to the decoded toroidal coordinates. Note the difference in the relative angle of the pair of stripes between the square and the hexagonal torus.

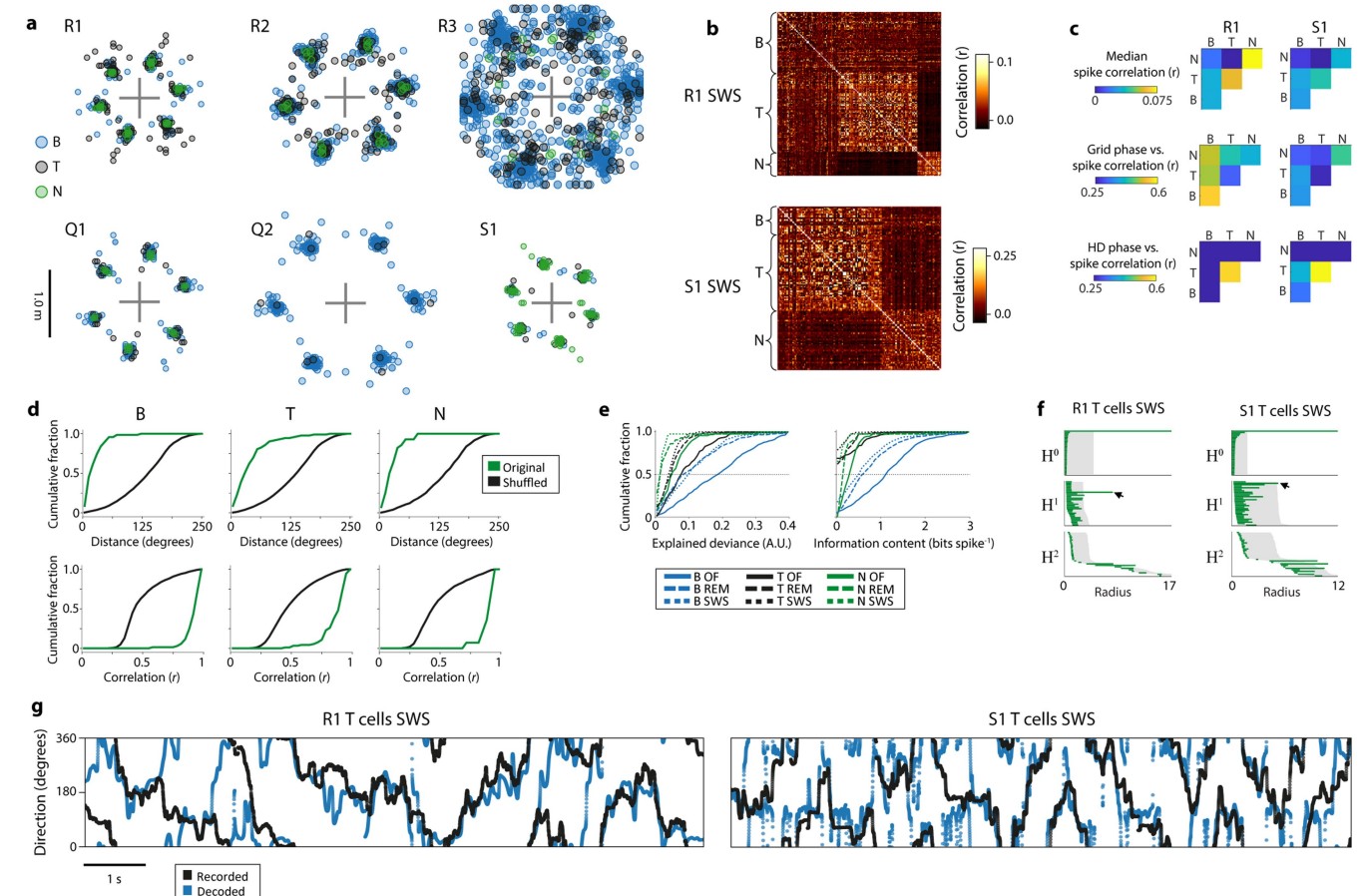

**Extended Data Fig. 8 | Subpopulations of grid cells with different temporal spiking statistics have different degrees of toroidal selectivity.**
**a**, Geometry of grid-cell pattern of all six modules with classes of grid cells (B, bursty; T, theta-modulated; N, non-bursty; as defined in Fig. 4). Each plot shows the locations of the innermost six peaks of the spatial autocorrelogram for every grid cell in one module. Each dot indicates the position of one peak from one cell (total of 6 dots per cell); dots are coloured by the cell's class. The grey crosshair indicates the centre of the autocorrelogram. **b**, Correlation matrix showing pairwise correlation of firing rates for all grid cells belonging to S1 (left; $n = 73$ cells) and R1 (right – same data as for autocorrelogram distance matrix in Fig. 4b; $n = 111$ cells). Correlation is colour-coded according to the scale bar, with minimum and maximum defined as the 1st and 99th percentile, respectively, of the pairwise correlation distribution for each module. Rows and column (cells) are ordered according to class, as assigned by the clustering analysis shown in Fig. 4. Each cluster displays strong inner correlation structure for both modules during SWS. Cluster boundaries are indicated on the $x$ axis of the correlation matrix. **c**, Summary of pairwise correlations of SWS activity for grid cells in modules R1 and S1, shown according to cell class. In each matrix plot, rows and columns indicate cell classes, and each element represents all pairs of grid cells from the classes corresponding to the row and column. Matrix elements are colour-coded to represent (top) the median of the spike train Pearson correlation $r$ value across all cell pairs, (middle) Spearman rank correlation between cell pairs' grid (toroidal) phase offsets and their spike train Pearson correlation $r$ values, (bottom) same as middle, but for head-direction phase instead of grid phase. Number of cell pairs were as follows: module R1,

B-B 2346, B-T 6348, B-N 1932, T-T 4186, T-N 2576, N-N 378; module S1 B-B 378, B-T 1680, B-N 1456, T-T 1770, T-N 3120, N-N 1326. Note that, in agreement with the topological analyses, the correlation between cell pairs' grid phases and their spike-time correlations are weaker for theta-modulated cells than non-bursty and particularly bursty cells. This drop is explained by an increase in the correlation with head direction, suggesting, as expected in conjunctive cells, that head direction accounts for much of the variation in these cells, unlike the other classes. Furthermore, the median spike correlation for pairs of theta-modulated and non-bursty cells is higher than for bursty cells, indicating a stronger positive correlation bias, consistent with more global fluctuations of activity in these populations. **d**, Cumulative distributions showing distance between toroidal field centres (upper) and Pearson correlation $r$ values (lower) for toroidal rate maps of grid cells in each class as in Fig. 2f, g, but here comparing awake behaviour in OF with SWS, $n$ cells = 523(B), 229(T) and 95(N) cells for OF and 495(B), 169(T), 43(N) cells for REM and SWS. $n = 1,000$ shuffles. **e**, Cumulative distributions showing toroidal explained deviance (left) and information content (right) for all grid cells in each class – bursty (B), theta-modulated (T) and non-bursty (N) – and for each of three conditions – OF, REM and SWS. Cells are from all modules. $n$ cells as in **d**. **f**, Barcode of T-class grid cells from modules R1 (left; $n = 92$ cells) and S1 (right; $n = 60$ cells) during SWS reveals a single prominent long-lived $H^1$ bar (indicated by black arrow). **g**, Cohomological decoding of the longest-lived $H^1$ bar in each barcode in **f** reveals strong correlation with recorded head direction. Recorded head direction (black) and decoded direction (blue) are shown as a function of time (total snippet length 10 s).

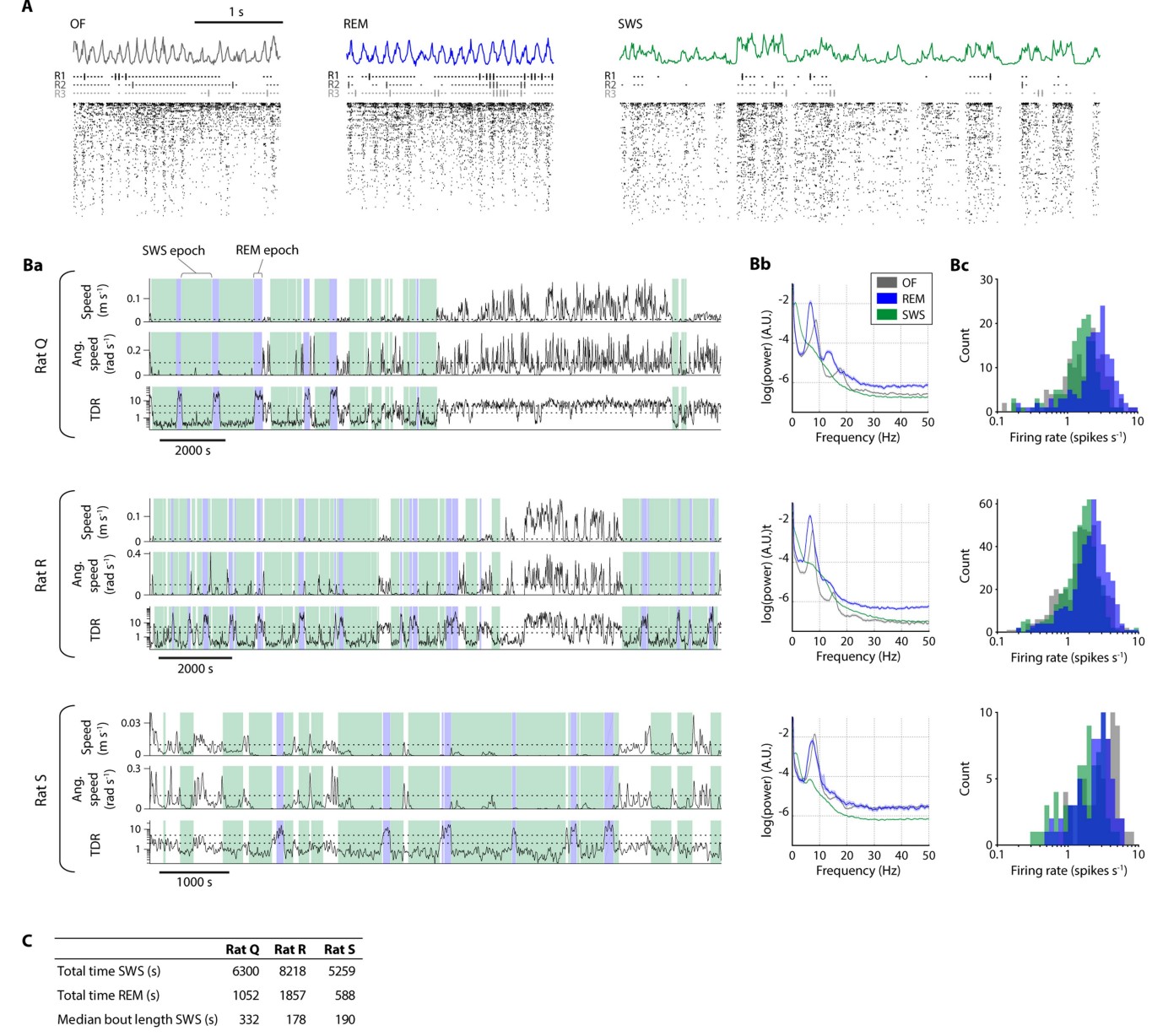

| C | Rat Q | Rat R | Rat S |
|---|---|---|---|
| Total time SWS (s) | 6300 | 8218 | 5259 |
| Total time REM (s) | 1052 | 1857 | 588 |
| Median bout length SWS (s) | 332 | 178 | 190 |
| Median bout length REM (s) | 175 | 79 | 96 |

**Extended Data Fig. 9 | Classification of sleep and wake states based on behavioural and neural activity during rest sessions. A**, Example traces of MEC multi-unit activity (upper; coloured lines), and rasters of spike times of 444 grid cells (lower; black dots) recorded from rat 'R' during OF foraging, REM sleep and slow-wave sleep (SWS). Cells are ranked from top to bottom by the number of spikes fired during the example time window. Note the presence of regular theta waves (5–10 Hz) during OF and REM, and presence of slower, more irregular fluctuations between active "up-states" and silent "down-states" during SWS. Middle: times of population activity vectors (calculated in 10 ms time bins) which were selected for persistent cohomology analysis, for each module (R1-R3). Each dot indicates a vector which was included in the initial downsampled set of 15,000 vectors with the highest mean firing rate across cells in the module. Vertical ticks indicate the subset of these vectors which were retained after using a density-based method to reduce the data to a representative point cloud. Note that during SWS, all of the selected population activity vectors occurred during up-states. **B**, Classification of sleep/wake states based on behavioural and neural activity during rest sessions. Each of the three horizontal blocks shows a recording from one animal. Rat 'R' day 1 did not contain a rest session and is not shown on this figure. **Ba**, Detection of REM and SWS sleep epochs in the rest session. The plots show the time courses of the three variables used for detecting REM and SWS epochs. Top panel of each block: animal locomotion speed; middle panel: the animal's head angular speed; bottom panel: the ratio of the amplitude of theta (5–10 Hz) and delta (1–4 Hz) frequency bands in the multi-unit spiking activity (theta/delta ratio, TDR). **Bb**, Log-power spectra of MEC multi-unit activity during each sleep/wake state. The line and shaded area indicate the mean and 95% bootstrap confidence intervals, calculated across time windows (confidence intervals are narrow). Note the pronounced peak corresponding to the theta band (5–10 Hz) during OF and REM, and the higher power in the delta band (1–4 Hz) during SWS. **Bc**, Histograms showing distributions of firing rates for all grid cells during each sleep/wake state (number of grid cells: rat 'Q' 159, rat 'R' 428, rat 'S' 72). **C**, Table showing total time and median bout length of recorded sleep for each animal.

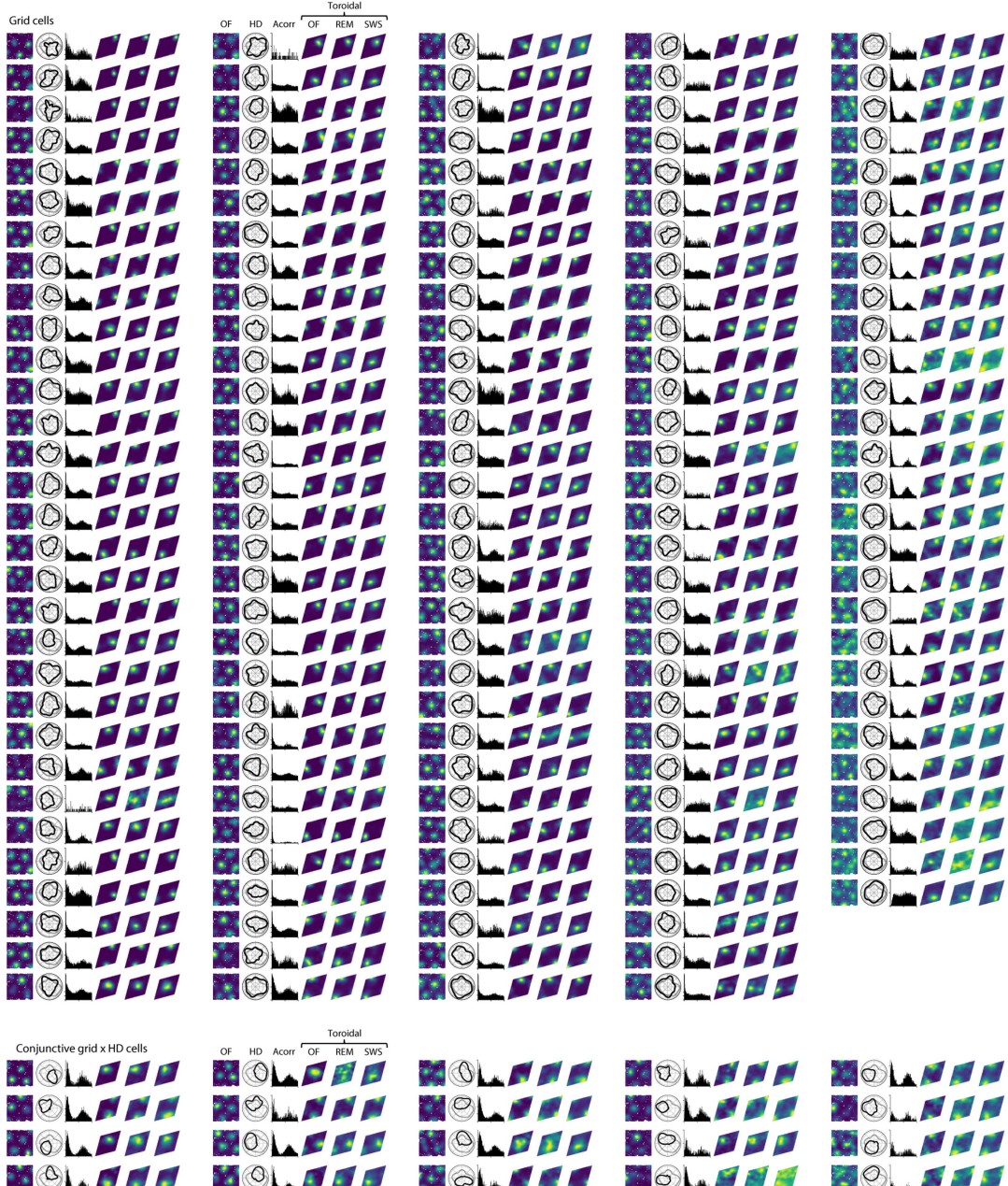

**Extended Data Fig. 10 | Tuning to coordinates in space and on the inferred torus for all grid cells of module R2 (separated into pure and conjunctive categories) on recording day 2.** Plots show all 152 cells in module R2, a subset of which is shown in Fig. 3b. Plots from left to right: OF firing rate map, head-direction tuning curve (black) compared to occupancy of head directions (light grey), temporal autocorrelogram, toroidal firing rate maps for OF, REM and SWS. The full set of plots, for all remaining grid cells of all recordings, is shown in Supplementary Information.

Nils A. Baas
Benjamin A. Dunn
Edvard I. Moser

# nature research

# Reporting Summary

Nature Research wishes to improve the reproducibility of the work that we publish. This form provides structure for consistency and transparency in reporting. For further information on Nature Research policies, see our Editorial Policies and the Editorial Policy Checklist.

## Statistics

For all statistical analyses, confirm that the following items are present in the figure legend, table legend, main text, or Methods section.

| n/a | Confirmed | |
|---|---|---|
| ☐ | ☒ | The exact sample size ($n$) for each experimental group/condition, given as a discrete number and unit of measurement |
| ☐ | ☒ | A statement on whether measurements were taken from distinct samples or whether the same sample was measured repeatedly |
| ☐ | ☒ | The statistical test(s) used AND whether they are one- or two-sided<br>*Only common tests should be described solely by name; describe more complex techniques in the Methods section.* |
| ☐ | ☒ | A description of all covariates tested |
| ☐ | ☒ | A description of any assumptions or corrections, such as tests of normality and adjustment for multiple comparisons |
| ☐ | ☒ | A full description of the statistical parameters including central tendency (e.g. means) or other basic estimates (e.g. regression coefficient) AND variation (e.g. standard deviation) or associated estimates of uncertainty (e.g. confidence intervals) |
| ☐ | ☒ | For null hypothesis testing, the test statistic (e.g. $F$, $t$, $r$) with confidence intervals, effect sizes, degrees of freedom and $P$ value noted<br>*Give P values as exact values whenever suitable.* |
| ☒ | ☐ | For Bayesian analysis, information on the choice of priors and Markov chain Monte Carlo settings |
| ☒ | ☐ | For hierarchical and complex designs, identification of the appropriate level for tests and full reporting of outcomes |
| ☐ | ☒ | Estimates of effect sizes (e.g. Cohen's $d$, Pearson's $r$), indicating how they were calculated |

*Our web collection on statistics for biologists contains articles on many of the points above.*

## Software and code

Policy information about availability of computer code

| Data collection | Commercial software: Motive (OptiTrack) version 2.2.0; MATLAB (MathWorks) version r2019b<br>Open-source software: SpikeGLX (https://billkarsh.github.io/SpikeGLX) versions 20190724 and 20190919 |
|---|---|
| Data analysis | Commercial software: MATLAB (MathWorks) version r2020b, Python version 3.7<br><br>Open-source code (for MATLAB):<br>Kilosort version 2.5 (https://github.com/MouseLand/Kilosort)<br>UMAP version 1.4.1 (https://www.mathworks.com/matlabcentral/fileexchange/71902-uniform-manifold-approximation-and-projection-umap)<br>Chronux version 2.10 (http://chronux.org/)<br><br>Open-source Python packages:<br>umap  0.3.10<br>ripser 0.4.1<br>numba 0.48.0<br>scipy 1.4.1<br>numpy 1.18.1<br>scikit-learn 0.22.1<br>matplotlib 3.1.3<br>h5py 2.10.0<br>gudhi 3.4.1.post1 |

For manuscripts utilizing custom algorithms or software that are central to the research but not yet described in published literature, software must be made available to editors and reviewers. We strongly encourage code deposition in a community repository (e.g. GitHub). See the Nature Research guidelines for submitting code & software for further information.

## Data

Policy information about availability of data

All manuscripts must include a data availability statement. This statement should provide the following information, where applicable:

- Accession codes, unique identifiers, or web links for publicly available datasets
- A list of figures that have associated raw data
- A description of any restrictions on data availability

> The datasets generated during the current study will be available after publication at https://figshare.com/articles/dataset/
> Toroidal_topology_of_population_activity_in_grid_cells/16764508

# Field-specific reporting

Please select the one below that is the best fit for your research. If you are not sure, read the appropriate sections before making your selection.

☒ Life sciences     ☐ Behavioural & social sciences     ☐ Ecological, evolutionary & environmental sciences

For a reference copy of the document with all sections, see nature.com/documents/nr-reporting-summary-flat.pdf

# Life sciences study design

All studies must disclose on these points even when the disclosure is negative.

| | |
|---|---|
| Sample size | Samples included all available cells that matched the classification criteria for the relevant cell type. |
| Data exclusions | Cells with very low firing rates (below 0.05 Hz) were excluded because of their unsuitability for spike-train analysis. All non-grid cells were excluded, because they were irrelevant for analyses of toroidal structure. |
| Replication | For the six grid modules included in the study, in the results text we indicate for each result the number of modules in which the effect was found. For each statistical test we state the sample size (n) in the manuscript. |
| Randomization | The study did not involve any experimental subject groups; therefore, random allocation did not apply and was not performed. |
| Blinding | The study did not involve any experimental subject groups; therefore, experimenter blinding did not apply and was not performed. |

# Reporting for specific materials, systems and methods

We require information from authors about some types of materials, experimental systems and methods used in many studies. Here, indicate whether each material, system or method listed is relevant to your study. If you are not sure if a list item applies to your research, read the appropriate section before selecting a response.

### Materials & experimental systems

| n/a | Involved in the study |
|---|---|
| ☒ ☐ | Antibodies |
| ☒ ☐ | Eukaryotic cell lines |
| ☒ ☐ | Palaeontology and archaeology |
| ☐ ☒ | Animals and other organisms |
| ☒ ☐ | Human research participants |
| ☒ ☐ | Clinical data |
| ☒ ☐ | Dual use research of concern |

### Methods

| n/a | Involved in the study |
|---|---|
| ☒ ☐ | ChIP-seq |
| ☒ ☐ | Flow cytometry |
| ☒ ☐ | MRI-based neuroimaging |

## Animals and other organisms

Policy information about studies involving animals; ARRIVE guidelines recommended for reporting animal research

| | |
|---|---|
| Laboratory animals | Long Evans rats, male, age 3-4 months (300-500 g) |
| Wild animals | None |
| Field-collected samples | None |
| Ethics oversight | Protocols approved by the Norwegian Food Safety Authority (FOTS ID 18011 and 18013) . |

Note that full information on the approval of the study protocol must also be provided in the manuscript.

