## [Peer Review File · Nature]

Manuscript Title: Toroidal topology of population activity in grid cells

Editorial Notes:

Editorial Note

Reviewers were consulted about the authors' rebuttal and were satisfied with the responses at this stage.

Reviewer Comments & Author Rebuttals

.

Reviewer Reports on the Initial Version:

REF #1

Ever since the discovery of grid cells, it has been assumed that the population activity of grid cells having the same grid spacing and orientation should display a torus-like structure. The reason for this is simply that the hexagonal lattices associated to grid cell modules periodically identify disconnected regions in the plane, and the quotient space is a (hexagonal) torus. Seeing this structure in the data was just a matter of time, as it required high-density recordings (to get enough cells in a given module) as well as fast computational topology methods (to verify the topological structure within a high-dimensional space). So it is in some sense not surprising to see a verification of toroidal topology in grid cell population activity. It had to be there, simply because of the structure of grid fields.

Nevertheless, this paper is highly significant and goes well beyond a simple verification that there's a torus in the neural activity. The authors have collected a stunningly high-quality data set that allows them to detect and visualize the torus in multiple (six!) grid modules with a large number of cells (~150) per module. They have also tested for toroidal structure in environments where the hexagonal structure of grid fields is distorted, and in sleep (SWS and REM). In the wagon-wheel experiments, neurons were better tuned to the torus than to physical space, even when the torus was inferred using the other (open field) environment. The analysis is very convincing and suggests that the torus-like manifold emerges from the intrinsic network organization, and is not inherited from sensory inputs. This is about as thorough of an investigation as one can expect on an extremely interesting and important topic. I thus believe this study warrants publication in Nature. Moreover, the paper is well-written and the figures are polished and clear.

The manuscript did raise several questions that I hope can be addressed in a revision.

1. Why is there so much emphasis on the CAN? Continuous attractor networks are not the only model of recurrent network organization that could produce the torus structure. The authors seem to be contrasting CANs only to feedforward mechanisms, such as in this sentence of the discussion: "While toroidal topology can be generated by both CAN^{1,13–16} and feedforward^{17–22} mechanisms, the persistence of an invariant toroidal manifold across environments and behavioural states, under conditions that give rise to changes in the correlation structure of place-cell activity in the hippocampus^{11,12}, is predicted only by CAN models." But CAN models are not general recurrent networks: they have a very precise connectivity structure and a posit a continuum of fixed point attractors. There are other ways a recurrent network could be wired so that the torus emerges, where the connectivity is not fine-tuned and purely geometric as in CANs, and the activity is not given by fixed point attractors. None of this is ruled out by the data.

The author's findings are certainly consistent with CANs, but I do not believe they provide direct evidence for them. What they find evidence for is the strong role of recurrent – as opposed to feedforward – mechanisms governing grid cell activity. The first paragraph of the Discussion is a

perfectly accurate summary of their findings, and they are very significant. There is no need to overlay the CAN spin, and I find this to be the weakest aspect of the paper.

2. Why do the authors use persistent cohomology as opposed to persistent homology? Persistent homology is far more standard in topological data analysis (TDA), and is the main computation done by packages like Ripser (which the authors use). Moreover, the entire description of the topological analysis in the main text is identical to what I would have expected if persistent homology were being used. So are the results. So why the choice to use cohomology? The only hint I got was some mention of the cup product in the Supplementary Methods as an additional way to verify the torus structure (since other topological spaces are consistent with the same bar codes). But this is never mentioned in the main text and is a rather unsatisfying afterthought in the Supplementary Methods, especially since the text leading up to it is a standard discussion of persistent homology.

An additional minor point: the choice of coefficients (Z_{47}) should be justified. I understand why Z_2 wouldn't be good, but why the prime $p = 47$ and not something else? If it's just an arbitrary choice, the authors should say so.

3. The torus visualization uses PCA to get down to 6 dimensions, and UMAP to further reduce to 3-d. Of course, the whole point of TDA methods is to allow topological structure in point cloud data to be identified in high-dimensional data, where it can not be seen by eye. The authors thus perform TDA on the 6-dim'l data after only the PCA reduction. This makes me wonder: why not perform TDA on the original N-dim'l data? Does it not work there? If so, it'd be good to know and also to have some discussion of what goes wrong that makes dimensional reduction a required pre-processing step before performing topological analyses.

4. Is there any relation between bar length and grid field spacing? I found myself wondering if the bar codes carried information about which module the activity came from. I also wonder if there is any systematic asymmetry in the lengths of the two β_1 bars. The grid field lattices are hexagonal with a fairly regular hexagon, as far as I understand. This should predict something about the relative sizes of the two one-dimensional holes in the torus. Is the data consistent with this prediction?

5. Reference 85 (Giusti et. al.) also saw consistent topological structure between open field and sleep data (in place cells). Do the authors believe that place fields are also a CAN? If not, how do they explain the evidence for recurrent features dominating over feedforward ones in that context? (This is related to my first question. There are other types of recurrent network that are not CANs...)

REF #2

In the present manuscript, Gardner et al. have studied the organization of grid cell populations in the medial entorhinal cortex. Specifically, they used topological data analysis techniques to demonstrate that grid cell population activity has a toroidal topology. Although this is not the first time that these techniques are used to analyze neuronal population activity (e.g. for head-direction cells), applying these technique to grid cells requires the monitoring of large ensembles of neurons, which had never been achieved so far. To this end, the authors used Neuropixel probes, reporting up to 150 neurons within a single "module" of grid cells. The authors further demonstrate that the topological structure of population activity is preserved across environments and brain states, suggesting that the system is governed by local circuits and dynamics. While these findings are exciting and constitute an important milestone in our understanding of neuronal activity in vivo, the present manuscript suffers from a number of limitations and inconsistencies that need to be addressed. Overall, the dataset is quite inhomogeneous, raising concerns about the validity of some of the claims.

While the authors report a stunning recording of 149 grid cells from the same module in one rat, the authors should provide more details regarding the dataset. First, the authors presented the

same histology sections for the left and right hemispheres of rat Q, and again the same histology sections for shanks 2 and 3 of rat S. While this is certainly an honest mistake, this needs to be corrected. Furthermore, the sections should be presented without the probe overlaying to clearly show the electrode tracks. Next, it is interesting to see how grid cells are anatomically distributed, but it would be even more interesting to show how this compares to all other recorded neurons. The authors should report the total number of cells along the electrodes, and at least the proportion of “pure” grid cells (the only one considered for the topological analyses), conjunctive grid cells, and HD cells.

Overall, it would be interesting to present a table that precisely describes the content of each module. Presently, this information is difficult to access. For example, the number of neurons per module can be found in Ext. Data Fig 2e in a very small panel. By the way, this panel reveals another inconsistency, as it is stated that module R2 has 168 neurons, while in Fig. 1, it is said that it contains 149 neurons.

Along the same lines, the authors should clarify whether neurons were also recorded from the prosubiculum (PoR, the result section in the main text only refers to the MEC and parasubiculum). Although only a minority of cells in the dataset seem to be located in the PoR, this should be stated and/or discussed. In rat R (L), it is unclear whether grid cells of the module R2 were recorded exactly at the same location of the module R1 (color is confusing). Modules were determined with an interesting clustering procedure. The authors should briefly discuss whether the separation of the R1 module in two sub-modules corresponds to differences in anatomical distribution of the neurons (PoR/MEC? Left/Right?).

Analyzing the population data from each module separately, the authors demonstrate with topological data analysis that each module shows a toroidal structure. This is the main claim of the paper and the result is unambiguous. However, it would be interesting to compare these results with an analysis including all grid cells, including conjunctive ones - and why not, all recorded cells. Does it really distort the torus? One can argue that, by selecting pure grid cells alone, the result is obvious. The strength of such analysis would be to find a toroidal structure from the ensembles of neurons within one module (whose anatomical boundaries can be determined from the spatial properties of grid cells) and to infer post-hoc which cells are grid cells depending on their tuning on the toroidal manifold. This is what was done for example with head-direction cells (Chaudhuri et al., 2019). If this analysis does not work, the authors should provide some explanation why.

Next, the authors provide evidence of the internal nature of toroidal topology (and thus, of grid firing) by investigating the topology of population activity during sleep. This echoes previous findings from the same lab (Gardner et al., 2019) showing that pairwise coordination between grid cells is preserved across brain states. Here, 5/6 modules show toroidal topology during REM and 4/6 during SWS. One rat (S) did not show any organization during sleep. To address this point the authors suggest that the grid cells in rat S (and in module R1) are not of the same kind as the others: they are characterized by different theta modulation and ISI. To be honest, the manuscript becomes extremely unclear at this point. The authors should consider changing the presentation of these classes, and perhaps just focus on REM sleep.

If the authors decide to present the grid cell class analysis, they must address the following concerns. First, they did not record actual LFP, and theta phase and amplitude are obtained from population activity. But then, the problem becomes circular: if different cell types show different phase preference to theta, how come the theta phase extracted from the population is not itself biased by the relative ratio of grid classes recorded in a given session? This conundrum seems almost impossible to solve. The authors should adopt a more rigorous approach: quantify theta modulation strength from spike train autocorrelograms and forget about the absolute phase. Combined with ISI, this should be sufficient to cluster the data in different classes. Second, the authors should reconcile these findings with their previous report of preserved correlations during sleep. Importantly, are grid cells in R1 and rat S coordinated and, for some reasons, do not show a toroidal structure during sleep? Third, the authors did not mention or analyze the effect of the slow oscillation on population activity. This can well be an explanation of why some modules do not show a toroidal structure during SWS (as the fluctuation between DOWN and UP states introduces a non-specific coordination, biasing pairwise correlations towards positive values). The same authors had

adopted an interesting approach in their previous paper to address this problem, it is unfortunate they did not analyze SWS data more rigorously in the present manuscript. Finally, the authors mention that the different classes of grid cells showed different modulation with micro-arousal. This is potentially interesting but quite disconnected from the rest of the manuscript. I suggest to either fully characterize this phenomenon or to get rid of this observation.

Minor concerns:

What is the exact measure of spatial information? "Bits/s" is confusing, is it bit per spike or bit per second? Methods suggest that this is bit per spike (which it should be).

The authors should indicate in which brain state the correlation matrix in 5b was computed.

REF #3

This paper is elegantly done and interesting. The observation of toroidal topology in grid cell responses is important and of interest to a wide audience. The number of recordings is impressive and the analysis skillful. My only technical comment is that, unless I missed it, the use of 6 PCs does not seem to be well justified and the effect of changing this number is not well investigated.

I do have a broader comment. The paper starts in the abstract with a bold statement that grid cells are explained by CAN dynamics. This is exciting, but I found that as I read the paper I got progressively more confused about what actual message is being conveyed. First, grid cells are only a small fraction of the recorded neurons, and it has been claimed that there is no real distinction between grid cells and other MEC neurons. In other words, the cells being discussed here are the tail of a continuous distribution. Whether or not the authors agree with this, they are certainly talking about a small set of selected cells. Then, to the authors' credit, they point to a gradation in the evidence for toroidal topology across modules. This leads to the conclusion that more bursty grid cells are the most CAN like. The last line of the paper points out that cells can be grid-like even without showing CAN dynamics. This sentence is a far cry from the ending of the abstract. So what exactly IS the message: a subset of grid cells which are a small subset of MEC cells show CAN dynamics?

As I said, the authors should be praised for raising the issues commented upon in the previous paragraph. I think the paper would be improved with some clarity along the lines discussed above, but also with some ideas about how cells sitting next to each other can show such a high range of CAN dynamics. Are the CAN-like cells particular cells that are more strongly coupled to the network than other MEC cells? Is the only correlate of CAN dynamics other than topology burstiness? I guess what I am looking for here, and what I think the readers would greatly benefit from, is a discussion of the WHOLE circuit, how a CAN is embedded in it, and what the difference is between a toroidal, CAN-like grid cell, a non-CAN-like grid cell and a non-grid MEC cell. A lot to ask, I appreciate, but it would definitely help clarify the broader point of the paper.

REF #4

An influential class of network models suggest that grid cells form a continuous attractor network, characterized by a toroidal topology. These models explain many features of grid cell activity, but their core aspect – i.e. the toroidal topology of grid cells activity on the population level, was never tested experimentally. In this work, Gardner et al. were able to record a large number of grid cells simultaneously, by conducted high-density electrophysiological recordings using Neuropixels probes in the medial entorhinal cortex of rats. By employing advanced topological analysis of neural activity, the authors demonstrated that the collective activity of grid cells during navigation in open field arenas has a toroidal topology, as predicted by continuous attractor models. The authors did

not stop there, but also measured grid cell activity in physical environments, where the grid code is distorted by the environmental boundaries, and during sleep, when neuronal activity is decoupled from the positional cues. Under all these conditions, grid cells preserved their toroidal topology, which suggests that the toroidal manifold is derived from the intrinsic network organization of grid cells in the entorhinal cortex. Taken together, these findings represent a major evidence in favor of the continuous attractor model of grid cells.

I found these results extremely important and novel. Gardner et al. very elegantly utilized the power of population recordings to extract the internal organizational principle of grid cell network activity. My main comments/questions pertain to the dynamics of the “bump of activity” on the toroidal manifold that the authors found. I believe that adding the analyses, which I outlined in my comments below, will make the paper even stronger. Overall, I think that this beautiful work represents a major contribution to the field of neuroscience in general, and I strongly recommend this article for publication.

Major Comments

1. Most grid cells have single fields when plotted on the inferred torus, which implies a single bump of activity, as predicted by the continuous attractor models. However, some cells in Extended Data Figure 9 do show multiple fields on the torus (and these fields do not seem to be just trivial repetitions of a single field because of the boundary conditions).

I think it would be important to test explicitly for the existence of a single bump on the population level and to characterize its profile on the manifold (across the different environments and behavioral states). Could it be that in addition to the main bump, there are smaller bumps away from the main bump, which might explain why some cells may appear to have more than one field on the torus?

What is the profile of the bump? Is it symmetric? Does it become skewed along the movement direction?

2. What are the dynamics of the bump during sleep? Does it move smoothly on the toroidal manifold, or are there moments when it “jumps” abruptly? Is there any evidence for more than one bump appearing during sleep? Is the speed of movement of the bump comparable to that during running?

3. Does the bump move coherently on the different manifolds corresponding to different modules across behavioral states, especially during sleep? I expect the answer would be that it moves coherently during running, but independently during sleep (given the previous measurements of pairwise correlations in Gardner et al 2019).

4. During running, the bump is expected to move more slowly on manifolds representing modules with larger grid spacing, compared to modules with smaller grid spacing. Is it indeed the case? Does this expected negative relationship between average bump speed and grid spacing exist during sleep? It would be interesting to test if such putative speed-spacing relationship can be observed even if the bump does not move coherently across modules during sleep (see my previous comment). In other words, does the average speed of the bump on the manifold is a characteristic property of the module across behavioral states?

5. What happens to the bump during moments of immobility, which the authors excluded from the analysis? Does it become pinned on the torus or does it start to drift/disappear?

6. What is the vector field on the toroidal manifold and in its vicinity? Does it appear to be as expected from CAN models, for example is it identical in all directions along the toroidal surface?

A related question: is the toroidal manifold an attractive manifold? In other words, does the activity tend to be pulled towards the manifold on those instances when it deviates from it (as in Fig.3 of

Chaudhuri et al 2019), especially during sleep? I realize that there might be some circularity in this question: unless this was the case, the authors would not be able to detect the toroidal structure in the first place. But perhaps by inferring the torus using half of the data, the authors can test if the manifold is attractive using the remaining of the data.

7. The authors discarded grid cells with prominent head-direction tuning from the analyses. Head-direction cells are expected to have a ring topology, and therefore conjunctive grid X head-direction cells may form a 3-dimensional torus. I am curious if the authors have enough cells/coverage to see any evidence for such 3-dimensional torus?

8. Could the heterogeneity of grid cells reported in Figure 5 merely reflect the fact that the activity of some grid cells is modulated by other signals, in addition to grid tuning? I am saying that because if the authors used running epochs to compute the correlation matrix in Figure 5b (the information about what epochs were used for computing correlations is missing from the methods), then pairwise correlations should be low for most grid cells, except for those that have the same phase. Therefore, grid cells are not expected to be grouped into an apparent cluster, unless their firing is modulated by other factors. Indeed, C1 cells do not seem to represent a clearly defined cluster (but rather an “absence of a cluster”) – as would be expected from pure grid cells. In contrast, the much more pronounced C2 and C3 clusters suggest that cells in these clusters are modulated by other factors, which are not related to grid firing (e.g., theta phase). Accordingly, the spatial information in physical space for cells in clusters C2 and C3 appears to be lower than in cluster C1 (Fig. 5d).

It is thus might be expected that cells, which are modulated by other factors (in addition to grid tuning), would contribute “noise” (with respect to the grid signal) and thus reduce the chance to reveal the toroidal tuning. It is probably the reason why the authors excluded conjunctive grid X head-direction cells from the topological analysis. Therefore, I wonder, if the clustering analysis/theta modulations reported in Figure 4, could be replaced by a measurement that quantifies directly how much of the firing rate of the cells is explained by grid-like tuning, and whether such measurement will be a more direct criterion for the ability of cells to reveal the toroidal structure.

Minor Comments

1. It might be good to include 1-2 sentences in the main text to explain to the non-expert reader why topological analyses were done by first separating grid cells into modules.

2. In some cases, the modules that result from the clustering algorithm (Extended Data Figure 2) seem to be very broad in terms of grid spacing and orientation spanned by cells that belong to a given module. For example, cells in what is identified as module “R3” (Extended Data Figure 2c) have grid spacings ranging from ~40 to 140 cm and orientations spanning the entire range of $\pm\pi$. This might suggest that this module was not extracted correctly by the clustering algorithm. Also, in some modules (e.g., Extended Data Figure 2b “R3”, and Extended Data Figure 2c “R2”) there was a considerable number of cells with relatively low grid score (their grid score values were comparable to that of the “main” cloud). The authors may consider applying a subsequent criterion (e.g., a grid score threshold) to “denoise” the modules.

3. The authors reduced the dimensionality of the data by taking the first 6 PCs. By analogy to unimodal circular tuning, if the tuning is wide (cosine tuning) one needs 2 PCs (sine and cosine) to generate all tuning curves with different angular offsets. However, more PCs are needed when the tuning gets narrower. The authors make a related comment in Supplementary Math Note: “If the tuning curve of individual cells is sufficiently wide ... the modes that correspond to the first six PCA components are expected to capture a large fraction of the variance”.

By extending this argument to grid cells from different modules, I would expect that more PCs are needed to explain the variance of grid cells from modules with smaller grid-spacing, because their tuning is narrower, compared to grid cells from modules with larger grid-spacing. Also, the variance explained by these 6 PCs is rather modest (together they explain only 25% of the variance). While the results presented in the paper are very convincing, I am wondering if the toroidal manifold might be even cleaner if more than 6 PCs are used, or if the number of PCs used for analyses, should be adjusted depending on the spacing of each module.

4. Fig. 1c-d: The activity of example cells appears on the same positions on the torus. Was the toroidal representation rotated to achieve this match? Or does it mean that there was no remapping between OF and WW environments?

5. I found the section “Cohomological decoding” in the Methods quite hard to follow. Since this is an important part of the analysis, some additional clarifications might be needed for the non-expert reader.

Also, it might help to illustrate the “Rhomboidal fitting and toroidal alignment” section in the Methods with a Supplementary Figure showing the results of the fitting procedure.

6. Fig. 3c: Can the fact that grid cells have higher spatial information in toroidal space compared to physical space, be simply because in toroidal space grid cells have a single field, whereas in physical space they have multiple fields?

Author Rebuttals to Initial Comments:

Referee #1 (Remarks to the Author):

Ever since the discovery of grid cells, it has been assumed that the population activity of grid cells having the same grid spacing and orientation should display a torus-like structure. The reason for this is simply that the hexagonal lattices associated to grid cell modules periodically identify disconnected regions in the plane, and the quotient space is a (hexagonal) torus. Seeing this structure in the data was just a matter of time, as it required high-density recordings (to get enough cells in a given module) as well as fast computational topology methods (to verify the topological structure within a high-dimensional space). So it is in some sense not surprising to see a verification of toroidal topology in grid cell population activity. It had to be there, simply because of the structure of grid fields.

Nevertheless, this paper is highly significant and goes well beyond a simple verification that there's a torus in the neural activity. The authors have collected a stunningly high-quality data set that allows them to detect and visualize the torus in multiple (six!) grid modules with a large number of cells (~150) per module. They have also tested for toroidal structure in environments where the hexagonal structure of grid fields is distorted, and in sleep (SWS and REM). In the wagon-wheel experiments, neurons were better tuned to the torus than to physical space, even when the torus was inferred using the other (open field) environment. The analysis is very convincing and suggests that the torus-like manifold emerges from the intrinsic network organization, and is not inherited from sensory inputs. This is about as thorough of an investigation as one can expect on an extremely interesting and important topic. I thus believe this study warrants publication in Nature. Moreover, the paper is well-written and the figures are polished and clear.

The manuscript did raise several questions that I hope can be addressed in a revision.

1. Why is there so much emphasis on the CAN? Continuous attractor networks are not the only model of recurrent network organization that could produce the torus structure. The authors seem to be contrasting CANs only to feedforward mechanisms, such as in this sentence of the discussion: “While toroidal topology can be generated by both CAN1,13–16 and feedforward17–22 mechanisms, the

persistence of an invariant toroidal manifold across environments and behavioural states, under conditions that give rise to changes in the correlation structure of place-cell activity in the hippocampus^{11,12}, is predicted only by CAN models.” But CAN models are not general recurrent networks: they have a very precise connectivity structure and a posit a continuum of fixed point attractors. There are other ways a recurrent network could be wired so that the torus emerges, where the connectivity is not fine-tuned and purely geometric as in CANs, and the activity is not given by fixed point attractors. None of this is ruled out by the data.

The author’s findings are certainly consistent with CANs, but I do not believe they provide direct evidence for them. What they find evidence for is the strong role of recurrent – as opposed to feedforward – mechanisms governing grid cell activity. The first paragraph of the Discussion is a perfectly accurate summary of their findings, and they are very significant. There is no need to overlay the CAN spin, and I find this to be the weakest aspect of the paper.

We thank the Reviewer for raising this point, which identifies a need for clarification from our side. Although some of us have worked on CANs for more than a decade, we were not able to guess exactly what kind of non-CAN recurrent network the Reviewer had in mind when suggesting that the manuscript does not rule out non-CAN recurrent network models of the grid pattern. We are wondering if we are using different definitions of a CAN. What we mean by CAN models in the present paper is a network where recurrent connectivity is structured such that the network activity patterns are strongly attracted to a continuous, low-dimensional stable manifold that is invariant under a wide range of external inputs. Activity can drift around on this attractor manifold but is largely constrained to the manifold. Mathematically, this is all that a CAN means to us. When used in this sense, the term *does not imply a specific geometric form of connectivity* of the kinds that have been proposed in CAN models of grid cell models (such as Mexican hat connectivity), other than that the network is recurrently connected.

There are several examples of CANs whose connectivity does not have simple geometric structure: this is the case, for example, in CANs proposed for oculomotor integration¹, and in recurrent networks that were trained to perform evidence accumulation tasks and where CAN dynamics developed in an unsupervised manner during learning². These and related³ networks started out with random connections but during learning, synaptic weights were gradually adjusted to produce a continuous attractor. In all of the examples, connectivity was not strictly geometrically organized, though mathematically the networks are still CANs because the connectivity keeps the activity on the low-dimensional manifold. Along the same lines, for grid cells, a recent preprint from the Ganguli group⁴ demonstrates that neural networks trained to perform spatial path integration robustly develop activity patterns that lie on a toroidal manifold. The underlying recurrent connectivity that emerges in these networks is not nearly as geometrically structured as in the typical “engineered” networks of early CAN models for grid cells. Still, these networks are CANs, because their recurrent connectivity constrains activity patterns to lie on the toroidal manifold.

We infer that when speaking about recurrent networks that are not CANs, the Reviewer refers to the more non-symmetrical non-engineered types of networks in the models above. If so, we may have been using the concept of CANs more broadly than the Reviewer. In the Introduction of the revised manuscript, we have therefore tried to prevent this type of confusion by now spelling out from the onset (second sentence) how we define a CAN. We also state explicitly (in the third sentence) that CANs can emerge from a variety of connection architectures, from highly structured to random (with subsequent learning). Alternative CAN geometries are cited¹⁻⁴. In addition, we address the issue of alternative recurrent networks at the end of the final paragraph of the Discussion, where we relate the findings of toroidal topology to CAN models of grid cells. We emphasize now that the current data, although consistent with predictions from CAN models for grid cells, do not imply any particular form of geometric connectivity but may, in principle, be implemented by any form of recurrent connectivity that keeps the activity bump on a low-dimensional toroidal manifold, including implementations that involve learning by adjustment of synaptic weights^{3,4}.

Finally, we have checked the entire text and made sure that no statement inadvertently implies any subclass of CAN model. We have chosen to continue to interpret the data within a (now broader) CAN framework, given that the experiments were motivated by, and validate, central non-trivial predictions of CAN models for grid cells (such as the persistence of toroidal topology in sleep). Because no other theory makes similar predictions, to our knowledge, the findings go far in identifying the set of mechanisms that might account for the grid pattern. The final paragraph of the Discussion is dedicated to this question, but we have rephrased it to be clearer about the limitations of the study, i.e. while central predictions of CAN networks are validated, the geometry of the connectivity cannot be inferred from the present data. Functional connectivity experiments and

analyses will for sure address this next step in the years to come.

2. Why do the authors use persistent cohomology as opposed to persistent homology?

Persistent homology is far more standard in topological data analysis (TDA), and is the main computation done by packages like Ripser (which the authors use). Moreover, the entire description of the topological analysis in the main text is identical to what I would have expected if persistent homology were being used. So are the results. So why the choice to use cohomology?

We believe this statement is incorrect. The standard computation of the Ripser package uses persistent cohomology, which gives rise to representative cocycles. Persistent homology (and correspondingly, representative cycles) were added as an option in an experimental Github repository branch (<https://github.com/Ripser/ripser/tree/representative-cycles>), but to our knowledge, the computation of persistent cohomology as used in Ripser, and as first proposed by De Silva et al⁵, is currently the fastest.

Cohomology is specifically chosen in our study in order to obtain representative cocycles of the cohomology classes, which in turn allow activity to be mapped onto the torus (decoding analyses). The principle behind the decoding procedure is the observation that for a topological space, each one-dimensional integer-valued cohomology class (H^1 bars) corresponds to an equivalence class of continuous maps from the space to the circle S^1 - equivalence given if the maps are *homotopic* (i.e. one map can be deformed continuously into the other⁶). This motivates the derivation of two representative circle-valued maps of the two longest-lived persistent cohomology classes, which in combination results in toroidal coordinates of the dataset. Though attempts have been made to find parametrizations of the dataset based on homology, such as localization and homological coordinatization⁷, these have different caveats, and a similar theorem connecting cycles to circle-valued maps does not exist for homology. Because of its suitability for decoding of toroidal position, persistent cohomology and the circular coordinatization approach is thus a natural choice for the present purposes.

Finally, when using a field coefficient (such as \mathbb{Z}_{47} , or any other prime), homology and cohomology barcodes are indeed identical and thus give the same results (hence the terms are often used interchangeably, “homology” commonly used as the interpretation is more intuitive).

To clarify these issues, we have added a statement in the Online Methods (in the “Persistent cohomology” section) where we in the first paragraph explain why cohomology was chosen instead of homology.

The only hint I got was some mention of the cup product in the Supplementary Methods as an additional way to verify the torus structure (since other topological spaces are consistent with the same bar codes). But this is never mentioned in the main text and is a rather unsatisfying afterthought in the Supplementary Methods, especially since the text leading up to it is a standard discussion of persistent homology.

We agree that the reference to the cup product in the Supplementary Methods, as presented, comes as an add-on at the end of the supplementary text and does not contribute a lot to the interpretation. Because further analysis of the cup product would be computationally extremely demanding and further theoretical advances are needed (though we note recent work by Contessoto et al on an invariant called the *persistent cup-length*⁸), we have chosen to remove the section on the cup product. The rationale for using cohomology is better explained by the suitability for decoding of toroidal position.

An additional minor point: the choice of coefficients (Z_{47}) should be justified. I understand why Z_2 wouldn't be good, but why the prime $p = 47$ and not something else? If it's just an arbitrary choice, the authors should say so.

The choice of $p = 47$ is arbitrary; it must be a prime to give field coefficients and must be a number that does not divide the torsion subgroup of the homology of the space to assert the correct Betti numbers (as extra bars equal to the dimension of the torsion subgroup divisible by 47 – the 47-torsion - would in that case be present in the barcode). We chose 47 because the underlying space is unlikely to have 47-torsion. We obtained similar barcodes with a different coefficient ($p = 43$), which we have tested (Rebuttal Fig. 1). A statement about the choice of coefficient and the results given different primes, has been added to the Persistent cohomology section (third paragraph).

Rebuttal fig. 1

Example of barcode calculated using different coefficients - Z_{43} (left) and Z_{47} (right, as previously found in Fig. 1) - in computing the persistent cohomology of grid cells from R3 during OF. Note that all bars are preserved, showing identical results regardless of choice of coefficients and thus excluding the possibility of 47-torsion in the cohomology (see "Persistent cohomology" Online Methods).

3. The torus visualization uses PCA to get down to 6 dimensions, and UMAP to further reduce to 3-d. Of course, the whole point of TDA methods is to allow topological structure in point cloud data to be identified in high-dimensional data, where it can not be seen by eye. The authors thus perform TDA on the 6-dim'l data after only the PCA reduction. This makes me wonder: why not perform TDA on the original N-dim'l data? Does it not work there? If so, it'd be good to know and also to have some discussion of what goes wrong that makes dimensional reduction a required pre-processing step before performing topological analyses.

PCA preprocessing was added in order to de-noise the data, i.e. to reduce the influence of noisy fluctuations in neuronal firing rates. We show in Extended Data Fig. 3 (b, c and d) that the population structure is reliably accounted for by the 6 first principal components, which all have a grid-like representation in the open field, unlike the higher principal components (which may be related to

‘other’ correlates of the activity, such as theta modulation or head-direction influences). In order to extract the dominant features of the data, in particular those related to the grid structure of the cells, we chose to perform the further topological analysis on those components, while we reduce the influence of ‘other’ contributions that may not be identifiable with the current size of data.

We understand the concern that the choice of 6 PCs seems a bit arbitrary. In the original version of Extended Data Fig. 3, we showed explained variance for a single example recording (panel b). In retrospect, we realize that the example may have appeared hand-picked. In the revised version, we therefore show explained-variance curves for all modules during OF, REM and SWS (Extended Data Fig. 3b). Moreover, for each module, we show the PCs’ mapping onto the environment (Extended Data Fig. 3c), and we quantify the spatial selectivity of this mapping by fitting a GLM model to each principal component using the spatial coordinates as covariates, obtaining explained deviance scores for each map (Extended Data Fig. 3d). It can be seen from these new figures that in all modules the first 6 components stand out, with a sharp drop in both explained variance and explained deviance from the 6th to the 7th PC. This shift is particularly strong in the OF, but visible across all behaviour conditions. The 6 first components show grid-like patterns in the open field, while higher components typically do not (Extended Data Fig. 3c). We hope that our decision to focus on the first 6 PCs appears less arbitrary now that we present the full set of data with quantitative analysis. In addition to showing the data, we have further explained in the legend and in the Online Methods the rationale for choosing the first 6 components and discarding the higher ones.

In addition, we provide evidence for the choice of 6 components by performing, for one module, an analysis where we kept from 3 to N principal components and plotted the normalized lifetime of the three longest-lived \bar{v} bars in the barcode. The lifetime of the two longest-lived \bar{v} bars (expected to be long for a torus) was divided by the lifetime of the third longest-lived \bar{v} bar (expected to be short) as a function of number of principal components. A sharp peak in this ratio was seen with 6 components. We have added a panel in Extended Data Fig. 3e in which this is shown.

While we feel these additional analyses justify the choice of the first 6 PC components for subsequent processing, a valid question is, however, whether toroidal topology could be identified without any of these de-noising steps. In asking so, we face the well-known problem in statistical learning known as the *curse of dimensionality*. This term (first coined by Bellmann^{9,10}) refers to the exponential increase in volume associated with adding extra dimensions to Euclidean space. For example, 100 evenly-spaced sample points suffice to sample a unit interval with 0.01 distance between points; an equivalent sampling of a 10-dimensional unit hypercube with a grid with spacing of 0.01 between adjacent points would require 10^{20} sample points¹¹. Similarly, the number of points needed to faithfully sample the torus in high-dimensional space quickly increases with the number of units, and for computation reasons, we are severely limited in the number of points from which we can compute barcodes. Thus, reducing the number of dimensions allows us to sample the space with a lower number of points, enabling persistent (co)homology computation.

Furthermore, the curse of dimensionality relates to how proximity in high-dimensional data may not be qualitatively meaningful, as the ratio of distances to a point’s nearest neighbor over its farthest neighbor approaches 1 for high-dimensional Euclidean spaces, i.e., all points become approximately equidistant from each other¹². Thus, meaningful dissimilarities between points – e.g. the geodesic distance between points on the underlying manifold – and dissimilarities caused by small, noisy fluctuations in the firing rates become indistinguishable. Reducing the number of dimensions may reduce the influence of noise when computing nearest neighbours for each point, as is done in the preprocessing step succeeding PCA¹³.

We have added a statement in the Online Methods to explain how the preprocessing procedure allows us to circumvent the curse of dimensionality (“Preprocessing of population activity for topological analyses” section).

4. Is there any relation between bar length and grid field spacing? I found myself wondering if the bar codes carried information about which module the activity came from. I also wonder if there is any systematic asymmetry in the lengths of the two beta1 bars. The grid field lattices are hexagonal with a fairly regular hexagon, as far as I understand. This should predict something about the relative sizes of the two one-dimensional holes in the torus. Is the data consistent with this prediction?

The barcodes express the lifetimes of holes in dimension 0, 1 and 2. While we did not expect the scale of lifetimes to reflect grid spacing, we have taken the Reviewer’s advice and performed the analysis for the open-field data. Note that the proposed analysis requires a normalization. We chose to divide the bar lengths with reference to the mean of the lifetimes of the corresponding bars in the shuffled distribution. There was no significant correlation between mean spacing of grid modules and the normalized bar length in either dimension (Rebuttal Fig. 2a-d).

Persistence lifetime of the two longest 1 bars (left) and the longest 2 bar, as a function of grid module scale. Each dot represents bars of one grid module. Grid scale was calculated as shown in External Data Fig. 4. The absence of correlation shows that barcode lifetime is unrelated to grid module scale. T- and P-values were, for the linear model of the grid module scale relationship with: a) the mean of the lifetimes of the two most prominent 1 bars the lifetimes, T = 0.75 and P = 0.46; b) the longest-lived 2 bar T = -0.34 and P = 0.73; c-d) each 1 separately T = 0.42 and P = 0.68, and T = 1.05 and P = 0.31; and e) the relative difference in lifetimes of the two longest-lived 1 bars, T = 1.2 and P = 0.24.

Note that we obtain geometric relationships corresponding to the circular features captured by each beta_1-bar through the cohomological decoding analysis (see table in Extended Data Fig. 4d). These analyses suggest similar size of the one-dimensional holes in the torus and we do not see a systematic asymmetry in their relative bar length difference (Rebuttal Fig. 2e).

The normalized bar lengths did, however, scale differently with increasing size of the temporal smoothing used to determine the firing rates (Rebuttal Fig. 3). Thus, modules of larger grid field spacing could be identified as those maintaining long bars at longer temporal scales but at the temporal scale used in this work, there is no clear relationship.

Rebuttal fig. 3

Effect of varying temporal smoothing on the barcode. Barcodes were calculated after applying Gaussian temporal smoothing kernels of different widths to the binned spike trains. Each line indicates the normalized lifetime of the longest- or second-longest-lived H1 for a module, as a function of the Gaussian sigma parameter (temporal smoothing width). H1 lifetimes were normalized by dividing by the median H1 lifetime from the same barcode. Note the peak H1 lifetime occurs at ~100 ms for all modules; however, for greater amounts of temporal smoothing the H1s live longer for increasing module spacing (ordered R1 < R2 < R3), as seen between 250 ms and 400 ms for all three modules, and 400 ms and 850 ms for R3 versus R1 and R2.

5. Reference 85 (Giusti et. al.) also saw consistent topological structure between open field and sleep data (in place cells). Do the authors believe that place fields are also a CAN? If not, how do they explain the evidence for recurrent features dominating over feedforward ones in that context? (This is related to my first question. There are other types of recurrent network that are not CANs...)

Using a pairwise correlation approach, Giusti et al. inferred topological structure in place cells from CA1 of the hippocampus. While there was no hint at toroidal topology in these data, the correlation structure exhibited similarities between open field foraging, wheel running and REM sleep, pointing possibly to some sort of preserved structure, which then could be consistent with continuous attractor dynamics on a low-dimensional manifold. However, as acknowledged by the authors, the data are difficult to compare across conditions and the lack of direct quantification of the topology prohibits strong conclusions regarding the dimensionality of the data and thus whether the network is a CAN or not.

CAN dynamics, if present in place cells, would be facilitated by recurrent network structure, which is known to be present in the neighbouring region of CA3 (or could be inherited through inputs from MEC). However, in this study, we prefer to stay away from an in-depth discussion of the relationship of place cells to network connectivity since it is only tangential to the question of the topology of grid cell activity and we expect to be asked to cut down on the length of the manuscript if it is accepted. While Giusti et al demonstrate that low-dimensional structure may be present also in sleep, the evidence for preservation of the structure across states and tasks (awake-sleep, open field – wheel running) is hard to evaluate, given the scarcity of data and analyses (one paragraph and one simple figure). Since place cells are known from hundreds of studies to remap between environments and tasks, we feel it would not be justified at this time to conclude that place cells are part of a CAN and that they operate on a low-dimensional manifold, although we cannot also rule out this possibility either.

We now address the Giusti study by pointing out, briefly at the end of the revised Discussion that CANs for grid cells may co-exist with other CANs in the hippocampal-entorhinal network, e.g. for head direction cells and place cells, and that understanding spatial coding ultimately requires investigation of the mechanisms by which those networks are linked and interact. The Giusti paper on place cells is now cited in this multi-CAN context.

Referee #2 (Remarks to the Author):

In the present manuscript, Gardner et al. have studied the organization of grid cell populations in the medial entorhinal cortex. Specifically, they used topological data analysis techniques to demonstrate that grid cell population activity has a toroidal topology. Although this is not the first time that these techniques are used to analyze neuronal population activity (e.g. for head-direction cells), applying these technique to grid cells requires the monitoring of large ensembles of neurons, which had never been achieved so far. To this end, the authors used Neuropixel probes, reporting up to 150 neurons within a single “module” of grid cells. The authors further demonstrate that the topological structure of population activity is preserved across environments and brain states, suggesting that the system is governed by local circuits and dynamics. While these findings are exciting and constitute an important milestone in our understanding of neuronal activity in vivo, the present manuscript suffers from a number of limitations and inconsistencies that need to be addressed. Overall, the dataset is quite inhomogeneous, raising concerns about the validity of some of the claims.

While the authors report a stunning recording of 149 grid cells from the same module in one rat, the authors should provide more details regarding the dataset. First, the authors presented the same histology sections for the left and right hemispheres of rat Q, and again the same histology sections for shanks 2 and 3 of rat S. While this is certainly an honest mistake, this needs to be corrected.

We thank the Reviewer for spotting the duplication mistake! The correct brain sections have now been provided for Rat Q (right) and Rat S (left, shank 3) in Extended Data Fig. 1.

Furthermore, the sections should be presented without the probe overlaying to clearly show the electrode tracks.

We agree and are now showing brain sections without overlying probes, as requested. Instead of superimposing the probes we have added pairs of arrowheads on the clean brain sections to indicate the range of active recording sites (corresponding to the yellow region of the superimposed probes in the previous version).

Next, it is interesting to see how grid cells are anatomically distributed, but it would be even more interesting to show how this compares to all other recorded neurons. The authors should report the total number of cells along the electrodes, and at least the proportion of “pure” grid cells (the only one considered for the topological analyses), conjunctive grid cells, and HD cells.

In the original manuscript, the number of “pure” (not conjunctive) grid cells was presented for each module within insets in each panel of Extended Data Fig. 2, as was the number of cells recorded on all active sites. We have now instead produced two tables, inserted in Extended Data Fig. 2, in which we, for each grid module (panel g) and for each recording session (panel h), show the number of (pure) grid cells, and the number of conjunctive grid × head direction cells. For each module we also show the division of cells between hemispheres. For Rat S (with 4 shanks), we further show a breakdown of cells on individual shanks. Head direction cells are not shown in the table because these were not identified or analysed (not part of this study); we only looked for head direction selectivity among those cells that were already identified as grid cells, in order to identify

conjunctive grid × head direction cells. References to the tables with cell numbers are provided in the main text (first results paragraph).

We have chosen not to add plots for the distribution of each cell type along the track of every electrode in every animal (we assume this is what the Reviewer wanted to see in the request for number of cells ‘along the electrodes’). This would have added a total of 24 distribution plots for the 8 recording shanks (adding, for each shank, 1: the distribution of grid cells, 2: the distribution of conjunctive grid × head direction cells, and 3: the distribution of all cells). There would not be space in Extended Data Fig. 1 for these plots. Information on the intra-regional anatomical distribution (in MEC and PaS) of head direction cells and conjunctive grid × head direction cells would provide little insight into the core questions of this study, but we are happy to add these 24 plots if the Editors can advise on how to prioritize when reducing the size of display items in the next round.

We have also chosen not to provide numbers of cells per anatomical region (PaS or MEC), given that anatomical boundaries in the MEC-PaS-PrS-PoS region are not sharp and cell signals may cross the borders. For the present paper, where location of the grid cells is secondary, it may be better to consider MEC and the adjacent PaS as a single functional area.

The number of cells (of each type) per hemisphere is available from the new table in Extended Data Fig. 2g.

Overall, it would be interesting to present a table that precisely describes the content of each module. Presently, this information is difficult to access. For example, the number of neurons per module can be found in Ext. Data Fig 2e in a very small panel. By the way, this panel reveals another inconsistency, as it is stated that module R2 has 168 neurons, while in Fig. 1, it is said that it contains 149 neurons.

We have followed the Reviewer’s advice and moved the content of the insets in Extended Data Fig. 2a-d to a table in panels g and h of the same display item (the same table as referred in response to the previous comment; please see above). We have also added a statement on the number of grid cells (lowest-to-highest) in the main text (first paragraph of “visualization of toroidal manifold”), along with a reference to the table for the details.

We thank the Reviewer for pointing out the apparent discrepancy in numbers of cells in module R2 in Extended Data Fig. 2. The discrepancy is explained by the fact that panel e included both pure and conjunctive grid cells, whereas panels a-d showed only pure grid cells. In the revised version, both of the new tables separate pure and conjunctive grid cells. In the main text (end of first paragraph in

“Visualization...”, we now explicitly point out that the numbers reported there are for pure and conjunctive cells altogether.

Along the same lines, the authors should clarify whether neurons were also recorded from the prosubiculum (PoR, the result section in the main text only refers to the MEC and parasubiculum). Although only a minority of cells in the dataset seem to be located in the PoR, this should be stated and/or discussed. In rat R (L), it is unclear whether grid cells of the module R2 were recorded exactly at the same location of the module R1 (color is confusing).

We assume that with prosubiculum the Reviewer means postrhinal cortex (PoR). Indeed the number of grid cells in PoR is very small and most of these are recorded near the border to MEC and parasubiculum, i.e. we are not confident that all or most of these cells have soma in PoR. It is important to keep in mind here that the boundaries and recording locations of Neuropixels (or classical tetrode) studies are not exact. We wish to emphasize three factors. First, anatomical boundaries cannot be overlaid blindly on brain sections; they may depend on factors such as the shape and orientation of the brain circuit compared to the orientation of the section (boundaries may be different on neighbouring sections, which are not shown). Second, we do not have a method to precisely align the probe recording locations with the histology, and, third, tetrodes, and by implication Neuropixels probes, are thought to pick up signals from neurons within a radius of at least 50 μm away from the recording site¹⁴. It is thus possible that signals from cells with soma in MEC or PaS show up on recording sites across the border in PoR.

While grid cells in PoR may largely be explained by their closeness to MEC/PaS, as explained above, there were 6 PoR grid cells in Rat R (right hemisphere) that were quite far away from the border. These cells were all from the grid module with the largest scale, which is most prone to erroneous classification because of the small number of fields. Upon closer inspection, we found that these 6 cells had highly irregular firing fields, looking quite different from the average cell recorded in MEC and PaS. We also found that these cells passed the classification as grid cells only in one of the two recording sessions conducted for this rat. Because this was the shorter of the two sessions (total time spent moving 21 minutes, compared with 128 minutes in the longer session), the risk of misclassification is larger. Thus, in the revised figure, we have chosen to show the cell distributions from the other session (the longer one). Here, with more data, the classification is more reliable and grid cells do not appear centrally in PoR. The location of active sites was identical to that of the other day. For readers this solution will probably be less confusing, since the 6 cells were apparently misclassifications and might falsely suggest that PoR has grid cells. We thank the Reviewer for drawing our attention to this issue.

In addition to swapping recording sessions for Rat R, we have added a statement in the legend of Extended Data Fig. 1 reminding the reader that recording locations are approximate (for the reasons given above), and that signals may cross these boundaries¹⁴. Due to the uncertainty about anatomical borders and crossing of signals, we do not provide counts per region (MEC, PaS, PoR). However, the numbers of grid cells can be read out from the plots that show distributions of cells along the shank (with approximate regional boundaries included).

As for the color for module R1 in Rat R (left hemisphere), the problem was that the R1 bar is exactly superimposed on the R2 bar (same location). We are now using line shading (hatching) to show both bars in the same location; hopefully this resolves the confusion.

Modules were determined with an interesting clustering procedure. The authors should briefly discuss whether the separation of the R1 module in two sub-modules corresponds to differences in anatomical distribution of the neurons (PoR/MEC? Left/Right?).

Indeed R1 splits into 2 UMAP clusters (named R1a and R1b). We checked early on whether these submodules differ in location and found that they don't split between anatomical regions. Due to the approximate nature of the regional boundaries for this animal (the left hemisphere in particular), we chose not to include the data in the manuscript. But at the Reviewer's request we have now added the information about hemisphere locations in the table in Extended Data Fig. 2g (rows "R1a, day 1" and "R1b, day 1") and we have separated R1a and R1b in the plots in Extended Data Fig. 1 where we show distribution of cells along the track (R1a solid, R1b hatched). The two submodules split across hemispheres: left/right cell numbers are 66/68 for R1a and 4/28 for R1b. It can be seen that in the left hemisphere, R1a and R1b cells are split roughly evenly between PoR and MEC/PaS (on both sides close to the border, and with uncertainty about the exact location of the boundary, as noted above). In the right hemisphere, all R1a and R1b cells appear to be located in MEC.

Analyzing the population data from each module separately, the authors demonstrate with topological data analysis that each module shows a toroidal structure. This is the main claim of the paper and the result is unambiguous. However, it would be interesting to compare these results with an analysis including all grid cells, including conjunctive ones - and why not, all recorded cells.

Performing the bar-code analysis on all cells is unfortunately not within reach with the present amounts of data or the present data analysis methods. While it is interesting to analyse the structure of several modules of grid cells, it is not theoretically established what topology the multi-module activity constitutes; this is what motivated us to start out, in the present study, by studying single modules to affirm and understand the topology of single module activity. If we assume that combined grid module tori (grid cells of multiple modules, or conjunctive grid \times head direction cells) form a higher-dimensional torus, the number of grid cells needed to truly detect such structure would exceed that within the current dataset, as shown by Kang et al¹⁵ with simulated grid cell activity. As for analysis of other functionally modulated neurons (not grid cells), such analysis needs further theoretical development not within the scope of this paper. For instance, the use of PCA and number of dimensions kept will play a significant role in the determination of the manifold (as shown for the case of the torus). Fig. 5 already indicates the difficulty in showing specific topological structure, as the torus only appears in assemblies of grid cells with very similar functional properties (specifically the "bursty" class of pure grid cells); when grid cells with different spike-time characteristics but belonging to the same module were lumped together, the barcode no longer showed the characteristic toroidal (1,2,1) signature, and was difficult to decipher.

Despite these concerns, we have attempted to decipher, for the Reviewers, one example of multi-module grid-cell data (Rebuttal Fig. 4 Rat R, all pure grid cells from all 3 grid modules). The cells are the same as shown in the representative illustration in Fig. 1. In the case of dimension-reduction to 6 dimensions, the analyses seem to show toroidal structure akin to the module with the largest spacing (determined by comparing the cohomological decoding as a function of physical space). Whether this is because each module "latches onto" a single torus or if the dimensionality reduction only distils the information from this particular module is not easily determined. Keeping more components (e.g. 20, as in the figure) makes the determination of the topology more difficult, and no typical topological space comes to mind. Because these mixed-module analyses are hard to interpret, we have chosen, at this point, not to include the figure in the manuscript.

Combining the pure grid cells of R1, R2 and R3 with varying dimension-reduction and decoding of the two most persistent H1s. Note that with 6 dimensions, it seems to find a torus consistent with the spacing of R3, possibly because the toroidal pattern of the smaller-spaced modules revolves around that of the largest torus, but at a different speed (see Rebuttal Fig. 3) and thus, the largest spacing will explain the variance of the neural data of the smaller grid module as well, whose remaining variance is of higher-resolution explained by the higher principal components.

Because of these constraints, we have instead chosen to address the Reviewer's question by providing more information about the topological structure of activity in conjunctive grid \times head direction cells, which is somewhat more attainable. In the revised version of the paper, we have included the barcode during SWS for the theta-modulated class of grid module R1 and S1, which contains a large proportion of conjunctive grid \times head direction cells (Extended Data Fig. 7e-f). The barcode for this class of neurons can be seen to be clearly different from that of the pure grid cells, with a single prominent ¹bar exceeding the shuffled line, that when decoded, seems to match the recorded head direction (Extended Data Fig. 7g-h). The decoding of the longest-lived ¹bar clearly correlates with the head direction circle, but even for this population further interpretation of both barcodes and decoding is difficult as we would need more conjunctive populations with a larger number of neurons to provide a confident statement about the topology of the conjunctive grid cell network¹⁵.

In addition to showing an example barcode for the mixed data (pure and conjunctive cells) in Extended Data Fig. 7, as well as the multi-module data in Rebuttal Fig. 4 above, we have added a statement in the concluding sentence of the Introduction, where the topology predictions of the CAN models are laid out. We now emphasize here that toroidal topology is predicted for grid cells within the same module due to the requirement for shared geometric features among grid cells participating in the network, and that the study will focus on individual modules for this reason, as well as because addressing multi-module topology requires cell numbers well beyond those collected with the present approach.

In summary, the added complexity to the analysis and the requirement of larger neuronal populations and more animals for which multiple modules are recorded to identify more complex topological structure, discourages mixed-cell analyses. We hope that with the demonstration for mixed cells (examples above), and with the clarification of the theoretical predictions that motivated the study and its focus on individual modules, readers will agree that analyses should stay focused on single modules in the present paper.

Does it really distort the torus? One can argue that, by selecting pure grid cells alone, the result is obvious. The strength of such analysis would be to find a toroidal structure from the ensembles of neurons within one module (whose anatomical boundaries can be determined from the spatial properties of grid cells) and to infer post-hoc which cells are grid cells depending on their tuning on the toroidal manifold. This is what was done for example with head-direction cells (Chaudhuri et al., 2019). If this analysis does not work, the authors should provide some explanation why.

We are not quite sure we understand which result is obvious. Toroidal topology is indeed expected for pure grid cells in the open field environment, based on the spatial tuning curves of individual cells (still, it is good to see that the topological analysis is able to pick up the toroidal topology in an unsupervised manner). But the result is far from obvious when looking at different environments and in sleep. We showed not only that the topology is preserved, but also that the tuning of individual cells to the toroidal coordinates is preserved, pointing to a very rigid repertoire of population activity patterns across conditions. How these findings extend our knowledge is spelled out in the Discussion, but we are happy to elaborate further if the Reviewer can point further to what he or she thinks is obvious.

In the second part of the question, the Reviewer asks whether we can perform the cohomology analyses on cells from blocks of anatomical tissue (we hope this is a correct reading of the Reviewer comment). If toroidal topology could be inferred from the mixed cell population of a given MEC location, we could then next identify grid cells as those cells that map onto a discrete single location on the torus (although this was not the purpose of the study). The results with mixed grid modules in response to the previous comment, with accompanying clarifications in the manuscript, suggest, however, that this proposed analysis would not be successful. Throughout the MEC, tissue blocks would consist of mixed cell populations, including multiple grid modules (since grid modules generally overlap anatomically) as well as a variety of other cell types. Since a torus cannot be identified in grid cells from mixed modules (as shown above), toroidal topology would also not be expected to appear in blocks that contain an even wider range of cell types, and then tuning to the torus can of course not be determined either.

The Reviewer is right that the proposed approach did work for head-direction cells from the anterior thalamus (Chaudhuri et al). However, anterior thalamus is different from MEC in that, in some recordings, more than 50% of the cells are head direction cells (e.g. in the Peyrache-Buzsaki study¹⁶ that collected the data in Chaudhuri et al.). Moreover, these cells may not segregate into modules. MEC, in contrast, has anatomically interleaved cell populations, among which grid cells of a single module account for only a small percentage – too small for a torus to be identifiable in mixed-cell cohomology analyses with our sample sizes, as shown in Rebuttal Fig. 4.

We hope that the new analysis in Rebuttal Fig. 4 takes care of the question and that it is apparent from the new panel why an all-inclusive regional analysis would not identify a torus.

Next, the authors provide evidence of the internal nature of toroidal topology (and thus, of grid firing) by investigating the topology of population activity during sleep. This echoes previous findings from the same lab (Gardner et al., 2019) showing that pairwise coordination between grid cells is

preserved across brain states. Here, 5/6 modules show toroidal topology during REM and 4/6 during SWS. One rat (S) did not show any organization during sleep. To address this point the authors suggest that the grid cells in rat S (and in module R1) are not of the same kind as the others: they are characterized by different theta modulation and ISI. To be honest, the manuscript becomes extremely unclear at this point. The authors should consider changing the presentation of these classes, and perhaps just focus on REM sleep.

We agree that the complexity of the study is elevated in Fig. 5 and that the findings should have been presented more clearly here. We do not agree, however, that a solution might be to remove the SWS data. Preserved topology in the absence of sensory inputs, such as in SWS and REM sleep, is among the strongest predictions of CAN models for grid cells, and the strength of the present paper is that we put this prediction to test, and that the data support it. Yet there is one single module where significant toroidal topology was not obtained in REM sleep, and there are two modules where it was not obtained in SWS. We found that these grid modules consist of classes of grid cells with different temporal firing properties (“bursty”, “non-bursty”, and “theta-modulated”) and that two of the classes (non-bursty and theta-modulated) – both quite rare in the overall grid population where we recorded – lack a clear toroidal topology. If analysis is confined to the major class of grid cells (the bursty subtype), toroidal topology appears in the modules that had an adequate cell number in this class, even the one where it was not previously obtained in SWS. Thus, while the majority of grid cells operate on strict toroidal manifolds, there are smaller subpopulations with weaker tuning. These cell populations can be identified *a priori* based on burst-firing and theta-modulation characteristics. The heterogeneity in the grid-cell population is important to include in the presentation for a full understanding of grid-cell CAN dynamics. Thus, instead of removing SWS entirely, we have followed the Reviewer’s alternative suggestions regarding how this part of the presentation can be simplified and clarified (please see the next comment) and completely rewritten the section about subtypes and reanalysed and redesigned Fig. 5.

If the authors decide to present the grid cell class analysis, they must address the following concerns. First, they did not record actual LFP, and theta phase and amplitude are obtained from population activity. But then, the problem becomes circular: if different cell types show different phase preference to theta, how come the theta phase extracted from the population is not itself biased by the relative ratio of grid classes recorded in a given session? This conundrum seems almost impossible to solve. The authors should adopt a more rigorous approach: quantify theta modulation strength from spike train autocorrelograms and forget about the absolute phase. Combined with ISI, this should be sufficient to cluster the data in different classes.

This is a valid point and we thank the Reviewer for addressing it. Indeed, it is reasonable to assume that the phase of the overall population theta rhythm is biased by the local composition of classes of grid cells. Determining the theta phase relationship of individual cells might then become circular. This problem would apply not only to the population unit activity but also to theta rhythm in the LFP, which is a reflection of the population activity. Based on the concerns about circularity, we have followed the Reviewer’s recommendation and entirely removed theta phase from the class analysis. As now shown in the revised version of Fig 5, the classification of grid cells may be performed exclusively on the dissimilarity between the normalized spike train autocorrelograms of all grid cells (clustering the cosine distances between pairs of autocorrelograms, as shown in the distance matrix in Fig. 5b). In comparing autocorrelograms of different cells, we could identify clusters (panel b) with differences in both burstiness (panel d, first peak) and theta modulation (panel d, second peak). We have updated the description in the main text, in Fig. 5 and in the Methods.

The new clustering identified three classes of grid cells: “bursty”, “theta-modulated” and “non-bursty”, which overlapped strongly with the previously termed classes C1-C3. These classes are

separated well by both the autocorrelogram peak lag (latency to the first peak) and the spike waveform width (shown in two new panels: Fig. 5ef), suggesting they may be different functional classes. The “theta-modulated” class contains a large fraction of the conjunctive grid cells. The two modules where toroidal topology is not obtained during SWS (R1 and S1) contains a substantial proportion of “non-bursty” cells, along with some “theta-modulated” cells.

Finally, in addition to changing the criteria for classification, and the naming of the cell classes, we have removed the final paragraph on this topic in the Discussion as it came as an anticlimax and distracted from the main result. Instead we have inserted a shorter paragraph on the topic in the middle of the Discussion. We hope this helps improving the flow and getting across the significance of the findings in a better way.

Second, the authors should reconcile these findings with their previous report of preserved correlations during sleep. Importantly, are grid cells in R1 and rat S coordinated and, for some reasons, do not show a toroidal structure during sleep?

To reconcile findings with those of Gardner et al (2019), we have performed a similar analysis of pairwise correlation analyses here. We chose not to use the GLM-based coupling measure used in the 2019 paper, because this method assumes there to be a single “population firing rate” which can be estimated from multi-unit activity and hence used to remove the contribution of common inputs shared across the population. This approach would not be suitable here because the results in the present study show that the co-fluctuation of grid cells’ firing rates is organized differently across cell classes (Extended Data Fig. 7b), especially for the “theta-modulated” versus “non-bursty” classes. Therefore, we instead opted to quantify pairwise correlations using the more conventional Pearson correlation.

The new pairwise correlation analysis shows that while there is still strong internal correlation structure (“coordination”) in modules R1 and S1 during slow-wave sleep, this correlation structure cannot be as well explained by the cells’ grid tuning. In these modules, cells from the theta-modulated and non-bursty classes exhibited a positive bias in their pairwise correlations (apparent in the correlation matrix in Extended Data Fig. 7b), indicating a larger amount of global modulation of the firing rates in these two populations. Furthermore, cells from the theta-modulated class (which is rich in conjunctive grid \times head direction cells) had pairwise correlations which correlated more strongly with the cells’ HD tuning than with their grid tuning. We also note that there is little correlation structure between the theta-modulated and non-bursty classes (evident from the small r values in the correlation matrix in Extended Data 7b). Collectively, these factors mean that correlation structure within modules R1 and S1 is less coherently “grid-like” than in the other modules which consist overwhelmingly of bursting grid cells.

A summary of these results has been added as new panel c in Extended Data Fig. 7. In order to show pairwise correlations for all cells, we are presenting the data in 3 \times 3 matrix consisting of cell types B, T and N, with correlation values indicated by colour of the matrix elements. It can be seen that for R1 and S1, during slow-wave sleep, the correlation between cell pairs’ grid phases and their spike-time correlations are weaker for theta-modulated cells than non-bursty and particularly bursty cells. This drop is explained by a stronger correlation with head direction, suggesting, as expected in conjunctive cells, that head direction accounts for much of the variation in these cells, unlike the other classes. Furthermore, the median spike correlation for pairs of theta-modulated and non-bursty cells is higher than for bursty cells, indicating a stronger positive correlation bias, consistent with more global fluctuations of activity in these populations.

These data are entirely consistent with the topological analyses, as pointed out in the legend for the new figure panel (Extended Data Fig. 7).

Below we have included more detailed plots showing the relationship between pairwise correlations and toroidal / HD phase for all cell-class combinations in Rebuttal Fig. 5.

Rebuttal fig. 5a

Pairwise correlations of firing rates during SWS, for pairs of grid cells from module R1, as a function of toroidal (left) and HD (right) phase offset. Cell pairs were selected according to their assigned class (“bursty”, “theta-mod.”, “non-bursty”). Each dot indicates the correlation (Pearson r-value) of the two cells’ firing rates, smoothed with a 15-ms Gaussian window, plotted at the x-axis location corresponding to the toroidal or HD phase offset of the cell pair.

Rebuttal fig. 5b

Same as Rebuttal Fig. 5a, but showing data from module S1.

Third, the authors did not mention or analyze the effect of the slow oscillation on population activity. This can well be an explanation of why some modules do not show a toroidal structure during SWS (as the fluctuation between DOWN and UP states introduces a non-specific coordination, biasing pairwise correlations towards positive values). The same authors had adopted an interesting approach in their previous paper to address this problem, it is unfortunate they did not analyze SWS data more rigorously in the present manuscript.

Slow oscillations are taken care of by the fact that only high-activity periods (UP states) are used. Samples from DOWN states are discarded, so there should be no artefactual coordination. We have now spelled this out more clearly in the Methods section (“Preprocessing of population activity for topological analyses”) and we have added annotations to Extended Data Fig. 8a which explain the downsampling procedure. As shown, the population activity vectors (time bins) were first downsampled to a set of 15,000, consisting of the vectors with the highest mean firing rate in the module. Among these vectors, 1,200 were retained after a density-based method was used to reduce the data to a representative point cloud (as was and is further explained in the Methods). It can be seen from the figure that all of the selected population activity vectors occurred during UP states. We have made a remark on this selectivity in the figure legend.

Finally, the authors mention that the different classes of grid cells showed different modulation with micro-arousal. This is potentially interesting but quite disconnected from the rest of the manuscript. I suggest to either fully characterize this phenomenon or to get rid of this observation.

Since the manuscript is too long, we have chosen to remove the micro-arousal data, as suggested by the Reviewer. We agree that they are somewhat disconnected from the rest of the study. We hope Fig. 5 and the class analysis are now more accessible, after we have (i) removed these data, (ii) included all grid cells in the clustering, and (iii) identified classes solely based on the spike train autocorrelograms, and introduced functional names for the classes.

Minor concerns:

What is the exact measure of spatial information? "Bits/s" is confusing, is it bit per spike or bit per second? Methods suggest that this is bit per spike (which it should be).

Bits/s means bits/second, in accordance with SI unit nomenclature. We agree that when comparing (classes of) different cells, bits/spike would be a more correct measure. Thus, we have changed the measure to be consistent throughout (as can be seen, the results are not changed).

The authors should indicate in which brain state the correlation matrix in 5b was computed.

The original Fig. 5b has now been replaced with a new distance matrix, but we have followed this recommendation and have clarified that we used the spike trains from the open field session for the new analysis.

Referee #3 (Remarks to the Author):

This paper is elegantly done and interesting. The observation of toroidal topology in grid cell responses is important and of interest to a wide audience. The number of recordings is impressive and the analysis skillful. My only technical comment is that, unless I missed it, the use of 6 PCs does not seem to be well justified and the effect of changing this number is not well investigated.

As explained in response to a similar comment by Reviewer 1, PCA preprocessing was added in order to de-noise the data, i.e. to reduce the influence of noisy fluctuations in neuronal firing rates. We have considerably extended the analysis to show in Extended Data Fig. 3 (b, c and d) that the population structure is consistently (across all modules) accounted for by the 6 first principal components, which in the open field correspond strongly with the animal's 2-D location and have a grid-like representation, unlike the higher principal components (which may be related to 'other' correlates of the activity, such as theta modulation or head-direction influences). In order to focus on the dominant features of the data, in particular those related to the grid structure of the cells, we chose to perform the further topological analysis on those components, while we reduce the influence of 'other' contributions that may not be identifiable with the current size of data.

We appreciate the concern that the choice of 6 PCs may seem a bit arbitrary. As explained in our response to Reviewer 1, in the original version of the figure, we showed explained variance for only one example recording (panel b). In the revised version, we show explained-variance curves for all modules during OF, REM and SWS (Extended Data Fig. 3b). Moreover, for each module, we have added colour maps illustrating the PCs' mapping onto the environment (Extended Data Fig. 3c), and we have quantified the spatial selectivity of this mapping by computing explained deviance scores for each map (Extended Data Fig. 3d). It can be seen from these new figures that (i) the first 6 components stand out, with a sharp drop in explained variance from the 6th to the 7th PC, across modules and behaviour conditions, (ii) these 6 components map most strongly onto the open-field environment, and (iii) these 6 components show grid-like patterns in the open field, while higher

components typically do not. In addition to presenting the full set of data with quantitative analysis, we have further explained in the legend the rationale for choosing the first 6 components and discarding the higher ones. Moreover, we have now included in Extended Data Fig. 3e an analysis of varying the number of components for an example module, and see that 6 components gives the largest discrepancy between the two longest-lived H1 bars and the third-longest-lived H1 bar (a heuristic showing when the toroidal topology, with two long-lived H1 bars, is “clearest”).

We further refer the Reviewer to the discussion of why preprocessing was conducted in our response to Reviewer 1 (the curse of dimensionality issue, including the lack of adequate sampling of the representational space that would occur with too many dimensions).

I do have a broader comment. The paper starts in the abstract with a bold statement that grid cells are explained by CAN dynamics. This is exciting, but I found that as I read the paper I got progressively more confused about what actual message is being conveyed. First, grid cells are only a small fraction of the recorded neurons, and it has been claimed that there is no real distinction between grid cells and other MEC neurons. In other words, the cells being discussed here are the tail of a continuous distribution.

We are aware that it has been claimed by some researchers that entorhinal neurons generally exhibit mixed selectivity and that grid cells are the tail of a mixed continuous distribution. We think this is simply wrong. The claim can be traced back to GLM analyses where most recorded MEC neurons were shown to be modulated by multiple factors, including position, head direction, speed and theta modulation¹⁷. While this finding confirms earlier observations of conjunctive coding of position and head direction^{18,19} as well as weak influences of speed in grid and head direction cells^{18,20}, the GLM findings have been presented in some contexts as evidence that entorhinal cortex does not consist of functionally discrete cell types. This has been extended to include the claim that firing patterns of position-modulated cells such as grid cells, border cells and object vector cells are distributed in the population, beyond these cells themselves, with different degrees of tuning in different cells. However, all MEC recording studies show that while mixing (conjunctive coding) is widespread, the mixing is between position, direction and speed (as shown in ref. 17 too). Different types of position signals have not been shown to overlap (combinations of positional cells were indeed not tested in ref. 17). The present topology analyses confirms that grid cells are distinct by showing very clearly that these cells (and modules of grid cells) stand out completely from the rest of the cell population cloud. If grid cells were merely the tail of a continuous distribution, we would not see so consistent and strong tuning to a toroidal manifold.

While this discussion is somewhat orthogonal to the testing of toroidal topology in grid cells, the UMAP clustering provides an opportunity to remind readers about the distinctness of grid cells and grid modules. We have therefore added text in the section on identification of grid cells and grid cell modules (under “Visualization of toroidal manifold”), as well as in the legend of Extended Data Fig. 2, where we emphasize that clusters did not overlap with each other or with residual non-grid cells. The significance of this observation for the claim that grid cells are functionally distinct is noted in the figure legend for panel 2a.

Whether or not the authors agree with this, they are certainly talking about a small set of selected cells. Then, to the authors' credit, they point to a gradation in the evidence for toroidal topology across modules. This leads to the conclusion that more bursty grid cells are the most CAN like. The last line of the paper points out that cells can be grid-like even without showing CAN dynamics. This sentence is a far cry from the ending of the abstract. So what exactly IS the message: a subset of grid cells which are a small subset of MEC cells show CAN dynamics?

We are sorry for the confusion that we generated in the previous version. Indeed, the message is not that “a small subset of MEC cells show CAN dynamics”. Instead, the message is that the vast majority of grid cells show CAN dynamics. Only a small minority of cells in two of the grid modules failed to demonstrate significant tuning to the torus during sleep (while toroidal topology was still present in the awake states also in these modules). We realize that our presentation of the data in Fig. 5 was convoluted (see Reviewer 2) and that the small size of the deviating subpopulation may not have come through in our presentation. We have now shortened and simplified the description of this part of the study (related to Fig. 5), as outlined in our response to the related comment by Reviewer 2. We have also pointed out in the revised Discussion (2nd paragraph) that the majority of grid cells (the bursting subtype) operate on a toroidal manifold and that the non-bursting and theta-modulated subsets with weaker toroidal tuning are smaller in size (as shown for 5 out of 6 modules in the new Fig. 5c panel). As for the Abstract, because toroidal tuning is the dominant response among grid cells, we feel that the concluding sentence of the original Abstract is accurate and can stand (“This demonstration of network dynamics on a toroidal manifold provides the first population-level visualization of CAN dynamics in grid cells.”).

If the Reviewer instead (or additionally) refers to the proportion of grid cells compared to other cells, we are not quite sure we understand why this would be an issue. The paper sets out to test a theoretical model of the origin of grid patterns and finds evidence for one of its strongest predictions. Whether the grid pattern to be explained is expressed by 10%, 20% or 50% of the total MEC cell population does not really matter in our view, but we do agree that it would be useful background for readers to know what proportion of the MEC cells, at the recording locations, we are talking about. Although the fraction can be inferred from the numbers of grid cells compared to total cells in the extended data figures (and now in the tables in Extended Data Fig. 2g-h), we are now in the revised paper pointing out explicitly the proportion of grid cells in the six modules (from 7.8% to 25.6 %; first paragraph of Results: Visualization...). In addition, Fig. 5c shows that in 5 out of 6 modules, the majority of recorded cells are of the bursting class, which exhibited toroidal tuning. Hopefully these changes clarify that we are not describing a small set of selected cells. We are investigating a distinct cell population and the vast majority of cells within that population.

As I said, the authors should be praised for raising the issues commented upon in the previous paragraph. I think the paper would be improved with some clarity along the lines discussed above, but also with some ideas about how cells sitting next to each other can show such a high range of CAN dynamics. Are the CAN-like cells particular cells that are more strongly coupled to the network than other MEC cells? Is the only correlate of CAN dynamics other than topology burstiness? I guess what I am looking for here, and what I think the readers would greatly benefit from, is a discussion of the WHOLE circuit, how a CAN is embedded in it, and what the difference is between a toroidal, CAN-like grid cell, a non-CAN-like grid cell and a non-grid MEC cell. A lot to ask, I appreciate, but it would definitely help clarify the broader point of the paper.

The Reviewer is raising a number of interesting questions: how do grid cells with strong CAN dynamics differ from grid cells with weaker dynamics, which cell types are involved, where in the MEC-PaS circuit are they located, what inputs do they receive, how are they coupled (do the non-bursty ones have weaker recurrent connections?), and how do they connect with the rest of the MEC network and its many other cell types? These are questions that we will address in the years to come. We have added pointers to these targets for future research at the end of the final paragraph of the revised Discussion.

Note that we have also added panel e in Fig. 5, which by showing that the three cell classes differ in waveform shape suggest that these cells differ in morphology and/or biophysical properties.

As for correlates, yes burstiness is a distinct correlate but not the only one (see the new version of Fig. 5, the totally revised text regarding cell classes, and the Discussion under Review 2).

Referee #4 (Remarks to the Author):

An influential class of network models suggest that grid cells form a continuous attractor network, characterized by a toroidal topology. These models explain many features of grid cell activity, but their core aspect – i.e. the toroidal topology of grid cells activity on the population level, was never tested experimentally. In this work, Gardner et al. were able to record a large number of grid cells simultaneously, by conducted high-density electrophysiological recordings using Neuropixels probes in the medial entorhinal cortex of rats. By employing advanced topological analysis of neural activity, the authors demonstrated that the collective activity of grid cells during navigation in open field arenas has a toroidal topology, as predicted by continuous attractor models. The authors did not stop there, but also measured grid cell activity in physical environments, where the grid code is distorted by the environmental boundaries, and during sleep, when neuronal activity is

decoupled from the positional cues. Under all these conditions, grid cells preserved their toroidal topology, which suggests that the toroidal manifold is derived from the intrinsic network

organization of grid cells in the entorhinal cortex. Taken together, these findings represent a major evidence in favor of the continuous attractor model of grid cells.

I found these results extremely important and novel. Gardner et al. very elegantly utilized the power of population recordings to extract the internal organizational principle of grid cell network activity. My main comments/questions pertain to the dynamics of the “bump of activity” on the toroidal manifold that the authors found. I believe that adding the analyses, which I outlined in my comments below, will make the paper even stronger. Overall, I think that this beautiful work represents a major contribution to the field of neuroscience in general, and I strongly recommend this article for publication.

The focus of the present paper is on the structure or topology of the population activity in grid cells. In the specific comments below, the Reviewer raises a number of exciting questions about how the findings can be extended to understand the dynamics of activity on the torus. When writing the first version of the paper, we chose from the outset to stay away from questions about the dynamics because of all the new questions that came up and because we could not fit more into the paper. Questions about dynamics included the very same ones as raised here by Reviewer 4, such as how the bump moves around on the torus during sleep, whether it is smooth or jumps between specific ‘gravity points’, and if it is smooth whether the bump moves at a speed comparable to that of the bump in the running state (Reviewer Comment 4.2); whether the bump moves comparably on manifolds of different modules and whether a dissociation or coordination across modules is seen in sleep (Reviewer Comment 4.3); if there is a relationship between dynamics on the torus for different modules in relation to the animal’s speed (Reviewer Comment 4.4); what happens to the bump during moments of immobility (Reviewer Comment 4.5); and what is the vector field of movements on the torus (the distribution of trajectories; Reviewer Comment 4.6). The exciting questions that are suggested here contain work for several additional papers. In principle, we would have been happy to include current or future work on these topics ***if*** it had strengthened the present story. However we believe that it would not, simply because (i) Comments 4.2-4.6 address dynamics, not

structure, and responding to Comments 4.2-4.6 would not merely add a figure panel or two; they are major questions that each require their own controls and follow-up analyses. Uncertain about how to respond, we contacted the Editors, who agreed that the analyses suggested in 4.2-4.6 are outside of the scope of the paper and advised we do not provide additional data to address them. We have followed their advice but would like to thank the Reviewer for interesting suggestions that align well with work that we have planned for the years to come.

Major Comments

1. Most grid cells have single fields when plotted on the inferred torus, which implies a single bump of activity, as predicted by the continuous attractor models. However, some cells in Extended Data Figure 9 do show multiple fields on the torus (and these fields do not seem to be just trivial repetitions of a single field because of the boundary conditions).

I think it would be important to test explicitly for the existence of a single bump on the population level and to characterize its profile on the manifold (across the different environments and behavioral states). Could it be that in addition to the main bump, there are smaller bumps away from the main bump, which might explain why some cells may appear to have more than one field on the torus?

We thank the Reviewer for this comment and agree that the examples with multiple fields on the torus should not go uncommented. The first point we want to make here is that there are very few multi-peaked grid cells. In Rebuttal Fig. 6, we have tried to estimate the proportion of multi-peaked grid cells. It should be noted that we are using a very small amount of smoothing on the toroidal rate maps, which makes noisy fluctuations easy to see. Thus, we checked how the proportion varies with the spatial smoothing on the 50x50-binned toroidal rate maps.

Rate maps of all cells whose toroidal rate maps were classified as multi-peaked when using the standard smoothing width of 3 bins. Each peak detected on the toroidal rate map is indicated with a different colour. The text above each toroidal rate map gives the Q-value (“Q”) and the number of bumps detected (“B”). Note that in some cases, a bump may be split across the boundaries and is only counted once, but colored differently (e.g. the red and purple bumps given in the leftmost figures of S1 and Q1).

With no smoothing, we detect approximately 60% of the total number of cells to be multi-peaked. However, the number of multi-peaked cells decreases to less than 16% in each condition and module with a smoothing width of 2 bins, and to less than 5% in each case with a smoothing width of 4 bins. In Rebuttal Fig. 7, we additionally provide both the toroidal and the spatial rate maps of all multi-peaked cells in each module during open field foraging, using the smoothing width applied in the paper, and see no apparent pattern in their field locations.

One likely reason why some cells show multiple peaks might be cluster contamination. When we quantified the number of peaks in each rate map, we found that the best-isolated clusters in the kilosort spike sorting (indicated by a low 'Q' value) have the largest fraction of single-peaked rate maps, suggesting that to some degree the few multiple-peaked rate maps are artifacts. The results of this analysis are shown in Rebuttal Fig. 6 (top right).

Moreover, we show in Supplementary Video for Reviewers that the moment-to-moment dynamics of the mean population activity for R1, R2 and R3 in OF seem to correspond to a single moving bump.

To sum up, by introducing methods for measuring the number of fields, we find that the number of cells with more than one field is strikingly low. For the very few cells that come out with more than one field, we suspect that clustering errors may account for much of it. In the revised version of Extended Data Fig. 3, in panel g, we now include the proportion of cells with more than one field as a function of smoothing width. The field detection methods and new smoothing parameters are reported in the Online Methods "Toroidal peak detection".

Rebuttal fig. 7

Rate maps of all cells whose toroidal rate maps were classified as multi-peaked when using the standard smoothing width of 3 bins. Each peak detected on the toroidal rate map is indicated with a different colour. The text above each toroidal rate map gives the Q -value (“ Q ”) and the number of bumps detected (“ B ”). Note that in some cases, a bump may be split across the boundaries and is only counted once, but colored differently (e.g. the red and purple bumps given in the leftmost figures of S1 and O1).

What is the profile of the bump? Is it symmetric? Does it become skewed along the movement direction?

Skewness of the bump would be interesting in the context of how the bump moves on the torus (is the bump skewed in the direction of movement?). For the reasons outlined above, we have chosen not to analyse this question.

2. What are the dynamics of the bump during sleep? Does it move smoothly on the toroidal manifold, or are there moments when it “jumps” abruptly? Is there any evidence for more than one bump appearing during sleep? Is the speed of movement of the bump comparable to that during running?

Very interesting questions but please see general comment at the beginning of Review 4. After seeking advice from the Editors, we have chosen not to provide analysis on the dynamics of the bump on the torus.

3. Does the bump move coherently on the different manifolds corresponding to different modules across behavioral states, especially during sleep? I expect the answer would be that it moves coherently during running, but independently during sleep (given the previous measurements of pairwise correlations in Gardner et al 2019).

Same as above (4.2). As a side comment, please note that the Gardner 2019 paper identified weaker correlations in inter-module pairs compared to intra-module pairs both during foraging and during sleep. Neither of these results directly tells us whether the dynamics are smooth and coordinated during sleep. This interesting question is the topic of an ongoing separate study. However, we include Supplementary Video for Reviewers which shows the coordination of neural activity across modules R1-R3 during awake navigation in the open field.

4. During running, the bump is expected to move more slowly on manifolds representing modules with larger grid spacing, compared to modules with smaller grid spacing. Is it indeed the case? Does this expected negative relationship between average bump speed and grid spacing exist during sleep? It would be interesting to test if such putative speed-spacing relationship can be observed even if the bump does not move coherently across modules during sleep (see my previous comment). In other words, does the average speed of the bump on the manifold is a characteristic property of the module across behavioral states?

Same as above (4.2).

5. What happens to the bump during moments of immobility, which the authors excluded from the analysis? Does it become pinned on the torus or does it start to drift/disappear?

Same as above (4.2).

6. What is the vector field on the toroidal manifold and in its vicinity? Does it appear to be as expected from CAN models, for example is it identical in all directions along the toroidal surface? A related question: is the toroidal manifold an attractive manifold?

Same as above (4.2).

In other words, does the activity tend to be pulled towards the manifold on those instances when it deviates from it (as in Fig.3 of Chaudhuri et al 2019), especially during sleep? I realize that there might be some circularity in this question: unless this was the case, the authors would not be able to detect the toroidal structure in the first place. But perhaps by inferring the torus using half of the data, the authors can test if the manifold is attractive using the remaining of the data.

This question is also about the dynamics of the activity bump, which we have chosen to leave out of the paper, after advice from the Editors. But as a side comment, we agree with the Reviewer that this is a circular question. We don't think using half of the data would help much. The paper already shows that the toroidal manifold is preserved under much wider conditions than a comparison between the two halves of a given simulation.

7. The authors discarded grid cells with prominent head-direction tuning from the analyses. Head-direction cells are expected to have a ring topology, and therefore conjunctive grid X head-direction cells may form a 3-dimensional torus. I am curious if the authors have enough cells/coverage to see any evidence for such 3-dimensional torus?

This is an excellent question, and within the scope of the paper (structure, not dynamics). As mentioned in the comment to Reviewer 1 regarding analysis of multiple modules as well as conjunctive grid \times head direction cells, the higher complexity (going from a 2-dimensional to a 3-dimensional torus) requires better coverage of the representational space, which in turn requires more neurons. The number of conjunctive cells is only in a couple of cases at similar scale to the number of "pure" grid cells (the former range from 1 to 78 cells, see new table in Extended Data Fig. 2g). Thus, the number of modules for which any interesting analysis can be performed is low. Kang et al¹⁵ suggest that at least ~ 120 conjunctive cells are needed in order to detect the three-torus, way beyond the number of conjunctive cells recorded in the present study.

However, while we lack data to address this interesting question, we have chosen to include in the revised version of Extended Data Fig. 7e-f the barcodes of the theta-modulated class of modules S1 and R1 during SWS. Unlike the other modules, S1 and R1 have a particularly large number of conjunctive cells (see Fig 5c). Interestingly, we find that the ring topology of the head direction tuning is present in both cases, as expressed by a single long-lived $\bar{\nu}$ bar, which, when used to decode OF data, seems to correlate with the recorded head direction (see Extended Data Fig. 7g-h). We feel that presenting those examples is justified but the data are limited and the interpretations are not simple and must await recordings with a larger number of neurons as well as further development of topological analysis tools.

In addition to the examples in Extended Data Fig. 7e-h, we have added toroidal rate maps in Extended Data Fig. 9 in which we use the decoded toroidal positions of the pure grid cells to demonstrate that single toroidal firing fields can also be seen in the conjunctive grid \times head direction cells that we had discarded in the original version of the paper. This is now pointed out in the legend of Extended Data Fig. 9. Moreover, in Fig. 5 we include the conjunctive grid \times head direction cells in the subpopulations of grid cells. These new data show that although toroidal topology did not appear for some classes of grid cells, the various subpopulations maintain some degree of coherence on the decoded torus, suggesting that while a smaller subpopulation of grid cells has weaker toroidal topology, the entire population is still characterized by partial coherence.

8. Could the heterogeneity of grid cells reported in Figure 5 merely reflect the fact that the activity of some grid cells is modulated by other signals, in addition to grid tuning?

I am saying that because if the authors used running epochs to compute the correlation matrix in Figure 5b (the information about what epochs were used for computing correlations is missing from

the methods), then pairwise correlations should be low for most grid cells, except for those that have the same phase.

In fact, the pairwise correlation matrix was from SWS, not running, for similar reasons that we focused mainly on SWS pairwise correlations in the Gardner et al 2019 paper. We chose SWS for this analysis because the absence of theta oscillations and a stable hippocampal place-cell map makes it likely that SWS correlation structure more closely reflects the intrinsic connectivity of grid-cell circuits than the correlation structure from OF would. This information was unfortunately not included in the original version of the manuscript. We apologize for that and thank the Reviewer for identifying the omission. The content of Fig. 5 has now changed, but the equivalent correlation matrix is now shown in Extended Data Fig. 7b and we now specify in the figure legend (and main text) that the matrix is computed from SWS epochs.

Therefore, grid cells are not expected to be grouped into an apparent cluster, unless their firing is modulated by other factors. Indeed, C1 cells do not seem to represent a clearly defined cluster (but rather an “absence of a cluster”) – as would be expected from pure grid cells. In contrast, the much more pronounced C2 and C3 clusters suggest that cells in these clusters are modulated by other factors, which are not related to grid firing (e.g., theta phase). Accordingly, the spatial information in physical space for cells in clusters C2 and C3 appears to be lower than in cluster C1 (Fig. 5d). It is thus might be expected that cells, which are modulated by other factors (in addition to grid tuning), would contribute “noise” (with respect to the grid signal) and thus reduce the chance to reveal the toroidal tuning. It is probably the reason why the authors excluded conjunctive grid X head-direction cells from the topological analysis. Therefore, I wonder, if the clustering analysis/theta modulations reported in Figure 4, could be replaced by a measurement that quantifies directly how much of the firing rate of the cells is explained by grid-like tuning, and whether such measurement will be a more direct criterion for the ability of cells to reveal the toroidal structure.

Although the pairwise correlations in Fig. 5 and Extended Data Fig. 7 were from SWS (not running), the Reviewer is absolutely right that other factors than hexagonal firing might contribute to the firing of especially C2 and C3 cells (corresponding to “non-bursty (N)” and “theta-modulated (T)” cells in the revised manuscript). Lower spatial modulation in C2/N and C3/T cells than in bursty C1/B cells, and the increased correlation with head direction now shown in Extended Data Fig. 7c, would be consistent with this possibility. The suggestion to express how much of the firing is explained by grid-like tuning (= tuning to the torus in the open field) is excellent. In the original manuscript, we reported higher explained deviance and spatial information for toroidal firing in C1 cells than C2 and C3 cells, consistent with the idea that ‘other’ factors contribute noise that make detection of toroidal topology more difficult.

In the revised manuscript, we have reported toroidal explained deviance and information content for each “new” class (B/T/N) (see Extended Data Fig. 7d), as well as cumulative plots of the preservation of toroidal tuning across sessions (see Fig. 5h-j) (we chose to measure grid-like tuning as toroidal tuning to allow the analysis to be performed on sleep data). Interestingly, the toroidal tuning of all three classes is preserved, but the explained deviance and information content of the “bursty” cells is clearly higher than the other two classes, possibly suggesting that the latter are modulated by other factors, including globally coupled fluctuations in firing rate (implied by the larger positive bias in pairwise correlations, see Extended Data Fig. 7c). While many of the “theta-modulated” cells are modulated by head direction (see Fig. 5c and Extended Data Fig. 7c, g-h), it is not clear what the “non-bursty” cells are additionally encoding.

Minor Comments

1. It might be good to include 1-2 sentences in the main text to explain to the non-expert reader why topological analyses were done by first separating grid cells into modules.

This comment is similar to remarks made by Reviewers 1 and to some extent Reviewer 3. Please see our response to Reviewer 1 for the full explanation of (i) the theoretical motivation for expecting CANs to correspond to individual modules, and (ii) why a full N-dimensional analysis is not feasible with the present sample sizes or anything near it (curse of dimensionality). It is also worth noting that simulations¹⁵ suggest that a much bigger environment is needed to extract topology even for just two mixed modules.

Regarding the theoretical motivation, grid modules were analyzed one at a time in order to test central predictions of CAN models for grid cells, which propose that grid modules operate as semi-independent CANs (and so may have independent toroidal topologies). We have now emphasized this in a new concluding sentence of the Introduction.

2. In some cases, the modules that result from the clustering algorithm (Extended Data Figure 2) seem to be very broad in terms of grid spacing and orientation spanned by cells that belong to a given module. For example, cells in what is identified as module “R3” (Extended Data Figure 2c) have grid spacings ranging from ~40 to 140 cm and orientations spanning the entire range of +/-pi. This might suggest that this module was not extracted correctly by the clustering algorithm. Also, in some modules (e.g., Extended Data Figure 2b “R3”, and Extended Data Figure 2c “R2”) there was a considerable number of cells with relatively low grid score (their grid score values were comparable to that of the “main” cloud). The authors may consider applying a subsequent criterion (e.g., a grid score threshold) to “denoise” the modules.

We believe the UMAP clustering method is superior because it is less vulnerable to the errors which the conventional grid parameterization is prone to. Estimating grid spacing and orientation becomes increasingly error-prone with larger grids, which probably is why R3 shows the widest spread. Furthermore, field-to-field rate variability or other irregularities in the grid pattern may interfere with measurements of grid geometry (we note this particularly for UMAP cluster R1b, which expressed high field-to-field rate variability). The same logic applies to the grid score: this measure tends to be unreliable when grid patterns are distorted or noisy, or when field rates are highly variable and approach zero for some of the fields. UMAP is less impacted by these effects because it doesn't make any prior assumptions about grid geometry, apart from that it is periodic. We have added text to better explain the motivation behind the choice of UMAP in the Online Methods “Grid module classification”). Additionally, in Rebuttal Fig. 8 we show that there are cells with low grid scores (particularly in R3) which have high toroidal selectivity, which we believe vindicates our choice not to use the grid score to classify grid cells.

Rebuttal fig. 8

Grid score vs. toroidal explained deviance

Relationship between individual grid cells' toroidal selectivity (y-axis) and their grid score (x-axis). Each plot shows all of the pure grid cells (dots) in each module.

3. The authors reduced the dimensionality of the data by taking the first 6 PCs. By analogy to unimodal circular tuning, if the tuning is wide (cosine tuning) one needs 2 PCs (sine and cosine) to generate all tuning curves with different angular offsets. However, more PCs are needed when the tuning gets narrower. The authors make a related comment in Supplementary Math Note: "If the tuning curve of individual cells is sufficiently wide ... the modes that correspond to the first six PCA components are expected to capture a large fraction of the variance". By extending this argument to grid cells from different modules, I would expect that more PCs are needed to explain the variance of grid cells from modules with smaller grid-spacing, because their tuning is narrower, compared to grid cells from modules with larger grid-spacing.

This is not correct: what matters is the ratio of the field width to the grid spacing, which is more or less similar for all modules; any minor difference would not be sufficient to change the number of PCs needed. In Rebuttal Fig. 9 we have computed this ratio for all of the modules, using autocorrelograms calculated from rate maps based on the spatial coordinates (in OF), as well as from the toroidal coordinates in all conditions. The ratios for the cells in each module suggest that both the toroidal and spatial autocorrelogram field width is fairly equal across modules. For the spatial tuning, this is the case only when accounting for the spatial grid scale.

Rebuttal fig. 9

Detecting the size of the center bump of the spatial autocorrelogram shows that the toroidal bump size is independent of module scale and is equal for the ratio of bump size over module scale. Upper: histograms showing distribution of the circumference of the autocorrelogram center field for pure grid cells in each module. A sample of 1000 points in the 2D-plane was drawn, distributed according to the spatial distribution given by the autocorrelogram, and subsequently clustered by density. The center-field circumference was then calculated as the perimeter of the convex hull of the cluster of points closest to the center. Left: distribution of center-field circumferences for autocorrelograms of toroidal rate maps, middle: same but for autocorrelograms of spatial rate maps, right: same as middle, but with circumference normalized by the field spacing.

Lower: average rate-map autocorrelograms for each module. The top row shows toroidal autocorrelograms; spatial autocorrelograms are shown in the bottom row. The center-field area is marked with a red patch; the blue stippled line indicates the circumference. The circumference for the averaged autocorrelogram is given below each figure. Note the constancy of bump size on the torus across modules, in contrast to the bump size in physical space.

Also, the variance explained by these 6 PCs is rather modest (together they explain only 25% of the variance). While the results presented in the paper are very convincing, I am wondering if the toroidal manifold might be even cleaner if more than 6 PCs are used, or if the number of PCs used for analyses, should be adjusted depending on the spacing of each module.

We now show that the percentage of variance explained by PCs 1-6 is not affected by the module spacing (added in the legend of Extended Data Fig. 3c). Moreover, varying the number of dimensions included in the analysis, we see that the clarity of the toroidal topology is best when only 6 PCs are used (Extended Data Fig. 3e). Please see our reply to similar comments on the number of PCs from Reviewers 1 and 3.

4. *Fig. 1c-d: The activity of example cells appears on the same positions on the torus. Was the toroidal representation rotated to achieve this match? Or does it mean that there was no remapping between OF and WW environments?*

We didn't need to rotate because the positions on the torus during WW were determined using the UMAP transformation previously fitted to the OF data. In all subsequent analyses we did have to align the representations as they were identified separately in each condition to avoid the potential for circular arguments. We have clarified this in the legend of the figure.

However, the toroidal representation found using "cohomological decoding" requires rotation, as these representations are found independently. This has now been further clarified in the "Rhomboidal fitting and toroidal alignment"-section (see below).

5. *I found the section "Cohomological decoding" in the Methods quite hard to follow. Since this is an important part of the analysis, some additional clarifications might be needed for the non-expert reader.*

We have tried to reformulate the text to reach out wider. However, the method is published (Da Silva)⁶, so we refer readers to this source.

Also, it might help to illustrate the "Rhomboidal fitting and toroidal alignment" section in the Methods with a Supplementary Figure showing the results of the fitting procedure.

We do show results of the fitting procedure in Extended Data Fig. 4 but maybe did not refer to them enough. In the revised paper, we have added a panel 4b in which we show how the same grid pattern is explained by different sets of hexagonal axes, to help illustrate how and why we need to reorient the circular decoding to compare the results across conditions and animals. Hopefully, this addition, including rephrasing the text helps clarify this section.

6. *Fig. 3c: Can the fact that grid cells have higher spatial information in toroidal space compared to physical space, be simply because in toroidal space grid cells have a single field, whereas in physical space they have multiple fields?*

The spike information measure of Skaggs et al²¹ estimates the information that a spike from a given cell conveys about the animal's position in the arena. The measure is comparable as the measure depends on the number of bins covering the firing field(s). As shown in Rebuttal Fig. 9, the size of the grid fields in physical space depends on the scale of the grid module and the ratio of the two is practically constant for the six grid modules (Rebuttal Fig. 9 upper right). This suggests that the number of bins describing the firing fields will be constant for the different modules, since the increased size of the fields will "compensate" for the lower number of fields per rate map for the larger-scaled modules. Thus, as long as the discretization captures the relevant firing rate variations (which is the case as the same binning was used to identify the grid cells) and there is an even sampling across bins, then the measure would remain similar if space were enlarged and more firing fields were added. This observation may be extended to the toroidal rate maps, as the measure is invariant to the units used (either for position or for position on the toroidal manifold) and while the toroidal rate maps are single-peaked, the field size is relatively larger (see bottom rows of Rebuttal Fig. 9). Due to all these reasons, if position were precisely mapped to the toroidal coordinates without any drifts or imprecisions, we would expect this measure to be similar for space and for the

toroidal coordinate. In the revised paper, we have added a short remark in the Methods “Information content” on the invariance of spatial information to the number of spatial firing fields.

Supplementary Movie 1 for Reviewers: Toroidal population activity of three grid modules.

Video of mean firing rate across all grid cells in each module, shown as a function of the toroidal map shows a single bump for each module. The leftmost figure shows the recorded spatial trajectory of the rat, the green circle enclosing showing the position at the current time frame, red shows the last 500ms and blue shows all positions visited. The three rightmost figures display the summed contribution of all cells in each module at each time step. Each cell’s contribution is calculated as toroidal rate maps for the 1200 points that went into the persistent cohomology computation, weighted by the cell’s smoothed firing rate at the given time frame. Note that each bump seems to move in the same direction on the toroidal surface, in approximate alignment with the spatial trajectory of the rat, with the bump of R1 moving faster than that of R2 which is faster than that of R3, consistent with the grid module spacing.

References

1. Aksay, E. *et al.* Functional dissection of circuitry in a neural integrator. *Nat. Neurosci.* **10**, 494–504 (2007).
2. Sussillo, D. & Barak, O. Opening the black box: low-dimensional dynamics in high-dimensional recurrent neural networks. *Neural Comput.* **25**, 626–649 (2013).
3. Darshan, R. & Rivkind, A. Learning to represent continuous variables in heterogeneous neural networks. *bioRxiv* 2021.06.01.446635 (2021) doi:10.1101/2021.06.01.446635.
4. Sorscher, B., Mel, G. C., Ocko, S. A., Giacomo, L. & Ganguli, S. A unified theory for the computational and mechanistic origins of grid cells. *bioRxiv* 2020.12.29.424583 (2020) doi:10.1101/2020.12.29.424583.
5. De Silva, V., Morozov, D. & Vejdemo-Johansson, M. Dualities in persistent (co) homology. *Inverse Probl.* **27**, 124003 (2011).
6. De Silva, V., Morozov, D. & Vejdemo-Johansson, M. Persistent cohomology and circular coordinates. *Discrete Comput. Geom.* **45**, 737–759 (2011).
7. Perea, J. A. Multiscale projective coordinates via persistent cohomology of sparse filtrations. *Discrete Comput. Geom.* **59**, 175–225 (2018).
8. Contessoto, M., Mémoli, F., Stefanou, A. & Zhou, L. The persistent cup-length invariant. *ArXiv Prepr. ArXiv210701553* (2021).
9. Bellman, R. Dynamic programming. *Science* **153**, 34–37 (1966).
10. Bellman, R. E. *Adaptive control processes*. (Princeton university press, 2015).
11. Keogh, E. & Mueen, A. Curse of Dimensionality. *Encyclopedia of Machine Learning and Data Mining* 314–315 (2017).
12. Aggarwal, C. C., Hinneburg, A. & Keim, D. A. On the surprising behavior of distance metrics in high dimensional space. in *International conference on database theory* 420–434 (Springer, 2001).

13. Beyer, K., Goldstein, J., Ramakrishnan, R. & Shaft, U. When is “nearest neighbor” meaningful? in *International conference on database theory* 217–235 (Springer, 1999).
14. Gray, C. M., Maldonado, P. E., Wilson, M. & McNaughton, B. Tetrodes markedly improve the reliability and yield of multiple single-unit isolation from multi-unit recordings in cat striate cortex. *J. Neurosci. Methods* **63**, 43–54 (1995).
15. Kang, L., Xu, B. & Morozov, D. Evaluating State Space Discovery by Persistent Cohomology in the Spatial Representation System. *Front. Comput. Neurosci.* **15**, 616748 (2021).
16. Peyrache, A., Lacroix, M. M., Petersen, P. C. & Buzsáki, G. Internally organized mechanisms of the head direction sense. *Nat. Neurosci.* (2015) doi:10.1038/nn.3968.
17. Hardcastle, K., Maheswaranathan, N., Ganguli, S. & Giocomo, L. M. A Multiplexed, Heterogeneous, and Adaptive Code for Navigation in Medial Entorhinal Cortex. *Neuron* **94**, 375387.e7 (2017).
18. Sargolini, F. *et al.* Conjunctive representation of position, direction, and velocity in entorhinal cortex. *Science* **312**, 758–762 (2006).
19. Solstad, T., Boccara, C. N., Kropff, E., Moser, M.-B. & Moser, E. I. Representation of geometric borders in the entorhinal cortex. *Science* **322**, 1865–1868 (2008).
20. Kropff, E., Carmichael, J. E., Moser, M.-B. & Moser, E. I. Speed cells in the medial entorhinal cortex. *Nature* **523**, 419–424 (2015).
21. Skaggs, W., Mcnaughton, B. & Gothard, K. An information-theoretic approach to deciphering the hippocampal code. *Adv. Neural Inf. Process. Syst.* **5**, 1030–1037 (1992).

Reviewer Reports on the First Revision:

REF #1

The rebuttal is very thorough and has addressed all my questions and concerns. I am satisfied with the changes to the manuscript, and am happy to recommend the paper for publication.

REF #2

I thank the authors for carefully addressing all my concerns. The manuscript has been greatly improved and is much clearer now (especially Fig. 5). The authors have well justified their decision not to include or test some of the suggestions. I don't have any further comments and, in my opinion, the paper is now suitable for publication.

REF #4

I understand the authors' decision to conduct the analyses of dynamics of activity on the torus as part of a separate future work, rather than including them in the current manuscript. I would like to thank the authors for addressing the rest of my comments and revising the manuscript accordingly.

This is an elegant and important work and I am looking forward to seeing it published.